# Finite-Time Analysis of Single-Timescale Actor-Critic

**Xuyang Chen**
National University of Singapore
chenxuyang@u.nus.edu

**Lin Zhao**[*]
National University of Singapore
elezhli@nus.edu.sg

## Abstract

Actor-critic methods have achieved significant success in many challenging applications. However, its finite-time convergence is still poorly understood in the most practical single-timescale form. Existing works on analyzing single-timescale actor-critic have been limited to i.i.d. sampling or tabular setting for simplicity. We investigate the more practical online single-timescale actor-critic algorithm on continuous state space, where the critic assumes linear function approximation and updates with a single Markovian sample per actor step. Previous analysis has been unable to establish the convergence for such a challenging scenario. We demonstrate that the online single-timescale actor-critic method provably finds an $\epsilon$-approximate stationary point with $\widetilde{\mathcal{O}}(\epsilon^{-2})$ sample complexity under standard assumptions, which can be further improved to $\mathcal{O}(\epsilon^{-2})$ under the i.i.d. sampling. Our novel framework systematically evaluates and controls the error propagation between the actor and critic. It offers a promising approach for analyzing other single-timescale reinforcement learning algorithms as well.

## 1 Introduction

Actor-critic (AC) methods have achieved great success in solving many challenging reinforcement learning (RL) problems [17, 20, 24]. AC updates the actor (i.e., the policy) using the estimated policy gradient (PG), which is a function of the Q-value under the policy. Meanwhile, it employs a bootstrapping critic to estimate the Q-value, which often helps reduce variance and accelerates convergence in practice.

Despite the empirical success, the non-asymptotic convergence analysis of AC in the most practical single-timescale form remains underexplored. A large body of existing works consider the double-loop variants, where the critic runs many steps to accurately estimate the Q-value for a given actor [37, 16, 27, 2]. This leads to a decoupled convergence analysis of the critic and the actor, which involves a policy evaluation sub-problem in the inner loop and a perturbed gradient descent in the outer loop. Its finite-time convergence is relatively well understood [16, 37, 35]. Nevertheless, the double-loop setting is mainly for ease of analysis, which is barely adopted in practice. Since it requires an accurate critic estimation, it is typically sample inefficient. In fact, it is unclear whether an inner loop of accurate policy evaluation is really necessary since it only corresponds to one transient policy among many iterations.

Another body of works considers the (single-loop) two-timescale variants [31, 9, 36, 13], where the actor and the critic are updated simultaneously in each iteration with stepsizes of different timescales. The actor stepsize is typically smaller than that of the critic, with their ratio converging to zero as the iteration number goes to infinity. Hence, the actor is updated much slower than the critic. The two-timescale allows the critic to approximate the desired Q-value in an asymptotic way, which enables a decoupled convergence analysis of the actor and the critic. This variant is occasionally

---

[*]Corresponding author

37th Conference on Neural Information Processing Systems (NeurIPS 2023).

Table 1: Comparison with related single-timescale actor-critic algorithms

| Reference | Setting | | Sampling | | Sample Complexity |
|-----------|---------|---------|-------|--------|-------------------|
| | State Space | Reward | Actor | Critic | |
| [21] | Finite | Discounted | i.i.d. | i.i.d. | $\mathcal{O}(\epsilon^{-2})$ |
| [8] | Infinite | Discounted | i.i.d. | i.i.d. | $\mathcal{O}(\epsilon^{-2})$ |
| This Paper | Infinite | Average | Markovian | Markovian | $\widetilde{\mathcal{O}}(\epsilon^{-2})$ |

adopted to improve learning stability. However, it is still considered inefficient as the actor update is artificially slowed down.

In this paper, we consider the more practical single-timescale AC algorithm, which is the one introduced in many works of literature as well as in [25] as a classic AC algorithm. In single-timescale AC, the stepsizes of the critic and the actor diminish at the same timescale. Unlike the aforementioned variants, which have specialized designs aimed at simplifying the convergence analysis, the analysis of the single-timescale AC presents a greater challenge. Due to the substantial errors in critic estimation and the close coupling between the parallel critic update and actor update, the algorithm is more prone to unstable error propagation. It remains unclear under what condition the errors will converge to zero. To study its finite-time convergence, we consider the challenging undiscounted time-average reward formulation [25, 31, 37], which consists of three parallel updates: the (time-average) reward estimator, the critic estimator, and the actor estimator. We keep track of the reward estimation error, the critic error, and the policy gradient norm (which measures the actor error) by deriving an implicit bound for each of them. They are then analyzed altogether as an interconnected system inspired by [21] to establish the convergence simultaneously. Specifically, we identify the (constant) ratio between the actor stepsize and the critic stepsize, below which all three errors will diminish to zero, despite the inaccurate estimation in all three updates (reward estimation, critic, actor).

## 1.1 Main Contributions

We summarise our main contributions as follows:

• We provide a finite-time analysis for the single-timescale AC under the Markovian sampling and prove an $\widetilde{\mathcal{O}}(\epsilon^{-2})$ sample complexity, where $\widetilde{\mathcal{O}}(\cdot)$ hides additional logarithmic terms. We further show that this sample complexity can be improved to $\mathcal{O}(\epsilon^{-2})$ under i.i.d. sampling, which matches the state-of-the-art performance of SGD on general non-convex optimization problems. Our proof clearly shows that the additional logarithmic term under the Markovian sampling is introduced by the mixing time of the underlying Markov chain.

• Our result compares favorably to existing works on single-timescale AC. To our knowledge, the only other results of single-timescale AC in the general MDP (Markov decision process) case are from [8] and [21], both of which obtain a sample complexity of $O(\epsilon^{-2})$ under discounted reward setting. However, both [8] and [21] considered the i.i.d. sampling, where the transition tuples are independently sampled from stationary distribution and discounted state-action visitation distribution. In this paper, we consider the more practical Markovian sampling, where the transition tuples are generated from a single trajectory (see Table 1).

Furthermore, [8] follows an explicit Lyapunov analysis, where they leave a biased term in the critic and eliminated in the actor. Therefore, their proof framework cannot show the convergence of the critic. We instead give a neat proof framework to guarantee convergence for both the critic and the actor. Additionally, [21] only considered the tabular case (finite state-action space) where we allow the state space to be infinite (see Table 1). It is worth emphasizing that moving from a finite state space to an infinite state space takes significantly non-trivial effort in analysis. The analysis in [21] concatenates all state-action pairs to create a finite-dimensional feature matrix, which however becomes impossible in the infinite state space scenario. Consequently, their analysis technique and established results are not applicable in our context.

• Technically, we develop a new analysis framework that can establish the finite-time convergence for single-timescale AC under the general setting. The existing analysis for double-loop AC [37] and two-timescale AC [31] hinge on decoupling the analysis of actor and critic, which typically establishes the convergence of critic first and then actor [37, 31, 8]. We instead investigate the evolution of

the coupled estimation errors of the time-average reward, the critic, and the policy gradient norm altogether as an interconnected system in a much less conservative way. We emphasize that our analysis framework includes the discounted setting as a simple special case where the interconnected system is only two-dimensional without the time-average reward estimation.

## 1.2 Related Work

**Policy gradient methods.** Policy gradient methods [26, 25] learn a parameterized policy, constituting a departure from the value-based approach [28, 32, 40]. The asymptotic convergence of policy gradient methods has been well established in [29, 26, 3, 14] via stochastic approximation methods [6]. Some recent works have shown that PG methods can find the global optimum of some particular class of problems, such as LQR [11, 19] and tabular case problem [1]. Under general function approximation setting, finite-time convergence of PG methods was analyzed in [1, 38, 33, 34]. Specifically, [1] established the finite-time convergence of PG methods under both tabular policy parameterizations and general parametric policy classes. [38] showed that a variant of PG methods can attain an $\epsilon$-accurate stationary point at a sample complexity of $\mathcal{O}(\epsilon^{-2})$, where they adopted Monte-Carlo sampling to find an unbiased estimation of policy gradient. [33, 34] studied the variance reduction PG and acceleration PG.

**Actor-Critic methods.** The AC algorithm was initially proposed by [15]. Later, [14] extended it to the natural AC algorithm. The asymptotic convergence of AC algorithms has been well established in [14, 5, 7, 39] under various settings. Many recent works focused on the finite-time convergence of AC methods. Under the double-loop setting, [37] established the global convergence of AC methods for solving linear quadratic regulator (LQR). [27] studied the global convergence of AC methods with both the actor and the critic parameterized by neural networks. [16] studied the finite-time local convergence of a few AC variants with linear function approximation.

Under the two-timescale AC setting, [31] established the finite-time local convergence to a stationary point at a sample complexity of $\widetilde{\mathcal{O}}(\epsilon^{-2.5})$ under the undiscounted time-average reward setting. [36] studied both local convergence and global convergence for two-timescale (natural) AC, with $\widetilde{\mathcal{O}}(\epsilon^{-2.5})$ and $\widetilde{\mathcal{O}}(\epsilon^{-4})$ sample complexity, respectively, under the discounted accumulated reward. The algorithm collects multiple samples to update the critic. [13] proposed a two-timescale stochastic approximation algorithm for bilevel optimization and the algorithm was subsequently employed in the context of two-timescale AC. [9] established the global convergence of two-timescale AC methods for solving LQR, where only a single sample is used to update the critic in each iteration.

Under the single-timescale setting, [12] considered the least-squares temporal difference (LSTD) update for the critic and obtained the optimal policy within the energy-based policy class for both linear function approximation and nonlinear function approximation using neural networks. [41] studied single-timescale AC on LQR. In addition, [8] and [21] considered the single-timescale AC in general MDP cases, which have been reviewed and compared in Section 1.1.

**Notation.** We use non-bold letters to denote scalars and use lower and upper case bold letters to denote vectors and matrices respectively. Without further specification, we write $x_n = \mathcal{O}(y_n)$ if there exists an absolute positive constant $C$ such that $x_n \leq Cy_n$, for two sequences $\{x_n\}$ and $\{y_n\}$. We use $\tilde{\mathcal{O}}(\cdot)$ to hide logarithm factors. The total variation distance of two probability measure $\mu$ and $v$ is defined by $d_{TV}(\mu, v) := \frac{1}{2} \int_{\mathcal{X}} |\mu(dx) - v(dx)|$. In addition, we use $\mathbb{P}$ to denote a generic probability of some random event.

## 2 Preliminaries

In this section, we review the basics of the Markov decision process, policy gradient algorithm, and single-timescale AC with linear function approximation.

### 2.1 Markov decision process

We consider the standard Markov Decision Process (MDP) characterized by $(\mathcal{S}, \mathcal{A}, \mathcal{P}, r)$, where $\mathcal{S}$ is the state space and $\mathcal{A}$ is the action space. We consider a finite action space $|\mathcal{A}| < \infty$, whereas the state space can be either a finite set or an (unbounded) real vector space $\mathcal{S} \subset \mathbb{R}^n$. $\mathcal{P}(s_{t+1}|s_t, a_t) \in [0, 1]$ denotes the transition kernel. We consider a bounded reward $r : \mathcal{S} \times \mathcal{A} \to [-U_r, U_r]$, which is a

function of the state $s$ and action $a$. A policy $\pi_{\boldsymbol{\theta}}(\cdot|s) \in \mathbb{R}^{|\mathcal{A}|}$ parameterized by $\boldsymbol{\theta}$ is defined as a mapping from a given state to a probability distribution over actions.

The RL problem of consideration aims to find a policy $\pi_{\boldsymbol{\theta}}$ that maximizes the infinite-horizon time-average reward [26, 25, 37, 31], which is given by

$$J(\boldsymbol{\theta}) := \lim_{T \to \infty} \mathbb{E}_{\boldsymbol{\theta}} \frac{\sum_{t=0}^{T-1} r(s_t, a_t)}{T} = \mathbb{E}_{s \sim \mu_{\boldsymbol{\theta}}, a \sim \pi_{\boldsymbol{\theta}}} [r(s, a)],$$

where the expectation $\mathbb{E}_{\boldsymbol{\theta}}$ is over the Markov chain under the policy $\pi_{\boldsymbol{\theta}}$, and $\mu_{\boldsymbol{\theta}}$ denotes the stationary state distribution induced by $\pi_{\boldsymbol{\theta}}$. The existence of the stationary distribution can be guaranteed by the uniform ergodicity of the underlying MDP, which is a common assumption. Hereafter, we refer to $J(\boldsymbol{\theta})$ as the time-average reward (or exchangeably, performance function), which can be evaluated by the expected reward over the stationary distribution $\mu_{\boldsymbol{\theta}}$ and the policy $\pi_{\boldsymbol{\theta}}$.

The state-value function is used to evaluate the overall rewards starting from a state $s$ and following policy $\pi_{\boldsymbol{\theta}}$ thereafter, which is defined as

$$V_{\boldsymbol{\theta}}(s) := \mathbb{E}_{\boldsymbol{\theta}}\big[\sum_{t=0}^{\infty} (r(s_t, a_t) - J(\boldsymbol{\theta}))|s_0 = s\big],$$

where the action follows the policy $a_t \sim \pi_{\boldsymbol{\theta}}(\cdot|s_t)$ and the next state comes from the transition kernel $s_{t+1} \sim \mathcal{P}(\cdot|s_t, a_t)$. Similarly, we define the action-value (Q-value) function to evaluate the overall rewards starting from $s$, taking action $a$, and following policy $\pi_{\boldsymbol{\theta}}$ thereafter:

$$Q_{\boldsymbol{\theta}}(s, a) = \mathbb{E}_{\boldsymbol{\theta}}\big[\sum_{t=0}^{\infty} (r(s_t, a_t) - J(\boldsymbol{\theta}))|s_0 = s, a_0 = a\big]$$
$$\overset{(i)}{=} r(s, a) - J(\boldsymbol{\theta}) + \mathbb{E}[V_{\boldsymbol{\theta}}(s')],$$

where the expectation in (i) is taken over $s' \sim \mathcal{P}(\cdot|s, a)$.

## 2.2 Policy gradient theorem

The policy gradient theorem [26] provides an analytic expression for the gradient of the performance function $J(\boldsymbol{\theta})$ with respect to the policy parameter $\boldsymbol{\theta}$, which is given by:

$$\nabla_{\boldsymbol{\theta}} J(\boldsymbol{\theta}) = \mathbb{E}_{s \sim \mu_{\boldsymbol{\theta}}, a \sim \pi_{\boldsymbol{\theta}}} [Q_{\boldsymbol{\theta}}(s, a) \nabla_{\boldsymbol{\theta}} \log \pi_{\boldsymbol{\theta}}(a|s)]. \tag{1}$$

Evaluating this gradient requires the Q-value corresponding to the current policy $\pi_{\boldsymbol{\theta}}$. The REIN-FORCE [29] is a Monte Carlo-based episodic algorithm, which uses all the rewards collected along the sample trajectory (that is, the differential return) as an approximation to the true Q-value.

Note that for any function $b : \mathcal{S} \to \mathbb{R}$ that is independent of the action, we have

$$\sum_{a \in \mathcal{A}} b(s) \nabla \pi_{\boldsymbol{\theta}}(a|s) = b(s) \nabla (\sum_{a \in \mathcal{A}} \pi_{\boldsymbol{\theta}}(a|s)) = b(s) \nabla 1 = 0.$$

Therefore, the policy gradient theorem can be written equivalently as:

$$\nabla J(\boldsymbol{\theta}) = \mathbb{E}_{s \sim \mu_{\boldsymbol{\theta}}, a \sim \pi_{\boldsymbol{\theta}}} [(Q_{\boldsymbol{\theta}}(s, a) - b(s)) \nabla_{\boldsymbol{\theta}} \log \pi_{\boldsymbol{\theta}}(a|s)],$$

where $b(s)$ is called the *baseline* function. A popular choice of *baseline* is the state-value function, which leads to the following advantage-based policy gradient

$$\nabla_{\boldsymbol{\theta}} J(\boldsymbol{\theta}) = \mathbb{E}_{s \sim \mu_{\boldsymbol{\theta}}, a \sim \pi_{\boldsymbol{\theta}}} [\Delta_{\boldsymbol{\theta}}(s, a) \nabla_{\boldsymbol{\theta}} \log \pi_{\boldsymbol{\theta}}(a|s)],$$

where $\Delta_{\boldsymbol{\theta}} = Q_{\boldsymbol{\theta}}(s, a) - V_{\boldsymbol{\theta}}(s)$ is known as the advantage function. This is the "REINFORCE with baseline" [29]. The baseline function can help reduce variance. However, like all Monte Carlo-based methods, it can still suffer from high variance and thus learns slowly. In addition, it is inconvenient to implement the algorithm online for continuing tasks [25].

AC algorithm instead employs a bootstrapping critic to estimate the Q-value. We describe the classic single-timescale AC in the next subsection.

## 2.3 The single-timescale actor-critic algorithm

We consider the classic single-sample single-timescale AC method, where the critic is bootstrapping and uses a single sampled reward to update in each iteration. This directly accommodates online learning for continuing tasks. We consider the following linear function approximation of the state-value function:

$$\widehat{V}_{\boldsymbol{\theta}}(s; \boldsymbol{\omega}) = \boldsymbol{\phi}(s)^\top \boldsymbol{\omega},$$

where $\boldsymbol{\phi}(\cdot) : \mathcal{S} \to \mathbb{R}^d$ is a known feature mapping, which satisfies $\|\boldsymbol{\phi}(\cdot)\| \leq 1$. To drive $\widehat{V}_{\boldsymbol{\theta}}(s; w)$ towards its true value $V_{\boldsymbol{\theta}}(s)$, the semi-gradient TD(0) update is applied to estimate the linear coefficient $\boldsymbol{\omega}$ (hereafter referred to as the critic):

$$\boldsymbol{\omega}_{t+1} = \boldsymbol{\omega}_t + \beta_t[(r_t - J(\boldsymbol{\theta}) + \boldsymbol{\phi}(s_{t+1})^\top \boldsymbol{\omega}_t - \boldsymbol{\phi}(s_t)^\top \boldsymbol{\omega}_t)]\boldsymbol{\phi}(s_t), \tag{2}$$

where $\beta_t$ is the step size of the critic $\boldsymbol{\omega}$ and $r_t := r(s_t, a_t)$. Since $J(\boldsymbol{\theta})$ is unknown, the time-average reward setting introduces an additional estimator $\eta$ to estimate it. Hereafter, we simply refer to $\eta$ as the reward estimator. The temporal difference error can be defined as

$$\delta_t := r_t - \eta_t + \boldsymbol{\phi}(s_{t+1})^\top \boldsymbol{\omega}_t - \boldsymbol{\phi}(s_t)^\top \boldsymbol{\omega}_t.$$

Then, the update rule for the critic is given by

$$\eta_{t+1} = \eta_t + \gamma_t(r_t - \eta_t),$$
$$\boldsymbol{\omega}_{t+1} = \boldsymbol{\omega}_t + \beta_t \delta_t \boldsymbol{\phi}(s_t),$$

where $\gamma_t$ is the step size of the reward estimator $\eta_t$.

Since $\delta_t$ is an approximation of the advantage function, similar to REINFORCE with baseline, the corresponding update rule for the actor can be written as:

$$\boldsymbol{\theta}_{t+1} = \boldsymbol{\theta}_t + \alpha_t \delta_t \nabla_{\boldsymbol{\theta}} \log \pi_{\boldsymbol{\theta}_t}(a_t|s_t),$$

where $\alpha_t$ is the actor stepsize. The above-described AC is summarized in Algorithm 1, which is introduced in [25] as a classic online one-step AC algorithm. Algorithm 1 can be efficiently implemented for continuing tasks due to its online nature.

---

**Algorithm 1** Single-timescale Actor-Critic

1: **Input** initial actor parameter $\boldsymbol{\theta}_0$, initial critic parameter $\boldsymbol{\omega}_0$, initial reward estimator $\eta_0$, stepsize $\alpha_t$ for actor, $\beta_t$ for critic, and $\gamma_t$ for reward estimator.
2: Draw $s_0$ from some initial distribution
3: **for** $t = 0, 1, 2, \cdots, T-1$ **do**
4:     Take action $a_t \sim \pi_{\boldsymbol{\theta}_t}(\cdot|s_t)$
5:     Observe next state $s_{t+1} \sim \mathcal{P}(\cdot|s_t, a_t)$ and reward $r_t = r(s_t, a_t)$
6:     $\delta_t = r_t - \eta_t + \boldsymbol{\phi}(s_{t+1})^\top \boldsymbol{\omega}_t - \boldsymbol{\phi}(s_t)^\top \boldsymbol{\omega}_t$
7:     $\eta_{t+1} = \eta_t + \gamma_t(r_t - \eta_t)$
8:     $\boldsymbol{\omega}_{t+1} = \Pi_{U_{\boldsymbol{\omega}}}(\boldsymbol{\omega}_t + \beta_t \delta_t \boldsymbol{\phi}(s_t))$
9:     $\boldsymbol{\theta}_{t+1} = \boldsymbol{\theta}_t + \alpha_t \delta_t \nabla_{\boldsymbol{\theta}} \log \pi_{\boldsymbol{\theta}_t}(a_t|s_t)$
10: **end for**

---

Note that the "single-timescale" refers to the fact that the stepsizes $\alpha_t, \beta_t, \gamma_t$ are only constantly proportional to each other. In addition, this is a "single-sample" algorithm, since only one sample is needed for the update in each iteration. We remark that Algorithm 1 is more common in practice than double loop variants. In Line 8 of Algorithm 1, a projection ($\Pi_{U_{\boldsymbol{\omega}}}$) is introduced to keep the critic norm-bounded by $U_{\boldsymbol{\omega}}$, which is widely adopted in the literature [31, 37, 36, 8] for analysis. Note that the projection can be handled easily, which is relaxed using its non-expansive property in our analysis.

## 3 Main Results

We first present several standard assumptions that are common in the literature of analyzing AC with linear function approximation [12, 35, 31, 8, 21]. Insights into these conditions and connections with relevant works are also discussed.

### 3.1 Assumptions

By taking the expectation of $\boldsymbol{\omega}_{t+1}$ in (2) with respect to the stationary distribution, we have for any given $\boldsymbol{\omega}_t$

$$\mathbb{E}_{\boldsymbol{\theta}}[\boldsymbol{\omega}_{t+1}|\boldsymbol{\omega}_t] = \boldsymbol{\omega}_t + \beta_t(\boldsymbol{b}_{\boldsymbol{\theta}} + \boldsymbol{A}_{\boldsymbol{\theta}}\boldsymbol{\omega}_t), \tag{3}$$

where

$$\boldsymbol{A}_{\boldsymbol{\theta}} := \mathbb{E}_{(s,a,s')}[\phi(s)(\phi(s') - \phi(s))^\top)], \tag{4}$$

$$\boldsymbol{b}_{\boldsymbol{\theta}} := \mathbb{E}_{(s,a)}[(r(s,a) - J(\boldsymbol{\theta}))\phi(s)], \tag{5}$$

and $s \sim \mu_{\boldsymbol{\theta}}(\cdot), a \sim \pi_{\boldsymbol{\theta}}(\cdot|s), s' \sim \mathcal{P}(\cdot|s,a)$ is the subsequent state of the $(s,a)$. It can be easily shown that [25] the TD limiting point $\boldsymbol{\omega}^*(\boldsymbol{\theta})$ satisfies:

$$\boldsymbol{b}_{\boldsymbol{\theta}} + \boldsymbol{A}_{\boldsymbol{\theta}}\boldsymbol{\omega}^*(\boldsymbol{\theta}) = 0. \tag{6}$$

Note that $\boldsymbol{A}_{\boldsymbol{\theta}}$ reflects the exploration of the policy. To see this, note that without sufficient exploration, $\boldsymbol{A}_{\boldsymbol{\theta}}$ can be rank deficient and (6) can be unsolvable. Consequently, the critic update (2) will not converge. Hence, the following assumption is made to guarantee the problem's solvability.

**Assumption 3.1** (Exploration). For any $\boldsymbol{\theta}$, the matrix $\boldsymbol{A}_{\boldsymbol{\theta}}$ defined in (4) is negative definite and its maximum eigenvalue can be upper bounded by $-\lambda$.

Assumption 3.1 is commonly adopted in analyzing TD learning with linear function approximation [4, 42, 31, 23, 8, 21]. In particular, Assumption 3.1 holds if the policy $\pi_{\boldsymbol{\theta}}$ can explore all state-action pairs in the tabular case [21]. In addition, with this assumption, we can choose $U_{\boldsymbol{\omega}} = \frac{2U_r}{\lambda}$ so that all $\boldsymbol{\omega}^*$ lie within the projection radius $U_{\boldsymbol{\omega}}$ because $\|\boldsymbol{b}_{\boldsymbol{\theta}}\| \leq 2U_r$ and $\|\boldsymbol{A}_{\boldsymbol{\theta}}^{-1}\| \leq \lambda^{-1}$, which justifies the projection operator introduced in Line 8 of Algorithm 1.

**Assumption 3.2** (Uniform ergodicity). For any $\boldsymbol{\theta}$, denote $\mu_{\boldsymbol{\theta}}(\cdot)$ as the stationary distribution induced by the policy $\pi_{\boldsymbol{\theta}}(\cdot|s)$ and the transition probability measure $\mathcal{P}(\cdot|s,a)$. For a Markov chain generated by the policy $\pi_{\boldsymbol{\theta}}$ and transition kernel $\mathcal{P}$, there exists $m > 0$ and $\rho \in (0,1)$ such that

$$d_{TV}(\mathbb{P}(s_\tau \in \cdot|s_0 = s), \mu_{\boldsymbol{\theta}}(\cdot)) \leq m\rho^\tau, \forall \tau \geq 0, \forall s \in \mathcal{S}.$$

Assumption 3.2 assumes the Markov chain is geometrically mixing, which can be implied by the uniform ergodicity. It is commonly employed to characterize the noise induced by Markovian sampling in RL algorithms [4, 42, 31, 8, 21].

**Assumption 3.3** (Lipschitz continuity of policy). Let $\pi_{\boldsymbol{\theta}}(a|s)$ be a policy parameterized by $\boldsymbol{\theta} \in \mathbb{R}^d$. There exists positive constants $B, L_l$ and $L_\pi$ such that for any $\boldsymbol{\theta}, \boldsymbol{\theta}_1, \boldsymbol{\theta}_2 \in \mathbb{R}^d$, $s \in \mathcal{S}$, and $a \in \mathcal{A}$, it holds that:

(a) $\|\nabla \log \pi_{\boldsymbol{\theta}}(a|s)\| \leq B$

(b) $\|\nabla \log \pi_{\boldsymbol{\theta}_1}(a|s) - \nabla \log \pi_{\boldsymbol{\theta}_2}(a|s)\| \leq L_l \|\boldsymbol{\theta}_1 - \boldsymbol{\theta}_2\|$

(c) $|\pi_{\boldsymbol{\theta}_1}(a|s) - \pi_{\boldsymbol{\theta}_2}(a|s)| \leq L_\pi \|\boldsymbol{\theta}_1 - \boldsymbol{\theta}_2\|$

Assumption 3.3 is standard in the literature of policy gradient methods [22, 42, 38, 34, 31, 8, 21]. This assumption holds for many policy classes such as Gaussian policy [10], Boltzmann policy [15], and tabular softmax policy [1].

**Assumption 3.4.** For any $\boldsymbol{\theta}, \boldsymbol{\theta}' \in \mathbb{R}^d$, there exists constant $L_\mu$ such that $\|\nabla \mu_{\boldsymbol{\theta}} - \nabla \mu_{\boldsymbol{\theta}'}\| \leq L_\mu \|\boldsymbol{\theta} - \boldsymbol{\theta}'\|$, where $\mu_{\boldsymbol{\theta}}(s)$ is the stationary distribution under the policy $\pi_{\boldsymbol{\theta}}$.

Assumption 3.4 is introduced in [8] to show the smoothness of the optimal critic $\boldsymbol{\omega}^*(\boldsymbol{\theta})$, which is critical to guarantee the convergence of single-timescale AC. This assumption holds for the finite state-action space setting [18].

### 3.2 Finite-Time Analysis

We define the following uniform upper bound for the linear function approximation error of the critic:

$$\epsilon_{\mathrm{app}} := \sup_{\boldsymbol{\theta}} \sqrt{\mathbb{E}_{s \sim \mu_{\boldsymbol{\theta}}}(\phi(s)^\top \boldsymbol{\omega}^*(\boldsymbol{\theta}) - V_{\boldsymbol{\theta}}(s))^2}.$$

The error $\epsilon_{\text{app}}$ is zero if $V_{\boldsymbol{\theta}}$ is indeed a linear function for any $\boldsymbol{\theta}$. Naturally, it can be expected that the learning errors of Algorithm 1 depend on $\epsilon_{\text{app}}$.

We define the following integer $\tau_T$ that will be useful in the statement of the theorems, which depends on the number of total iterations $T$:

$$\tau_T := \min\{i \geq 0 \mid m\rho^{i-1} \leq \frac{1}{\sqrt{T}}\},$$

where $m, \rho$ are constants defined in Assumption 3.2. Therefore, we choose $\tau_T = \frac{\log m\rho^{-1}}{\log \rho^{-1}} + \frac{\log T}{2\log \rho^{-1}} = \mathcal{O}(\log T)$ such that $m\rho^{\tau_T - 1} \leq \frac{1}{\sqrt{T}}$. The integer $\tau_T$ represents the mixing time of an ergodic Markov chain, which will be used to control the Markovian noise in the analysis.

We quantify the learning errors by defining $y_t := \eta_t - J(\boldsymbol{\theta}_t)$, which is the difference between the reward estimator and the true time-average reward $J(\boldsymbol{\theta}_t)$ at time $t$. For the critic, we define, $\boldsymbol{z}_t := \boldsymbol{\omega}_t - \boldsymbol{\omega}_t^*$ with $\boldsymbol{\omega}_t^* := \boldsymbol{\omega}^*(\boldsymbol{\theta}_t)$ to measure the error between the critic and its target value at iteration $t$. The following two theorems summarize our main results.

**Theorem 3.5** (Markovian sampling). *Consider Algorithm 1 with $\alpha_t = \alpha = \frac{c}{\sqrt{T}}, \beta_t = \beta = \frac{1}{\sqrt{T}}, \gamma_t = \gamma = \frac{1}{\sqrt{T}}$, where $c$ is a constant depending on problem parameters. Suppose Assumptions 3.1-3.4 hold, we have for $T \geq 2\tau_T$,*

$$\frac{1}{T - \tau_T} \sum_{t=\tau_T}^{T-1} \mathbb{E}y_t^2 = \mathcal{O}(\frac{\log^2 T}{\sqrt{T}}) + \mathcal{O}(\epsilon_{\text{app}}),$$

$$\frac{1}{T - \tau_T} \sum_{t=\tau_T}^{T-1} \mathbb{E}\|\boldsymbol{z}_t\|^2 = \mathcal{O}(\frac{\log^2 T}{\sqrt{T}}) + \mathcal{O}(\epsilon_{\text{app}}),$$

$$\frac{1}{T - \tau_T} \sum_{t=\tau_T}^{T-1} \mathbb{E}\|\nabla J(\boldsymbol{\theta}_t)\|^2 = \mathcal{O}(\frac{\log^2 T}{\sqrt{T}}) + \mathcal{O}(\epsilon_{\text{app}}).$$

We defer the interpretation of the above results a bit and present below the analysis results under the i.i.d. sampling first for better comparison. The major difference of i.i.d. from Markovian sampling is that at the $t$-th iteration, the state $s_t$ is sampled from the stationary distribution $\mu_{\boldsymbol{\theta}_t}$ instead of the evolving Markov chain (see Algorithm 2 in Appendix E). The i.i.d. sampling simplifies the analysis in the way that many Markovian noise terms reduce to zero effectively. This leads to a tighter sample complexity bound compared to the Markovian sampling by up to logarithmic factors.

**Theorem 3.6** (i.i.d. sampling). *Consider Algorithm 2 (see Appendix E) with $\alpha_t = \alpha = \frac{c}{\sqrt{T}}, \beta_t = \beta = \frac{1}{\sqrt{T}}, \gamma_t = \gamma = \frac{1}{\sqrt{T}}$, where $c$ is a constant depending on problem parameters. Suppose Assumptions 3.1-3.4 hold, we have for $T \geq 2\tau_T$,*

$$\frac{1}{T - \tau_T} \sum_{t=\tau_T}^{T-1} \mathbb{E}y_t^2 = \mathcal{O}(\frac{1}{\sqrt{T}}) + \mathcal{O}(\epsilon_{\text{app}}),$$

$$\frac{1}{T - \tau_T} \sum_{t=\tau_T}^{T-1} \mathbb{E}\|\boldsymbol{z}_t\|^2 = \mathcal{O}(\frac{1}{\sqrt{T}}) + \mathcal{O}(\epsilon_{\text{app}}),$$

$$\frac{1}{T - \tau_T} \sum_{t=\tau_T}^{T-1} \mathbb{E}\|\nabla J(\boldsymbol{\theta}_t)\|^2 = \mathcal{O}(\frac{1}{\sqrt{T}}) + \mathcal{O}(\epsilon_{\text{app}}).$$

If the critic approximation error $\epsilon_{\text{app}}$ is zero, we see that the reward estimator, the critic, and the actor estimation errors all diminish at a sub-linear rate of $\widetilde{\mathcal{O}}(T^{-\frac{1}{2}})$. The additional logarithmic term hidden by $\widetilde{\mathcal{O}}(\cdot)$ is incurred by the mixing time of the Markov chain, which can be get rid of under the i.i.d. sampling. It also hides the polynomials of all other problem parameters. They are explicitly characterized in the proofs up to the last step of analyzing the overall interconnected error propagation system. One can easily keep and get the dependence orders of all parameters if needed. Here we focus on the dependence of the iteration number for ease of presentation.

To put the results into perspective, note that $\mathcal{O}(T^{-\frac{1}{2}})$ is the rate one would obtain from stochastic gradient descent (SGD) on general non-convex functions with unbiased gradient updates. In terms of sample complexity, to obtain an $\epsilon$-approximate stationary point, it takes a number of $\widetilde{\mathcal{O}}(\epsilon^{-2})$ samples for Markovian sampling (Algorithm 1) and $\mathcal{O}(\epsilon^{-2})$ for i.i.d. sampling (Algorithm 2), which matches the state-of-the-art performance of SGD on non-convex optimization problems.

The obtained sample complexities are superior to those of other AC variants. Notably, [16] provided finite-time convergence for double-loop variant with a $\mathcal{O}(\epsilon^{-4})$ sample complexity and [31] analysed two-timescale variant, yielding a $\widetilde{\mathcal{O}}(\epsilon^{-2.5})$ sample complexity. The sample complexity gap is intrinsic to their inefficient usage of data. In the double-loop setting, the critic starts over to estimate the Q-value for an intermediate policy in the inner loop, ignoring the fact that the consecutive Q-values can be similar given a relatively minor policy update. The two-timescale setting artificially slows down the actor update by adopting an actor stepsize that decays faster than the critic. The single-timescale approach updates the critic and actor parallelly with proportional stepsizes and thus learns more efficiently.

Moreover, our result matches the $\mathcal{O}(\epsilon^{-2})$ sample complexity of policy gradient methods such as REINFORCE [2, 22] under the i.i.d sampling. It is previously found in [31] that there is a sample complexity gap between Algorithm 1 adopting two-timescale stepsizes and (variance-reduced) REINFORCE [22]. In this paper, we close this gap by providing a single-timescale analysis for Algorithm 1 which shows that this practical single-timescale AC can have the same sample complexity as REINFORCE.

### 3.3  Proof Sketch

The main challenge in the finite-time analysis lies in that the estimation errors of the time-average reward, the critic, and the policy gradient are strongly coupled. To overcome this difficulty, we view the propagation of these errors as an interconnected system and analyze them holistically. To better appreciate the advantage of our analysis framework over the decoupled methods that are traditionally adopted in analyzing double-loop and two-timescale variants, we sketch the main proof steps of Theorem 3.5 in the following. We also highlight the key challenges and techniques developed correspondingly. All supporting lemmas mentioned below can be found in Appendix.

We first derive implicit (coupled) upper bounds for the reward estimation error $y_t$, the critic error $z_t$, and the policy gradient $\nabla J(\boldsymbol{\theta}_t)$, respectively. Then, we solve a system of inequalities to establish finite-time convergence.

**Step 1: Reward estimation error analysis.** Using the reward estimator update rule (Line 7 of Algorithm 1), we decompose the reward estimation error into:

$$\begin{aligned} y_{t+1}^2 = {} & (1 - 2\gamma_t)y_t^2 + 2\gamma_t y_t(r_t - J(\boldsymbol{\theta}_t)) \\ & + 2y_t(J(\boldsymbol{\theta}_t) - J(\boldsymbol{\theta}_{t+1})) + (J(\boldsymbol{\theta}_t) - J(\boldsymbol{\theta}_{t+1}) + \gamma_t(r_t - \eta_t))^2. \end{aligned} \tag{7}$$

The second term on the right-hand side of (7) is a bias term caused by the Markovian sample, which is characterized in Lemma C.1. As shown in Lemma E.1, this bias reduces to 0 under i.i.d. sampling after taking the expectation. The third term captures the variation of the moving targets $J(\boldsymbol{\theta}_t)$. The double-loop variant of AC runs a policy evaluation sub-problem in the inner loop for each target $J(\boldsymbol{\theta}_t)$ to estimate the policy gradient accurately. This easily ensures the monotonic decreasing of $J(\boldsymbol{\theta}_t)$ and consequently the convergence. The two-timescale variant utilizes the additional property of $\lim_{t \to \infty} \alpha_t/\beta_t = 0$ to annihilate this term and consequently can have a decoupled analysis. In the case of single-timescale AC, we do not have the aforementioned special algorithm designs and properties to ease the analysis. Instead, we utilize the smoothness of $J(\boldsymbol{\theta})$ (see Lemma B.2) and derive an implicit upper bound for this term as a function of the norm of $y_t$ and $\nabla J(\boldsymbol{\theta}_t)$. This bound will be combined with the implicit bounds derived in Step 2 and Step 3 below to establish the non-asymptotic convergence altogether. The last term in (7) reflects the variance in reward estimation, which is bounded by $\mathcal{O}(\gamma_t)$.

**Step 2: Critic error analysis.** Using the critic update rule (Line 8 of Algorithm 1), we decompose the squared error by (we neglect the projection for the time being for the ease of comprehension. The complete analysis can be found in the appendix.)

$$\begin{aligned} \|\boldsymbol{z}_{t+1}\|^2 = {} & \|\boldsymbol{z}_t\|^2 + 2\beta_t\langle \boldsymbol{z}_t, \bar{g}(\boldsymbol{\omega}_t, \boldsymbol{\theta}_t)\rangle + 2\beta_t\Psi(O_t, \boldsymbol{\omega}_t, \boldsymbol{\theta}_t) + 2\beta_t\langle \boldsymbol{z}_t, \Delta g(O_t, \eta_t, \boldsymbol{\theta}_t)\rangle \\ & + 2\langle \boldsymbol{z}_t, \boldsymbol{\omega}_t^* - \boldsymbol{\omega}_{t+1}^*\rangle + \|\beta_t(g(O_t, \boldsymbol{\omega}_t, \boldsymbol{\theta}_t) + \Delta g(O_t, \eta_t, \boldsymbol{\theta}_t)) + \boldsymbol{\omega}_t^* - \boldsymbol{\omega}_{t+1}^*\|^2, \end{aligned} \tag{8}$$

where $O_t := (s_t, a_t, s_{t+1})$ is a tuple of observations and the definitions of $g, \bar{g}, \Delta g$, and $\Psi$ can be found in (12) and (13) in Appendix A. Without diving into the detailed definitions, here we focus on illustrating the high-level insights of our proof. First of all, the second term on the right-hand side of (8) can be bounded by $-2\lambda\beta_t\|\boldsymbol{z}_t\|^2$ under Assumption 3.1. It provides an explicit characterization of how sufficient exploration can help the convergence of learning. The third term is a Markovian noise, which is further bounded implicitly in Lemma C.3. For the i.i.d sampling case, as shown in Lemma E.1, this bias reduces to 0 after taking the expectation. The fourth term is caused by inaccurate reward and critic estimations, which can be bounded by the norm of $y_t$ and $\boldsymbol{z}_t$. The fifth term tracks both the critic estimation performance $\boldsymbol{z}_t$ and the difference between the drifting critic targets $\boldsymbol{\omega}_t^*$. Similar to the case of Step 1, the double-loop approach bounds this term relying on the accurate policy evaluation sub-problem in the inner loop for each target $\boldsymbol{\omega}_t^*$, whereas the two-timescale approach ensures its convergence by additionally requiring $\lim_{t\to\infty}\alpha_t/\beta_t = 0$. In contrast, we establish an implicit upper bound for this term as a function of $y_t$ and $\boldsymbol{z}_t$ by utilizing the smoothness of the optimal critic proved in Lemma B.4. Finally, the last term reflects the variances of various estimations, which is bounded by $\mathcal{O}(\beta_t)$.

**Step 3: Policy gradient norm analysis.** Using the actor update rule (Line 9 of Algorithm 1) and the smoothness property of $J(\boldsymbol{\theta})$ (see Lemma B.2), we derive

$$
\begin{aligned}
\|\nabla J(\boldsymbol{\theta}_t)\|^2 \leq \frac{1}{\alpha_t}(J(\boldsymbol{\theta}_{t+1}) - J(\boldsymbol{\theta}_t)) + \Theta(O_t, \boldsymbol{\theta}_t) - \langle \nabla J(\boldsymbol{\theta}_t), \Delta h(O_t, \eta_t, \boldsymbol{\omega}_t, \boldsymbol{\theta}_t)\rangle \\
- \langle \nabla J(\boldsymbol{\theta}_t), \mathbb{E}_{O_t'}[\Delta h'(O_t', \boldsymbol{\theta}_t)]\rangle + \frac{L_{J'}}{2}\alpha_t\|\delta_t\nabla\log\pi_{\boldsymbol{\theta}_t}(a_t|s_t)\|^2,
\end{aligned}
\tag{9}
$$

where $O_t'$ is a shorthand for an independent sample from stationary distribution $s \sim \mu_{\boldsymbol{\theta}_t}, a \sim \pi_{\boldsymbol{\theta}_t}, s' \sim \mathcal{P}(\cdot|s,a)$, $\Theta$ is defined in (13), and $L_{J'}$ is a constant. The first term on the right-hand side of (9) compares the actor's performances between consecutive updates, which can be bounded via Abel summation by parts. The second term is a noise term introduced by Markovian sampling, which is characterized in Lemma C.6. Again, as proven in Lemma E.1, this bias reduces to 0 under i.i.d. sampling after taking the expectation. The third term is an error introduced by the inaccurate estimations of both the time-average reward and the critic. This term was directly bounded to zero under both the double-loop setting and the two-timescale setting due to their particular algorithm design, to enable a decoupled analysis. We control this term by providing an implicit bound depending on $y_t$, $\boldsymbol{z}_t$, and $\nabla J(\boldsymbol{\theta}_t)$. The fourth term comes from the linear function approximation error. The last term captures the variance of the stochastic gradient update, which is bounded by $\mathcal{O}(\alpha_t)$.

**Step 4: Interconnected iteration system analysis.** Taking the expectation of and summing (7), (8), and (9) from $\tau_T$ to $T-1$, respectively, we obtain the following system of inequalities in terms of $Y_T$, $Z_T$, $G_T$:

$$
Y_T := \frac{1}{T-\tau_T}\sum_{t=\tau_T}^{T-1}\mathbb{E}y_t^2 \leq \mathcal{O}(\frac{\log^2 T}{\sqrt{T}}) + l_1\sqrt{Y_T G_T},
$$

$$
Z_T := \frac{1}{T-\tau_T}\sum_{t=\tau_T}^{T-1}\mathbb{E}\|\boldsymbol{z}_t\|^2 \leq \mathcal{O}(\frac{\log^2 T}{\sqrt{T}}) + \mathcal{O}(\epsilon_{\text{app}}) + l_2\sqrt{Y_T Z_T} + l_3\sqrt{Z_T(2Y_T + 8Z_T)},
$$

$$
G_T := \frac{1}{T-\tau_T}\sum_{t=\tau_T}^{T-1}\mathbb{E}\|\nabla J(\boldsymbol{\theta}_t)\|^2 \leq \mathcal{O}(\frac{\log^2 T}{\sqrt{T}}) + \mathcal{O}(\epsilon_{\text{app}}) + l_4\sqrt{G_T(2Y_T + 8Z_T)},
$$

where $l_1, l_2, l_3, l_4$ are positive constants. By solving the above system of inequalities, we further prove that if

$$
l_1(1 + 2l_4^2 + 8l_4^2(2l_2^2 + l_3)) \leq 1 \quad \text{and} \quad 16l_3 \leq 1,
$$

then $Y_T, Z_T, G_T$ converge at a rate of $\mathcal{O}(\frac{\log^2 T}{\sqrt{T}})$. This condition can be easily satisfied by choosing the stepsize ratio $c$ to be smaller than a threshold identified in Equation (28). Thus, it completes the proof.

The above proof applies to i.i.d sampling straightforwardly, with the corresponding terms pointed out in the above steps reducing to 0 in the analysis. The additional proof can be found in Lemma E.1.

# 4  Conclusion and Discussion

In this paper, we establish the finite-time analysis for single-timescale AC with Markovian sampling. Our work compares favorably to existing works in terms of analyzing online learning and considering the continuous state space. We developed a series of lemmas that characterize the propagation of errors, and establish their convergence simultaneously by solving a system of nonlinear inequalities. The proposed framework is general and can be applied to analyze other single-timescale stochastic approximation algorithms. Our future work includes further considering the continuous action space problems and developing new proof techniques that require fewer assumptions.

## Acknowledgements

This work was supported by the Singapore Ministry of Education Tier 1 Academic Research Funds (A0009030-00-00, 22-5460-A0001). The authors would like to thank the timely help from Yue Wu and Quanquan Gu for clarifying the proof of their seminar work on finite-time analysis of two-timescale actor-critic.

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

# Table of Contents

## A  Notation

We make use of the following auxiliary Markov chain to deal with the Markovian noise.

**Auxiliary Markov Chain:**

$$s_{t-\tau} \xrightarrow{\boldsymbol{\theta}_{t-\tau}} a_{t-\tau} \xrightarrow{\mathcal{P}} s_{t-\tau+1} \xrightarrow{\boldsymbol{\theta}_{t-\tau}} \widetilde{a}_{t-\tau+1} \xrightarrow{\mathcal{P}} \widetilde{s}_{t-\tau+2} \xrightarrow{\boldsymbol{\theta}_{t-\tau}} \widetilde{a}_{t-\tau+2} \cdots \xrightarrow{\mathcal{P}} \widetilde{s}_t \xrightarrow{\boldsymbol{\theta}_{t-\tau}} \widetilde{a}_t \xrightarrow{\mathcal{P}} \widetilde{s}_{t+1}. \tag{10}$$

For reference, we also show the original Markov chain.

**Original Markov Chain:**

$$s_{t-\tau} \xrightarrow{\boldsymbol{\theta}_{t-\tau}} a_{t-\tau} \xrightarrow{\mathcal{P}} s_{t-\tau+1} \xrightarrow{\boldsymbol{\theta}_{t-\tau+1}} \widetilde{a}_{t-\tau+1} \xrightarrow{\mathcal{P}} \widetilde{s}_{t-\tau+2} \xrightarrow{\boldsymbol{\theta}_{t-\tau+2}} \widetilde{a}_{t-\tau+2} \cdots \xrightarrow{\mathcal{P}} \widetilde{s}_t \xrightarrow{\boldsymbol{\theta}_t} \widetilde{a}_t \xrightarrow{\mathcal{P}} \widetilde{s}_{t+1}. \tag{11}$$

In the sequel, we denote by $\widetilde{O}_t := (\widetilde{s}_t, \widetilde{a}_t, \widetilde{s}_{t+1})$ the tuple generated from the auxiliary Markov chain in (10) while $O_t := (s_t, a_t, s_{t+1})$ denotes the tuple generated from the original Markov chain in (11).

We define the following functions, which will benefit to decompose the errors and simplify the presentation.

$$
\begin{aligned}
\Delta g(O, \eta, \boldsymbol{\theta}) &:= [J(\boldsymbol{\theta}) - \eta]\boldsymbol{\phi}(s), \\
g(O, \boldsymbol{\omega}, \boldsymbol{\theta}) &:= [r(s,a) - J(\boldsymbol{\theta}) + (\boldsymbol{\phi}(s') - \boldsymbol{\phi}(s))^\top \boldsymbol{\omega}]\boldsymbol{\phi}(s), \\
\bar{g}(\boldsymbol{\omega}, \boldsymbol{\theta}) &:= \mathbb{E}_{(s,a,s') \sim (\mu_{\boldsymbol{\theta}}, \pi_{\boldsymbol{\theta}}, \mathcal{P})}[[r(s,a) - J(\boldsymbol{\theta}) + (\boldsymbol{\phi}(s') - \boldsymbol{\phi}(s))^\top \boldsymbol{\omega}]\boldsymbol{\phi}(s)], \\
\Delta h(O, \eta, \boldsymbol{\omega}, \boldsymbol{\theta}) &:= (J(\boldsymbol{\theta}) - \eta + (\boldsymbol{\phi}(s') - \boldsymbol{\phi}(s))^\top (\boldsymbol{\omega} - \boldsymbol{\omega}^*(\boldsymbol{\theta}))\nabla \log \pi_{\boldsymbol{\theta}}(a|s), \\
\Delta h'(O, \boldsymbol{\theta}) &:= ((\boldsymbol{\phi}(s')\boldsymbol{\omega}^*(\boldsymbol{\theta}) - V_{\boldsymbol{\theta}}(s')) - (\boldsymbol{\phi}(s)^\top \boldsymbol{\omega}^*(\boldsymbol{\theta}) - V_{\boldsymbol{\theta}}(s)))\nabla \log \pi_{\boldsymbol{\theta}}(a|s), \\
h(O, \boldsymbol{\theta}) &:= (r(s,a) - J(\boldsymbol{\theta}) + \boldsymbol{\phi}(s')^\top \boldsymbol{\omega}^*(\boldsymbol{\theta}) - \boldsymbol{\phi}(s)^\top \boldsymbol{\omega}^*(\boldsymbol{\theta}))\nabla \log \pi_{\boldsymbol{\theta}}(a|s).
\end{aligned} \tag{12}
$$

We also define the following functions, which characterize the Markovian noise.

$$
\begin{aligned}
\Phi(O, \eta, \boldsymbol{\theta}) &:= (\eta - J(\boldsymbol{\theta}))(r(s,a) - J(\boldsymbol{\theta})), \\
\Psi(O, \boldsymbol{\omega}, \boldsymbol{\theta}) &:= \langle \boldsymbol{\omega} - \boldsymbol{\omega}_{\boldsymbol{\theta}}^*, g(O, \boldsymbol{\omega}, \boldsymbol{\theta}) - \bar{g}(\boldsymbol{\omega}, \boldsymbol{\theta}) \rangle, \\
\Theta(O, O', \boldsymbol{\theta}) &:= \langle \nabla J(\boldsymbol{\theta}), \mathbb{E}_{O'}[h(O', \boldsymbol{\theta})] - h(O, \boldsymbol{\theta}) \rangle, \\
\Xi(O, \boldsymbol{\omega}, \boldsymbol{\theta}) &:= \langle \boldsymbol{\omega} - \boldsymbol{\omega}_{\boldsymbol{\theta}}^*, (\nabla \boldsymbol{\omega}_{\boldsymbol{\theta}}^*)^\top (\mathbb{E}_{O'}[h(O', \boldsymbol{\theta})] - h(O, \boldsymbol{\theta})) \rangle,
\end{aligned} \tag{13}
$$

where $O'$ is a shorthand for an independent sample from stationary distribution $s \sim \mu_{\boldsymbol{\theta}}, a \sim \pi_{\boldsymbol{\theta}}, s' \sim \mathcal{P}$. Define $U_\delta := 2U_r + 2U_{\boldsymbol{\omega}}$ so that we have $|\delta_t| \le U_\delta$, where $\delta_t$ comes from Line 6 in Algorithm 1. Note that from Assumption 3.3, we have $\|\delta \nabla \log \pi_{\boldsymbol{\theta}}\| \le G := U_\delta B$.

## B  Preliminary Lemmas

**Lemma B.1** ([31], Lemma C.4)**.** *For any $\boldsymbol{\theta}_1, \boldsymbol{\theta}_2$, we have*

$$|J(\boldsymbol{\theta}_1) - J(\boldsymbol{\theta}_2)| \leq L_J \|\boldsymbol{\theta}_1 - \boldsymbol{\theta}_2\|,$$

*where $L_J = 2U_r |\mathcal{A}| L_\pi (1 + \lceil \log_\rho m^{-1} \rceil + \frac{1}{1-\rho})$.*

**Lemma B.2** ([38], Lemma 3.2)**.** *For the performance function $J(\boldsymbol{\theta})$, there exists a constant $L_{J'} > 0$ such that for all $\boldsymbol{\theta}_1, \boldsymbol{\theta}_2 \in \mathbb{R}^d$, it holds that*

$$\|\nabla J(\boldsymbol{\theta}_1) - \nabla J(\boldsymbol{\theta}_2)\| \leq L_{J'} \|\boldsymbol{\theta}_1 - \boldsymbol{\theta}_2\|, \tag{14}$$

*which further implies*

$$J(\boldsymbol{\theta}_2) \geq J(\boldsymbol{\theta}_1) + \langle \nabla J(\boldsymbol{\theta}_1), \boldsymbol{\theta}_2 - \boldsymbol{\theta}_1 \rangle - \frac{L_{J'}}{2} \|\boldsymbol{\theta}_1 - \boldsymbol{\theta}_2\|^2, \tag{15}$$

$$J(\boldsymbol{\theta}_2) \leq J(\boldsymbol{\theta}_1) + \langle \nabla J(\boldsymbol{\theta}_1), \boldsymbol{\theta}_2 - \boldsymbol{\theta}_1 \rangle + \frac{L_{J'}}{2} \|\boldsymbol{\theta}_1 - \boldsymbol{\theta}_2\|^2. \tag{16}$$

**Lemma B.3** ([31], Proposition 4.4)**.** *There exists a constant $L_* > 0$ such that*

$$\|\boldsymbol{\omega}^*(\boldsymbol{\theta}_1) - \boldsymbol{\omega}^*(\boldsymbol{\theta}_2)\| \leq L_* \|\boldsymbol{\theta}_1 - \boldsymbol{\theta}_2\|, \forall \boldsymbol{\theta}_1, \boldsymbol{\theta}_2 \in \mathbb{R}^d,$$

*where $L_* = (2\lambda^{-2} U_r + 3\lambda^{-1} U_r) |\mathcal{A}| L_\pi (1 + \lceil \log_\rho m^{-1} \rceil + \frac{1}{1-\rho})$.*

**Lemma B.4** ([8], Proposition 8)**.** *For any $\boldsymbol{\theta}_1, \boldsymbol{\theta}_2 \in \mathbb{R}^d$, we have*

$$\|\nabla \boldsymbol{\omega}^*(\boldsymbol{\theta}_1) - \nabla \boldsymbol{\omega}^*(\boldsymbol{\theta}_2)\| \leq L_s \|\boldsymbol{\theta}_1 - \boldsymbol{\theta}_2\|,$$

*where $L_s$ is a positive constant.*

**Lemma B.5** ([42],[31])**.** *For any $\boldsymbol{\theta}_1$ and $\boldsymbol{\theta}_2$, it holds that*

$$d_{TV}(\mu_{\boldsymbol{\theta}_1}, \mu_{\boldsymbol{\theta}_2}) \leq |\mathcal{A}|(\lceil \log_\rho m^{-1} \rceil + \frac{1}{1-\rho}) \|\boldsymbol{\theta}_1 - \boldsymbol{\theta}_2\|,$$

$$d_{TV}(\mu_{\boldsymbol{\theta}_1} \otimes \pi_{\boldsymbol{\theta}_1}, \mu_{\boldsymbol{\theta}_2} \otimes \pi_{\boldsymbol{\theta}_2}) \leq |\mathcal{A}| L_\pi (1 + \lceil \log_\rho m^{-1} \rceil + \frac{1}{1-\rho}) \|\boldsymbol{\theta}_1 - \boldsymbol{\theta}_2\|,$$

$$d_{TV}(\mu_{\boldsymbol{\theta}_1} \otimes \pi_{\boldsymbol{\theta}_1} \otimes \mathcal{P}, \mu_{\boldsymbol{\theta}_2} \otimes \pi_{\boldsymbol{\theta}_2} \otimes \mathcal{P}) \leq |\mathcal{A}| L_\pi (1 + \lceil \log_\rho m^{-1} \rceil + \frac{1}{1-\rho}) \|\boldsymbol{\theta}_1 - \boldsymbol{\theta}_2\|.$$

**Lemma B.6** ([31], Lemma B.2)**.** *Given time indexes $t$ and $\tau$ such that $t \geq \tau > 0$, consider the auxiliary Markov chain in (10). Conditioning on $s_{t-\tau+1}$ and $\boldsymbol{\theta}_{t-\tau}$, we have*

$$d_{TV}(\mathbb{P}(s_{t+1} \in \cdot), \mathbb{P}(\widetilde{s}_{t+1} \in \cdot)) \leq d_{TV}(\mathbb{P}(O_t \in \cdot), \mathbb{P}(\widetilde{O}_t \in \cdot)),$$

$$d_{TV}(\mathbb{P}(O_t \in \cdot), \mathbb{P}(\widetilde{O}_t \in \cdot)) = d_{TV}(\mathbb{P}((s_t, a_t) \in \cdot), \mathbb{P}((\widetilde{s}_t, \widetilde{a}_t) \in \cdot)),$$

$$d_{TV}(\mathbb{P}((s_t, a_t) \in \cdot), \mathbb{P}((\widetilde{s}_t, \widetilde{a}_t) \in \cdot)) \leq d_{TV}(\mathbb{P}(s_t \in \cdot), \mathbb{P}(\widetilde{s}_t \in \cdot)) + \frac{1}{2} |\mathcal{A}| \mathbb{E}[\|\boldsymbol{\theta}_t - \boldsymbol{\theta}_{t-\tau}\|].$$

## C  Proof of Main Theorem

### C.1  Step 1: Reward estimation error analysis

In this subsection, we will establish an implicit bound for estimator.

**Lemma C.1.** *From any $t \geq \tau > 0$, we have*

$$\mathbb{E}[\Phi(O_t, \eta_t, \boldsymbol{\theta}_t)] \leq 4U_r L_J \|\boldsymbol{\theta}_t - \boldsymbol{\theta}_{t-\tau}\| + 2U_r |\eta_t - \eta_{t-\tau}|$$

$$+ 2U_r^2 |\mathcal{A}| L_\pi \sum_{i=t-\tau}^t \mathbb{E}\|\boldsymbol{\theta}_i - \boldsymbol{\theta}_{t-\tau}\| + 4U_r^2 m \rho^{\tau-1}.$$

**Theorem C.2.** *Choose $\alpha_t = \frac{c}{\sqrt{T}}, \beta_t = \gamma_t = \frac{1}{\sqrt{T}}$, we have*

$$Y_T \leq \mathcal{O}(\frac{\log^2 T}{\sqrt{T}}) + cG\sqrt{Y_T G_T}. \tag{17}$$

*Proof.* From the update rule of reward estimator in Line 7 of Algorithm 1, we have

$$\eta_{t+1} - J(\boldsymbol{\theta}_{t+1}) = \eta_t - J(\boldsymbol{\theta}_t) + J(\boldsymbol{\theta}_t) - J(\boldsymbol{\theta}_{t+1}) + \gamma_t(r_t - \eta_t)$$

Then we have

$$
\begin{aligned}
y_{t+1}^2 &= (y_t + J(\boldsymbol{\theta}_t) - J(\boldsymbol{\theta}_{t+1}) + \gamma_t(r_t - \eta_t))^2 \\
&\leq y_t^2 + 2y_t(J(\boldsymbol{\theta}_t) - J(\boldsymbol{\theta}_{t+1})) + 2\gamma_t y_t(r_t - \eta_t) \\
&\quad + 2(J(\boldsymbol{\theta}_t) - J(\boldsymbol{\theta}_{t+1}))^2 + 2\gamma_t^2(r_t - \eta_t)^2 \\
&= (1 - 2\gamma_t)y_t^2 + 2\gamma_t y_t(r_t - J(\boldsymbol{\theta}_t)) + 2y_t(J(\boldsymbol{\theta}_t) - J(\boldsymbol{\theta}_{t+1})) \\
&\quad + 2(J(\boldsymbol{\theta}_t) - J(\boldsymbol{\theta}_{t+1}))^2 + 2\gamma_t^2(r_t - \eta_t)^2.
\end{aligned}
$$

Taking expectation up to $s_{t+1}$ (the whole trajectory), rearranging and summing from $\tau_T$ to $T - 1$, we have

$$
\sum_{t=\tau_T}^{T-1} \mathbb{E}[y_t^2] \leq \underbrace{\sum_{t=\tau_T}^{T} \frac{1}{2\gamma_t}\mathbb{E}(y_t^2 - y_{t+1}^2)}_{I_1} + \underbrace{\sum_{t=\tau_T}^{T-1} \mathbb{E}[y_t(r_t - J(\boldsymbol{\theta}_t))]}_{I_2} + \underbrace{\sum_{t=\tau_T}^{T-1} \frac{1}{\gamma_t}\mathbb{E}[y_t(J(\boldsymbol{\theta}_t) - J(\boldsymbol{\theta}_{t+1}))]}_{I_3}
$$

$$
+ \underbrace{\sum_{t=\tau_T}^{T-1} \frac{1}{\gamma_t}\mathbb{E}[(J(\boldsymbol{\theta}_t) - J(\boldsymbol{\theta}_{t+1}))^2]}_{I_4} + \underbrace{\sum_{t=\tau_T}^{T-1} \gamma_t \mathbb{E}[(r_t - \eta_t)^2]}_{I_5}.
$$

For term $I_1$, from Abel summation by parts, we have

$$
\begin{aligned}
I_1 &= \sum_{t=\tau_T}^{T-1} \frac{1}{2\gamma_t}(y_t^2 - y_{t+1}^2) \\
&= \sum_{t=\tau_T+1}^{T-1} y_t^2\left(\frac{1}{2\gamma_t} - \frac{1}{2\gamma_{t-1}}\right) + \frac{1}{2\gamma_{\tau_t}}y_{\tau_t}^2 - \frac{1}{\gamma_{T-1}}y_T^2 \\
&\leq \frac{2U_r^2}{\gamma_{T-1}} \\
&= 2U_r^2\sqrt{T}.
\end{aligned}
$$

For term $I_2$, from Lemma C.1, we have

$$
\begin{aligned}
\mathbb{E}[y_t(r_t - J(\boldsymbol{\theta}_t))] &\leq 4U_r L_J\|\boldsymbol{\theta}_t - \boldsymbol{\theta}_{t-\tau}\| + 2U_r|\eta_t - \eta_{t-\tau}| \\
&\quad + 2U_r^2|\mathcal{A}|L_\pi \sum_{i=t-\tau}^{t} \mathbb{E}\|\boldsymbol{\theta}_i - \boldsymbol{\theta}_{t-\tau}\| + 4U_r^2 m\rho^{\tau-1} \\
&\leq 4U_r L_J G\tau\alpha_{t-\tau} + 4U_r^2\tau\gamma_{t-\tau} + 2U_r^2|\mathcal{A}|L_\pi\tau(\tau+1)G\alpha_{t-\tau} + 4U_r^2 m\rho^{\tau-1} \\
&\leq (4U_r L_J G\tau + 2U_r^2|\mathcal{A}|L_\pi G\tau(\tau+1))\alpha_{t-\tau} + 4U_r^2\tau\gamma_{t-\tau} + 4U_r^2 m\rho^{\tau-1}.
\end{aligned}
$$

Choose $\tau = \tau_T$, we have

$$
\begin{aligned}
I_2 &= \sum_{t=\tau_T}^{T-1} \mathbb{E}[y_t(r_t - J(\boldsymbol{\theta}_t))] \\
&\leq (4U_r L_J G\tau_T + 2U_r^2|\mathcal{A}|L_\pi G\tau_T(\tau_T+1)) \sum_{t=\tau_T}^{T-1} \alpha_t \\
&\quad + 4U_r^2\tau_T \sum_{t=\tau_T}^{T--1} \gamma_t + 4U_r^2 \sum_{t=\tau_T}^{T-1} \frac{1}{\sqrt{T}} \\
&= (4U_r L_J G\tau_T + 2U_r^2|\mathcal{A}|L_\pi G\tau_T(\tau_T+1) + 4U_r^2\tau_T + 4U_r^2)\frac{T - \tau_T}{\sqrt{T}}.
\end{aligned}
$$

For $I_3$, if $y_t > 0$, from (15), we have

$$y_t(J(\boldsymbol{\theta}_t) - J(\boldsymbol{\theta}_{t+1})) \leq y_t(\frac{L_{J'}}{2}\|\boldsymbol{\theta}_t - \boldsymbol{\theta}_{t+1}\|^2 + \langle \nabla J(\boldsymbol{\theta}_t), \boldsymbol{\theta}_t - \boldsymbol{\theta}_{t+1}\rangle)$$
$$\leq L_{J'}U_r\|\boldsymbol{\theta}_t - \boldsymbol{\theta}_{t+1}\|^2 + |y_t|\|\boldsymbol{\theta}_t - \boldsymbol{\theta}_{t+1}\|\|\nabla J(\boldsymbol{\theta}_t)\|.$$

If $y_t \leq 0$, from (16), we have

$$y_t(J(\boldsymbol{\theta}_t) - J(\boldsymbol{\theta}_{t+1})) \leq y_t(-\frac{L_{J'}}{2}\|\boldsymbol{\theta}_t - \boldsymbol{\theta}_{t+1}\|^2 + \langle \nabla J(\boldsymbol{\theta}_t), \boldsymbol{\theta}_t - \boldsymbol{\theta}_{t+1}\rangle)$$
$$\leq L_{J'}U_r\|\boldsymbol{\theta}_t - \boldsymbol{\theta}_{t+1}\|^2 + |y_t|\|\boldsymbol{\theta}_t - \boldsymbol{\theta}_{t+1}\|\|\nabla J(\boldsymbol{\theta}_t)\|.$$

Overall, we get

$$I_3 = \sum_{t=\tau_T}^{T-1} \frac{1}{\gamma_t}\mathbb{E}[y_t(J(\boldsymbol{\theta}_t) - J(\boldsymbol{\theta}_{t+1}))]$$
$$\leq \sum_{t=\tau_T}^{T-1} \frac{1}{\gamma_t}\mathbb{E}[L_{J'}U_r\|\boldsymbol{\theta}_t - \boldsymbol{\theta}_{t+1}\|^2 + |y_t|\|\boldsymbol{\theta}_t - \boldsymbol{\theta}_{t+1}\|\|\nabla J(\boldsymbol{\theta}_t)\|]$$
$$\leq \sum_{t=\tau_T}^{T-1} \mathbb{E}[cL_{J'}U_rG^2\alpha_t + cG|y_t|\|\nabla J(\boldsymbol{\theta}_t)\|]$$
$$\leq cL_{J'}U_rG^2\frac{T-\tau_T}{\sqrt{T}} + cG(\sum_{t=\tau_T}^{T-1} \mathbb{E}y_t^2)^{\frac{1}{2}}(\sum_{t=\tau_T}^{T-1} \mathbb{E}\|\nabla J(\boldsymbol{\theta}_t)\|^2)^{\frac{1}{2}}.$$

For term $I_4$, we have

$$I_4 = \sum_{t=\tau_T}^{T-1} \frac{1}{\gamma_t}\mathbb{E}[(J(\boldsymbol{\theta}_t) - J(\boldsymbol{\theta}_{t+1}))^2]$$
$$\leq \sum_{t=\tau_T}^{T-1} \frac{1}{\gamma_t}L_J^2\mathbb{E}\|\boldsymbol{\theta}_t - \boldsymbol{\theta}_{t+1}\|^2$$
$$\leq \sum_{t=\tau_T}^{T-1} \frac{1}{\gamma_t}L_J^2G^2\alpha_t^2$$
$$= L_J^2G^2c^2\frac{T-\tau_T}{\sqrt{T}}.$$

For term $I_5$, we have

$$I_5 = \sum_{t=\tau_T}^{T-1} \gamma_t\mathbb{E}[(r_t - J(\boldsymbol{\theta}_t))^2]$$
$$\leq \sum_{t=\tau_T}^{T-1} 4U_r^2\gamma_t$$
$$= 4U_r^2\frac{T-\tau_T}{\sqrt{T}}.$$

Therefore, we get

$$\sum_{t=\tau_T}^{T-1} \mathbb{E}[y_t^2] \leq (4U_rL_JG\tau_T + 2U_r^2|\mathcal{A}|L_\pi G\tau_T(\tau_T + 1)$$
$$+ 4U_r^2(\tau_T + 2) + c^2G^2(L_{J'}U_r + L_J^2))\frac{T-\tau_T}{\sqrt{T}}$$
$$+ 2U_r^2\sqrt{T} + cG(\sum_{t=\tau_T}^{T-1} \mathbb{E}y_t^2)^{\frac{1}{2}}(\sum_{t=\tau_T}^{T-1} \mathbb{E}\|\nabla J(\boldsymbol{\theta}_t)\|^2)^{\frac{1}{2}}.$$

Since $\tau_T = \mathcal{O}(\log T)$, we have $\frac{\sqrt{T}}{T-\tau_T} \leq \frac{2}{\sqrt{T}}$ for large $T$. Then we get

$$\frac{1}{T-\tau_T}\sum_{t=\tau_T}^{T-1}\mathbb{E}[y_t^2]$$

$$\leq (4U_r L_J G\tau_T + 2U_r^2|\mathcal{A}|L_\pi G\tau_T(\tau_T+1)$$

$$+ 4U_r^2(\tau_T+3) + c^2G^2(L_{J'}U_r + L_J^2))\frac{1}{\sqrt{T}}$$

$$+ cG\big(\frac{1}{T-\tau_T}\sum_{t=\tau_T}^{T-1}\mathbb{E}y_t^2\big)^{\frac{1}{2}}\big(\frac{1}{T-\tau_T}\sum_{t=\tau_T}^{T-1}\mathbb{E}\|\nabla J(\boldsymbol{\theta}_t)\|^2\big)^{\frac{1}{2}}$$

$$= \mathcal{O}\big(\frac{\log^2 T}{\sqrt{T}}\big) + cG\big(\frac{1}{T-\tau_T}\sum_{t=\tau_T}^{T-1}\mathbb{E}y_t^2\big)^{\frac{1}{2}}\big(\frac{1}{T-\tau_T}\sum_{t=\tau_T}^{T-1}\mathbb{E}\|\nabla J(\boldsymbol{\theta}_t)\|^2\big)^{\frac{1}{2}}.$$

Thus we finish the proof. $\qquad\qquad\square$

## C.2 Step 2: Critic error analysis

In this subsection, we will establish an implicit upper bound for critic.

**Lemma C.3.** *For any $t \geq \tau > 0$, we have*

$$\mathbb{E}[\Psi(O_t,\boldsymbol{\omega}_t,\boldsymbol{\theta}_t)] \leq C_1\|\boldsymbol{\theta}_t - \boldsymbol{\theta}_{t-\tau}\| + 6U_\delta\|\boldsymbol{\omega}_t - \boldsymbol{\omega}_{t-\tau}\| + U_\delta^2|\mathcal{A}|L_\pi G\tau(\tau+1)\alpha_{t-\tau} + 2U_\delta^2 m\rho^{\tau-1},$$

*where $C_1 = 2U_\delta^2|\mathcal{A}|L_\pi(1 + \lceil\log_\rho m^{-1}\rceil + \frac{1}{1-\rho}) + 2U_\delta L_J + 2U_\delta L_*$.*

**Lemma C.4.** *Given the definition of $\Xi(O_t,\boldsymbol{\omega}_t,\boldsymbol{\theta}_t)$, for any $t \geq \tau > 0$, we have*

$$\mathbb{E}[\Xi(O_t,\boldsymbol{\omega}_t,\boldsymbol{\theta}_t)] \leq C_2\|\boldsymbol{\theta}_t - \boldsymbol{\theta}_{t-\tau}\| + 2U_\delta B\|\boldsymbol{\omega}_t - \boldsymbol{\omega}_{t-\tau}\|$$

$$+ 2U_\delta^2 B|\mathcal{A}|L_\pi G\tau(\tau+1)\alpha_{t-\tau} + 4U_\delta^2 Bm\rho^{\tau-1}.$$

*where $C_2 := 3BU_\delta^2|\mathcal{A}|L_\pi(1 + \lceil\log_\rho m^{-1}\rceil + \frac{1}{1-\rho}) + 3U_\delta^2 L_l + 8U_\delta BL_*$.*

**Theorem C.5.** *Choose $\alpha_t = \frac{c}{\sqrt{T}}, \beta_t = \gamma_t = \frac{1}{\sqrt{T}}$, we have*

$$Z_T \leq \mathcal{O}\big(\frac{\log^2 T}{\sqrt{T}}\big) + \mathcal{O}(\epsilon_{\text{app}}) + \frac{2}{\lambda}\sqrt{Y_T Z_T} + \frac{2cBL_*}{\lambda}\sqrt{Z_T(2Y_T + 8Z_T)}. \qquad (18)$$

*Proof.* From the update rule of critic in Line 8 of Algorithm 1, we have

$$\|\boldsymbol{\omega}_{t+1} - \boldsymbol{\omega}_{t+1}^*\| = \|\Pi_{U_\omega}(\boldsymbol{\omega}_t + \beta_t\delta_t\boldsymbol{\phi}(s_t)) - \boldsymbol{\omega}_{t+1}^*\|$$

$$= \|\Pi_{U_\omega}(\boldsymbol{\omega}_t + \beta_t\delta_t\boldsymbol{\phi}(s_t)) - \Pi_{U_\omega}(\boldsymbol{\omega}_{t+1}^*)\|$$

$$\leq \|\boldsymbol{\omega}_t + \beta_t\delta_t\boldsymbol{\phi}(s_t) - \boldsymbol{\omega}_{t+1}^*\|$$

$$= \|\boldsymbol{\omega}_t + \beta_t(g(O_t,\boldsymbol{\omega}_t,\boldsymbol{\theta}_t) + \Delta g(O_t,\eta_t,\boldsymbol{\theta}_t)) - \boldsymbol{\omega}_{t+1}^*\|$$

$$= \|\boldsymbol{\omega}_t - \boldsymbol{\omega}_t^* + \beta_t(g(O_t,\boldsymbol{\omega}_t,\boldsymbol{\theta}_t) + \Delta g(O_t,\eta_t,\boldsymbol{\theta}_t)) + \boldsymbol{\omega}_t^* - \boldsymbol{\omega}_{t+1}^*\|.$$

Therefore, we have

$$\|\boldsymbol{z}_{t+1}\|^2 = \|\boldsymbol{z}_t + \beta_t(g(O_t,\boldsymbol{\omega}_t,\boldsymbol{\theta}_t) + \Delta g(O_t,\eta_t,\boldsymbol{\theta}_t)) + \boldsymbol{\omega}_t^* - \boldsymbol{\omega}_{t+1}^*\|^2$$

$$= \|\boldsymbol{z}_t\|^2 + 2\beta_t\langle\boldsymbol{z}_t, g(O_t,\boldsymbol{\omega}_t,\boldsymbol{\theta}_t)\rangle + 2\beta_t\langle\boldsymbol{z}_t, \Delta g(O_t,\eta_t,\boldsymbol{\theta}_t)\rangle$$

$$+ 2\langle\boldsymbol{z}_t, \boldsymbol{\omega}_t^* - \boldsymbol{\omega}_{t+1}^*\rangle + \|\beta_t(g(O_t,\boldsymbol{\omega}_t,\boldsymbol{\theta}_t) + \Delta g(O_t,\eta_t,\boldsymbol{\theta}_t)) + \boldsymbol{\omega}_t^* - \boldsymbol{\omega}_{t+1}^*\|^2$$

$$= \|\boldsymbol{z}_t\|^2 + 2\beta_t\langle\boldsymbol{z}_t, \bar{g}(\boldsymbol{\omega}_t,\boldsymbol{\theta}_t)\rangle + 2\beta_t\Psi(O_t,\boldsymbol{\omega}_t,\boldsymbol{\theta}_t) + 2\beta_t\langle\boldsymbol{z}_t, \Delta g(O_t,\eta_t,\boldsymbol{\theta}_t)\rangle$$

$$+ 2\langle\boldsymbol{z}_t, \boldsymbol{\omega}_t^* - \boldsymbol{\omega}_{t+1}^*\rangle + \|\beta_t(g(O_t,\boldsymbol{\omega}_t,\boldsymbol{\theta}_t) + \Delta g(O_t,\eta_t,\boldsymbol{\theta}_t)) + \boldsymbol{\omega}_t^* - \boldsymbol{\omega}_{t+1}^*\|^2$$

$$\leq \|\boldsymbol{z}_t\|^2 + 2\beta_t\langle\boldsymbol{z}_t, \bar{g}(\boldsymbol{\omega}_t,\boldsymbol{\theta}_t)\rangle + 2\beta_t\Psi(O_t,\boldsymbol{\omega}_t,\boldsymbol{\theta}_t) + 2\beta_t\langle\boldsymbol{z}_t, \Delta g(O_t,\eta_t,\boldsymbol{\theta}_t)\rangle$$

$$+ 2\langle\boldsymbol{z}_t, \boldsymbol{\omega}_t^* - \boldsymbol{\omega}_{t+1}^*\rangle + 2U_\delta^2\beta_t^2 + 2\|\boldsymbol{\omega}_t^* - \boldsymbol{\omega}_{t+1}^*\|^2.$$

Note that we have

$$\langle \boldsymbol{z}_t, \bar{g}(\boldsymbol{\omega}_t, \boldsymbol{\theta}_t) \rangle = \langle \boldsymbol{z}_t, \bar{g}(\boldsymbol{\omega}_t, \boldsymbol{\theta}_t) - \bar{g}(\boldsymbol{\omega}_t^*, \boldsymbol{\theta}_t) \rangle$$
$$= \langle \boldsymbol{z}_t, \mathbb{E}[(\boldsymbol{\phi}(s') - \boldsymbol{\phi}(s))^\top (\boldsymbol{\omega}_t - \boldsymbol{\omega}_t^*) \boldsymbol{\phi}(s)] \rangle$$
$$= \boldsymbol{z}_t^\top \mathbb{E}[\boldsymbol{\phi}(s)(\boldsymbol{\phi}(s') - \boldsymbol{\phi}(s))^\top] \boldsymbol{z}_t$$
$$= \boldsymbol{z}_t^\top A \boldsymbol{z}_t$$
$$\leq -\lambda \|\boldsymbol{z}_t\|^2,$$

where the first equality is due to the fact that $\bar{g}(\boldsymbol{\omega}_t^*, \boldsymbol{\theta}_t) = 0$ and the last inequality follows from Assumption 3.1.

Taking expectation up to $s_{t+1}$, we have

$$\mathbb{E}\|\boldsymbol{z}_{t+1}\|^2 \leq \mathbb{E}\|\boldsymbol{z}_t\|^2 + 2\beta_t \mathbb{E}\langle \boldsymbol{z}_t, \bar{g}(\boldsymbol{\omega}_t, \boldsymbol{\theta}_t) \rangle + 2\beta_t \mathbb{E}\Psi(O_t, \boldsymbol{\omega}_t, \boldsymbol{\theta}_t) + 2\beta_t \mathbb{E}\langle \boldsymbol{z}_t, \Delta g(O_t, \eta_t, \boldsymbol{\theta}_t) \rangle$$
$$+ 2\mathbb{E}\langle \boldsymbol{z}_t, \boldsymbol{\omega}_t^* - \boldsymbol{\omega}_{t+1}^* \rangle + 2U_\delta^2 \beta_t^2 + 2\mathbb{E}\|\boldsymbol{\omega}_t^* - \boldsymbol{\omega}_{t+1}^*\|^2$$
$$\leq (1 - 2\lambda\beta_t)\mathbb{E}\|\boldsymbol{z}_t\|^2 + 2\beta_t \mathbb{E}\Psi(O_t, \boldsymbol{\omega}_t, \boldsymbol{\theta}_t) + 2\beta_t \mathbb{E}\langle \boldsymbol{z}_t, \Delta g(O_t, \eta_t, \boldsymbol{\theta}_t) \rangle$$
$$+ 2\mathbb{E}\langle \boldsymbol{z}_t, \boldsymbol{\omega}_t^* - \boldsymbol{\omega}_{t+1}^* \rangle + 2U_\delta^2 \beta_t^2 + 2\mathbb{E}\|\boldsymbol{\omega}_t^* - \boldsymbol{\omega}_{t+1}^*\|^2$$
$$\leq (1 - 2\lambda\beta_t)\mathbb{E}\|\boldsymbol{z}_t\|^2 + 2\beta_t \mathbb{E}\Psi(O_t, \boldsymbol{\omega}_t, \boldsymbol{\theta}_t) + 2\beta_t \mathbb{E}\langle \boldsymbol{z}_t, \Delta g(O_t, \eta_t, \boldsymbol{\theta}_t) \rangle$$
$$+ 2\mathbb{E}\langle \boldsymbol{z}_t, \boldsymbol{\omega}_t^* - \boldsymbol{\omega}_{t+1}^* + (\nabla\boldsymbol{\omega}_t^*)^\top (\boldsymbol{\theta}_{t+1} - \boldsymbol{\theta}_t) \rangle$$
$$+ 2\mathbb{E}\langle \boldsymbol{z}_t, (\nabla\boldsymbol{\omega}_t^*)^\top (\boldsymbol{\theta}_t - \boldsymbol{\theta}_{t+1}) \rangle + 2U_\delta^2 \beta_t^2 + 2\mathbb{E}\|\boldsymbol{\omega}_t^* - \boldsymbol{\omega}_{t+1}^*\|^2$$
$$\overset{(1)}{\leq} (1 - 2\lambda\beta_t)\mathbb{E}\|\boldsymbol{z}_t\|^2 + 2\beta_t \mathbb{E}\Psi(O_t, \boldsymbol{\omega}_t, \boldsymbol{\theta}_t) + 2\beta_t \mathbb{E}\|\boldsymbol{z}_t\| |y_t| + L_s \mathbb{E}\|\boldsymbol{z}_t\| \|\boldsymbol{\theta}_{t+1} - \boldsymbol{\theta}_t\|^2$$
$$+ 2\alpha_t \mathbb{E}\langle \boldsymbol{z}_t, -(\nabla\boldsymbol{\omega}_t^*)^\top \delta_t \nabla\log\pi_{\boldsymbol{\theta}_t}(a_t|s_t) \rangle + 2U_\delta^2 \beta_t^2 + 2L_*^2 \mathbb{E}\|\boldsymbol{\theta}_t - \boldsymbol{\theta}_{t+1}\|^2$$
$$\leq (1 - 2\lambda\beta_t)\mathbb{E}\|\boldsymbol{z}_t\|^2 + 2\beta_t \mathbb{E}\Psi(O_t, \boldsymbol{\omega}_t, \boldsymbol{\theta}_t) + 2\beta_t \sqrt{\mathbb{E}y_t^2}\sqrt{\mathbb{E}\|\boldsymbol{z}_t\|^2}$$
$$+ \frac{L_s}{2}\mathbb{E}\|\boldsymbol{z}_t\|^2 \|\boldsymbol{\theta}_{t+1} - \boldsymbol{\theta}_t\|^2 + \frac{L_s}{2}\mathbb{E}\|\boldsymbol{\theta}_{t+1} - \boldsymbol{\theta}_t\|^2 + 2U_\delta^2 \beta_t^2 + 2L_*^2 G^2 \alpha_t^2$$
$$+ 2\alpha_t \mathbb{E}\langle \boldsymbol{z}_t, -(\nabla\boldsymbol{\omega}_t^*)^\top \delta_t \nabla\log\pi_{\boldsymbol{\theta}_t}(a_t|s_t) \rangle$$
$$\leq (1 - 2\lambda\beta_t)\mathbb{E}\|\boldsymbol{z}_t\|^2 + 2\beta_t \mathbb{E}\Psi(O_t, \boldsymbol{\omega}_t, \boldsymbol{\theta}_t) + 2\beta_t \sqrt{\mathbb{E}y_t^2}\sqrt{\mathbb{E}\|\boldsymbol{z}_t\|^2} + \frac{L_s G^2}{2}\alpha_t^2 \mathbb{E}\|\boldsymbol{z}_t\|^2$$
$$+ 2U_\delta^2 \beta_t^2 + (2L_*^2 + \frac{L_s}{2})G^2\alpha_t^2 + 2\alpha_t \mathbb{E}\langle \boldsymbol{z}_t, -(\nabla\boldsymbol{\omega}_t^*)^\top \delta_t \nabla\log\pi_{\boldsymbol{\theta}_t}(a_t|s_t) \rangle$$
$$\overset{(2)}{\leq} (1 - \lambda\beta_t)\mathbb{E}\|\boldsymbol{z}_t\|^2 + 2\beta_t \mathbb{E}\Psi(O_t, \boldsymbol{\omega}_t, \boldsymbol{\theta}_t) + 2\beta_t \sqrt{\mathbb{E}y_t^2}\sqrt{\mathbb{E}\|\boldsymbol{z}_t\|^2}$$
$$+ 2U_\delta^2 \beta_t^2 + (2L_*^2 + \frac{L_s}{2})G^2\alpha_t^2 + 2\alpha_t \mathbb{E}\langle \boldsymbol{z}_t, -(\nabla\boldsymbol{\omega}_t^*)^\top \delta_t \nabla\log\pi_{\boldsymbol{\theta}_t}(a_t|s_t) \rangle \qquad (19)$$

where (1) follows from the $L_s$-smoothness of $\boldsymbol{\omega}^*$ in Lemma B.4; (2) uses $\frac{L_s G^2}{2}\alpha_t^2 \leq \lambda\beta_t$ for large $T$.

For term $\mathbb{E}\langle \boldsymbol{z}_t, -(\nabla\boldsymbol{\omega}_t^*)^\top \delta_t \nabla\log\pi_{\boldsymbol{\theta}_t}(a_t|s_t) \rangle$, we have

$$\mathbb{E}\langle \boldsymbol{z}_t, -(\nabla\boldsymbol{\omega}_t^*)^\top \delta_t \nabla\log\pi_{\boldsymbol{\theta}_t}(a_t|s_t) \rangle$$
$$= \mathbb{E}\langle \boldsymbol{z}_t, (\nabla\boldsymbol{\omega}_t^*)^\top (-\Delta h(O_t, \eta_t, \boldsymbol{\omega}_t, \boldsymbol{\theta}_t) - h(O_t, \boldsymbol{\theta}_t)) \rangle$$
$$= -\mathbb{E}\langle \boldsymbol{z}_t, (\nabla\boldsymbol{\omega}_t^*)^\top \Delta h(O_t, \eta_t, \boldsymbol{\omega}_t, \boldsymbol{\theta}_t) \rangle$$
$$+ \mathbb{E}\langle \boldsymbol{z}_t, (\nabla\boldsymbol{\omega}_t^*)^\top (\mathbb{E}_{O_t'}[h(O_t', \boldsymbol{\theta}_t)] - h(O_t, \boldsymbol{\theta}_t) - \mathbb{E}_{O_t'}[h(O_t', \boldsymbol{\theta}_t)]) \rangle$$
$$= \mathbb{E}[\Xi(O_t, \boldsymbol{\omega}_t, \boldsymbol{\theta}_t)] - \mathbb{E}\langle \boldsymbol{z}_t, (\nabla\boldsymbol{\omega}_t^*)^\top \Delta h(O_t, \eta_t, \boldsymbol{\omega}_t, \boldsymbol{\theta}_t) \rangle - \mathbb{E}\langle \boldsymbol{z}_t, (\nabla\boldsymbol{\omega}_t^*)^\top \mathbb{E}_{O_t'}[h(O_t', \boldsymbol{\theta}_t)] \rangle \quad (20)$$

Note that from Cauchy-Schwartz inequality and $L_*$ is the Lipschitz constant of $\boldsymbol{\omega}^*$ in Lemma B.3, we have

$$-\mathbb{E}\langle \boldsymbol{z}_t, (\nabla\boldsymbol{\omega}_t^*)^\top \Delta h(O_t, \eta_t, \boldsymbol{\omega}_t, \boldsymbol{\theta}_t) \rangle \leq BL_*\sqrt{\mathbb{E}\|\boldsymbol{z}_t\|^2}\sqrt{2\mathbb{E}y_t^2 + 8\mathbb{E}\|\boldsymbol{z}_t\|^2}. \qquad (21)$$

Furthermore, it holds that

$$\begin{aligned}
\mathbb{E}_{O'}\|\nabla h'(O,\boldsymbol{\theta})\|^2 &= \mathbb{E}_{O'}\|((\boldsymbol{\phi}(s')^\top\boldsymbol{\omega}^* - V_{\boldsymbol{\theta}}(s')) - (\boldsymbol{\phi}(s)^\top\boldsymbol{\omega}^* - V_{\boldsymbol{\theta}}(s)))\nabla\log\pi_{\boldsymbol{\theta}}(a|s)\|^2 \\
&\leq \mathbb{E}_{O'}[B^2((\boldsymbol{\phi}(s')^\top\boldsymbol{\omega}^* - V_{\boldsymbol{\theta}}(s')) - (\boldsymbol{\phi}(s)^\top\boldsymbol{\omega}^* - V_{\boldsymbol{\theta}}(s)))^2] \\
&\leq \mathbb{E}_{O'}[2B^2(\boldsymbol{\phi}(s')^\top\boldsymbol{\omega}^* - V_{\boldsymbol{\theta}}(s'))^2 + (\boldsymbol{\phi}(s)^\top\boldsymbol{\omega}^* - V_{\boldsymbol{\theta}}(s))^2] \\
&= 4B^2\mathbb{E}_{O'}[(\boldsymbol{\phi}(s)^\top\boldsymbol{\omega}^* - V_{\boldsymbol{\theta}}(s))^2] \\
&= 4B^2\epsilon_{\text{app}}^2.
\end{aligned}$$

Therefore, we have

$$\begin{aligned}
-\langle\boldsymbol{z}_t, (\nabla\boldsymbol{\omega}_t^*)^\top\mathbb{E}_{O'}[\Delta h'(O',\boldsymbol{\theta}_t)]\rangle &\leq U_\delta L_*\sqrt{\|\mathbb{E}_{O'}[\Delta h'(O_t,\boldsymbol{\theta}_t)]\|^2} \\
&\leq U_\delta L_*\sqrt{\mathbb{E}_{O'}\|\Delta h'(O_t,\boldsymbol{\theta}_t)\|^2} \\
&\leq 2BU_\delta L_*\epsilon_{\text{app}}. \tag{22}
\end{aligned}$$

Substituting (21) and (22) into (20) yields

$$\begin{aligned}
\mathbb{E}\langle\boldsymbol{z}_t, -(\nabla\boldsymbol{\omega}_t^*)^\top\delta_t\nabla\log\pi_{\boldsymbol{\theta}_t}(a_t|s_t)\rangle &\leq \mathbb{E}\Xi(O_t,\boldsymbol{\omega}_t,\boldsymbol{\theta}_t) + 2BU_\delta L_*\epsilon_{\text{app}} \\
&\quad + BL_*\sqrt{\mathbb{E}\|\boldsymbol{z}_t\|^2}\sqrt{2\mathbb{E}y_t^2 + 8\mathbb{E}\|\boldsymbol{z}_t\|^2}. \tag{23}
\end{aligned}$$

Plugging (23) into (19), we have

$$\begin{aligned}
\mathbb{E}\|\boldsymbol{z}_{t+1}\|^2 &\leq (1-\lambda\beta_t)\mathbb{E}\|\boldsymbol{z}_t\|^2 + 2\beta_t\mathbb{E}\Psi(O_t,\boldsymbol{\omega}_t,\boldsymbol{\theta}_t) + 2\alpha_t\mathbb{E}\Xi(O_t,\boldsymbol{\omega}_t,\boldsymbol{\theta}_t) \\
&\quad + 2\beta_t\sqrt{\mathbb{E}y_t^2}\sqrt{\mathbb{E}\|\boldsymbol{z}_t\|^2} + 2BL_*\alpha_t\sqrt{\mathbb{E}\|\boldsymbol{z}_t\|^2}\sqrt{2\mathbb{E}y_t^2 + 8\mathbb{E}\|\boldsymbol{z}_t\|^2} \\
&\quad + 2U_\delta^2\beta_t^2 + (2L_*^2 + \frac{L_s}{2})G^2\alpha_t^2 + 4BU_\delta\alpha_t\epsilon_{\text{app}}. \tag{24}
\end{aligned}$$

Rearranging and summing from $\tau_T$ to $T$ gives

$$\begin{aligned}
\lambda\sum_{\tau_T}^{T-1}\mathbb{E}\|\boldsymbol{z}_t\|^2 &\leq \underbrace{\sum_{t=\tau_T}^{T-1}\frac{1}{\beta_t}(\mathbb{E}\|\boldsymbol{z}_t\|^2 - \mathbb{E}\|\boldsymbol{z}_{t+1}\|^2)}_{I_1} + 2\underbrace{\sum_{t=\tau_T}^{T-1}\mathbb{E}\Psi(O_t,\boldsymbol{\omega}_t,\boldsymbol{\theta}_t)}_{I_2} + 2c\underbrace{\sum_{t=\tau_T}^{T-1}\mathbb{E}\Xi(O_t,\boldsymbol{\omega}_t,\boldsymbol{\theta}_t)}_{I_3} \\
&\quad + 2\underbrace{\sum_{t=\tau_T}^{T-1}\sqrt{\mathbb{E}y_t^2}\sqrt{\mathbb{E}\|\boldsymbol{z}_t\|^2}}_{I_4} + 2cBL_*\underbrace{\sum_{t=\tau_T}^{T-1}\sqrt{\mathbb{E}\|\boldsymbol{z}_t\|^2}\sqrt{2\mathbb{E}y_t^2 + 8\mathbb{E}\|\boldsymbol{z}_t\|^2}}_{I_5} \\
&\quad + \sum_{t=\tau_T}^{T-1}(2U_\delta^2\beta_t + c(2L_*^2 + \frac{L_s}{2})G^2\alpha_t + 4cBU_\delta\epsilon_{\text{app}}).
\end{aligned}$$

In the sequel, we will tackle $I_1, I_2, I_3, I_4, I_5$ respectively.

For term $I_1$, from Abel summation by parts, we have

$$\begin{aligned}
I_1 &= \sum_{t=\tau_T}^{T-1}\frac{1}{\beta_t}(\mathbb{E}\|\boldsymbol{z}_t\|^2 - \mathbb{E}\|\boldsymbol{z}_{t+1}\|^2) \\
&= \sum_{t=\tau_T+1}^{T-1}(\frac{1}{\beta_t} - \frac{1}{\beta_{t-1}})\mathbb{E}\|\boldsymbol{z}_t\|^2 + \frac{1}{\beta_{\tau_T}}\mathbb{E}\|\boldsymbol{z}_{\tau_T}\|^2 - \frac{1}{\beta_{T-1}}\mathbb{E}\|\boldsymbol{z}_T\|^2 \\
&\leq U_\delta^2(\sum_{t=\tau_T+1}^{T-1}(\frac{1}{\beta_t} - \frac{1}{\beta_{t-1}}) + \frac{1}{\beta_{\tau_T}}) \\
&= U_\delta^2\sqrt{T},
\end{aligned}$$

where the inequality is due to $\mathbb{E}\|\boldsymbol{z}_t\|^2 \leq U_\delta^2$ and discard the last term.

For term $I_2$, from Lemma C.3, choose $\tau = \tau_T$, we have

$$\mathbb{E}\Psi(O_t, \boldsymbol{\omega}_t, \boldsymbol{\theta}_t) \leq C_1 \|\boldsymbol{\theta}_t - \boldsymbol{\theta}_{t-\tau_T}\| + U_\delta^2 |\mathcal{A}| L_\pi G \tau_T (\tau_T + 1) \alpha_{t-\tau_T}$$
$$+ 2U_\delta^2 m \rho^{\tau_T - 1} + 6U_\delta \|\boldsymbol{\omega}_t - \boldsymbol{\omega}_{t-\tau_T}\|$$
$$\leq C_1 \sum_{k=t-\tau_T}^{t-1} G\alpha_k + U_\delta^2 |\mathcal{A}| L_\pi G \tau_T (\tau_T + 1) \alpha_{t-\tau_T} + \frac{2U_\delta^2}{\sqrt{T}} + 6U_\delta \sum_{k=t-\tau_T}^{t-1} U_\delta \beta_k$$
$$\leq (C_1 G \tau_T + U_\delta^2 |\mathcal{A}| L_\pi G \tau_T (\tau_T + 1)) \alpha_{t-\tau_T} + \frac{2U_\delta^2}{\sqrt{T}} + 6U_\delta^2 \tau_T \beta_{t-\tau_T}.$$

Then we get

$$I_2 = 2 \sum_{T=\tau_T}^{T-1} \mathbb{E}\Psi(O_t, \boldsymbol{\omega}_t, \boldsymbol{\theta}_t)$$
$$\leq 2 \sum_{T=\tau_T}^{T-1} ((C_1 G \tau_T + U_\delta^2 |\mathcal{A}| L_\pi G \tau_T (\tau_T + 1)) \alpha_t + \frac{2U_\delta^2}{\sqrt{T}} + 6U_\delta^2 \tau_T \beta_t).$$

For term $I_3$, from Lemma C.4, choose $\tau = \tau_T$, we have

$$\mathbb{E}[\Xi(O_t, \boldsymbol{\omega}_t, \boldsymbol{\theta}_t)] \leq C_2 \|\boldsymbol{\theta}_t - \boldsymbol{\theta}_{t-\tau}\| + 2U_\delta B \|\boldsymbol{\omega}_t - \boldsymbol{\omega}_{t-\tau}\|$$
$$+ 2U_\delta^2 B |\mathcal{A}| L_\pi G \tau (\tau + 1) \alpha_{t-\tau} + 4U_\delta^2 B m \rho^{\tau-1}$$
$$\leq C_2 \sum_{k=t-\tau_T}^{t-1} G\alpha_k + 2U_\delta B \sum_{k=t-\tau_T}^{t-1} U_\delta \beta_k$$
$$+ 2U_\delta^2 B |\mathcal{A}| L_\pi G \tau (\tau + 1) \alpha_{t-\tau} + 4U_\delta^2 B m \rho^{\tau_T - 1}$$
$$\leq (C_2 G \tau_T + 2U_\delta^2 B |\mathcal{A}| L_\pi G \tau_T (\tau_T + 1)) \alpha_t + 2U_\delta^2 B \tau_T \beta_t + \frac{4U_\delta^2 B}{\sqrt{T}}.$$

Therefore, we have

$$I_3 = 2c \sum_{t=\tau_T}^{T-1} \mathbb{E}\Xi(O_t, \boldsymbol{\omega}_t, \boldsymbol{\theta}_t)$$
$$\leq 2c \sum_{t=\tau_T}^{T-1} ((C_2 G \tau_T + 2U_\delta^2 B |\mathcal{A}| L_\pi G \tau_T (\tau_T + 1)) \alpha_t + 2U_\delta^2 B \tau_T \beta_t + \frac{4U_\delta^2 B}{\sqrt{T}}).$$

For term $I_4$ and $I_5$, from Cauchy-Schwartz inequality, we have

$$I_4 \leq 2 (\sum_{t=\tau_T}^{T-1} \mathbb{E}y_t^2)^{\frac{1}{2}} (\sum_{t=\tau_T}^{T-1} \mathbb{E}\|\boldsymbol{z}_t\|^2)^{\frac{1}{2}},$$

$$I_5 \leq 2cBL_* (\sum_{t=\tau_T}^{T-1} \mathbb{E}\|\boldsymbol{z}_t\|^2)^{\frac{1}{2}} (2 \sum_{t=\tau_T}^{T-1} \mathbb{E}y_t^2 + 8 \sum_{t=\tau_T}^{T-1} \mathbb{E}\|\boldsymbol{z}_t\|^2)^{\frac{1}{2}}.$$

Overall, we get

$$\lambda \sum_{t=\tau_T}^{T-1} \mathbb{E}\|\boldsymbol{z}_t\|^2 \le 2(\sum_{t=\tau_T}^{T-1} \mathbb{E}y_t^2)^{\frac{1}{2}}(\sum_{t=\tau_T}^{T-1} \mathbb{E}\|\boldsymbol{z}_t\|^2)^{\frac{1}{2}}$$

$$+ 2cBL_*(\sum_{t=\tau_t}^{T-1} \mathbb{E}\|\boldsymbol{z}_t\|^2)^{\frac{1}{2}}(2\sum_{t=\tau_T}^{T-1} \mathbb{E}y_t^2 + 8\sum_{t=\tau_T}^{T-1} \mathbb{E}\|\boldsymbol{z}_t\|^2)^{\frac{1}{2}}$$

$$+ U_\delta^2\sqrt{T} + 2\sum_{T=\tau_T}^{T-1}((C_1G\tau_T + U_\delta^2|\mathcal{A}|L_\pi G\tau_T(\tau_T+1))\alpha_t + \frac{2U_\delta^2}{\sqrt{T}} + 6U_\delta^2\tau_T\beta_t)$$

$$+ 2c\sum_{t=\tau_T}^{T-1}((C_2G\tau_T + 2U_\delta^2B|\mathcal{A}|L_\pi G\tau_T(\tau_T+1))\alpha_t + 2U_\delta^2B\tau_T\beta_t + \frac{4U_\delta^2B}{\sqrt{T}})$$

$$+ \sum_{t=\tau_T}^{T-1}(2U_\delta^2\beta_t + c(2L_*^2 + \frac{L_s}{2})G^2\alpha_t + 4cBU_\delta\epsilon_{\text{app}}).$$

Therefore, we have

$$Z_T \overset{(1)}{\le} \frac{2}{\lambda}(\frac{1}{T-\tau_T}\sum_{t=\tau_T}^{T-1} \mathbb{E}y_t^2)^{\frac{1}{2}}(\frac{1}{T-\tau_T}\sum_{t=\tau_T}^{T-1} \mathbb{E}\|\boldsymbol{z}_t\|^2)^{\frac{1}{2}}$$

$$+ \frac{2cBL_*}{\lambda}(\frac{1}{T-\tau_T}\sum_{t=\tau_t}^{T-1} \mathbb{E}\|\boldsymbol{z}_t\|^2)^{\frac{1}{2}}(2\frac{1}{T-\tau_T}\sum_{t=\tau_T}^{T-1} \mathbb{E}y_t^2 + 8\frac{1}{T-\tau_T}\sum_{t=\tau_T}^{T-1} \mathbb{E}\|\boldsymbol{z}_t\|^2)^{\frac{1}{2}}$$

$$+ \frac{1}{\lambda}(\frac{2U_\delta^2}{\sqrt{T}} + 2(C_1G\tau_T + U_\delta^2|\mathcal{A}|L_\pi G\tau_T(\tau_T+1))\alpha_t + \frac{4U_\delta^2}{\sqrt{T}} + 12U_\delta^2\tau_T\beta_t$$

$$+ 2c(C_2G\tau_T + 2U_\delta^2B|\mathcal{A}|L_\pi G\tau_T(\tau_T+1))\alpha_t + 4cU_\delta^2B\tau_T\beta_t + \frac{8cU_\delta^2B}{\sqrt{T}}$$

$$+ 2U_\delta^2\beta_t + c(2L_*^2 + \frac{L_s}{2})G^2\alpha_t + 4cBU_\delta\epsilon_{\text{app}})$$

$$= \mathcal{O}(\frac{\log^2 T}{\sqrt{T}}) + \mathcal{O}(\epsilon_{\text{app}}) + \frac{2}{\lambda}(\frac{1}{T-\tau_T}\sum_{t=\tau_T}^{T-1} \mathbb{E}y_t^2)^{\frac{1}{2}}(\frac{1}{T-\tau_T}\sum_{t=\tau_T}^{T-1} \mathbb{E}\|\boldsymbol{z}_t\|^2)^{\frac{1}{2}}$$

$$+ \frac{2cBL_*}{\lambda}(\frac{1}{T-\tau_T}\sum_{t=\tau_t}^{T-1} \mathbb{E}\|\boldsymbol{z}_t\|^2)^{\frac{1}{2}}(2\frac{1}{T-\tau_T}\sum_{t=\tau_T}^{T-1} \mathbb{E}y_t^2 + 8\frac{1}{T-\tau_T}\sum_{t=\tau_T}^{T-1} \mathbb{E}\|\boldsymbol{z}_t\|^2)^{\frac{1}{2}},$$

where (1) follows from $\tau_T = \mathcal{O}(\log T)$ so that $T - \tau_T \ge \frac{1}{2}T$ for large $T$. Therefore, we have

$$Z_T \le \mathcal{O}(\frac{\log^2 T}{\sqrt{T}}) + \mathcal{O}(\epsilon_{\text{app}}) + \frac{2}{\lambda}\sqrt{Y_TZ_T} + \frac{2cBL_*}{\lambda}\sqrt{Z_T(2Y_T + 8Z_T)},$$

which completes the proof. $\qquad\square$

## C.3   Step 3: Policy gradient norm analysis

In this subsection, we will establish an implicit upper bound for policy gradient norm.

**Lemma C.6.** *For any $t \ge \tau > 0$, it holds that*

$$\mathbb{E}[\Theta(O_t, \boldsymbol{\theta}_t)] \le D_1\tau(\tau+1)G\alpha + D_2m\rho^{\tau-1},$$

*where $D_1 = \max\{U_\delta BL_{J'} + 3L_JL_h, 2U_\delta BL_J|\mathcal{A}|L_\pi\}$ and $D_2 = 4U_\delta BL_J$.*

**Theorem C.7.** *We have*

$$G_T \le \mathcal{O}(\frac{\log^2 T}{\sqrt{T}}) + \mathcal{O}(\epsilon_{\text{app}}) + B\sqrt{G_T(2Y_T + 8Z_T)}. \tag{25}$$

*Proof.* From the update rule of actor in Line 9 of Algorithm 1 and 15, we have

$$J(\boldsymbol{\theta}_{t+1}) \geq J(\boldsymbol{\theta}_t) + \langle \nabla J(\boldsymbol{\theta}_t), \boldsymbol{\theta}_{t+1} - \boldsymbol{\theta}_t \rangle - \frac{L_{J'}}{2}\|\boldsymbol{\theta}_1 - \boldsymbol{\theta}_2\|^2$$

$$= J(\boldsymbol{\theta}_t) + \langle \nabla J(\boldsymbol{\theta}_t), \delta_t \nabla \log \pi_{\boldsymbol{\theta}_t}(a_t|s_t) \rangle - \frac{L_{J'}}{2}\alpha_t^2\|\delta_t \nabla \log \pi_{\boldsymbol{\theta}_t}(a_t|s_t)\|^2$$

$$= J(\boldsymbol{\theta}_t) + \alpha_t\langle \nabla J(\boldsymbol{\theta}_t), \Delta h(O_t, \eta_t, \boldsymbol{\omega}_t, \boldsymbol{\theta}_t) \rangle + \alpha_t \langle \nabla J(\boldsymbol{\theta}_t), h(O_t, \boldsymbol{\theta}_t) \rangle$$
$$- \frac{L_{J'}}{2}\alpha_t^2\|\delta_t \nabla \log \pi_{\boldsymbol{\theta}_t}(a_t|s_t)\|^2$$

$$= J(\boldsymbol{\theta}_t) + \alpha_t\langle \nabla J(\boldsymbol{\theta}_t), \Delta h(O_t, \eta_t, \boldsymbol{\omega}_t, \boldsymbol{\theta}_t) \rangle - \alpha_t\Theta(O_t, \boldsymbol{\theta}_t)$$
$$+ \alpha_t\langle \nabla J(\boldsymbol{\theta}_t), \mathbb{E}_{O'}[h(O', \boldsymbol{\theta}_t)] \rangle - \frac{L_{J'}}{2}\alpha_t^2\|\delta_t \nabla \log \pi_{\boldsymbol{\theta}_t}(a_t|s_t)\|^2$$

$$= J(\boldsymbol{\theta}_t) + \alpha_t\langle \nabla J(\boldsymbol{\theta}_t), \Delta h(O_t, \eta_t, \boldsymbol{\omega}_t, \boldsymbol{\theta}_t) \rangle - \alpha_t\Theta(O_t, \boldsymbol{\theta}_t) + \alpha_t\|\nabla J(\boldsymbol{\theta}_t)\|^2$$
$$+ \alpha_t\langle \nabla J(\boldsymbol{\theta}_t), \mathbb{E}_{O'}[\Delta h'(O', \boldsymbol{\theta}_t)] \rangle - \frac{L_{J'}}{2}\alpha_t^2\|\delta_t \nabla \log \pi_{\boldsymbol{\theta}_t}(a_t|s_t)\|^2,$$

where the last equality is due to the fact

$$\mathbb{E}_{O'}[h(O', \boldsymbol{\theta}) - \Delta h'(O', \boldsymbol{\theta})] = \mathbb{E}_{O'}[(r(s,a) - J(\boldsymbol{\theta}) + V_{\boldsymbol{\theta}}(s') - V_{\boldsymbol{\theta}}(s))\nabla \log \pi_{\boldsymbol{\theta}}(a|s)] = \nabla J(\boldsymbol{\theta}).$$

Rearranging the above inequality and taking expectation, we have

$$\mathbb{E}\|\nabla J(\boldsymbol{\theta}_t)\|^2 \leq \frac{1}{\alpha_t}(\mathbb{E}[J(\boldsymbol{\theta}_{t+1}) - J(\boldsymbol{\theta}_t)]) - \mathbb{E}\langle \nabla J(\boldsymbol{\theta}_t), \Delta h(O_t, \eta_t, \boldsymbol{\omega}_t, \boldsymbol{\theta}_t) \rangle + \mathbb{E}[\Theta(O_t, \boldsymbol{\theta}_t)]$$
$$- \mathbb{E}\langle \nabla J(\boldsymbol{\theta}_t), \mathbb{E}_{O'}[\Delta h'(O', \boldsymbol{\theta}_t)] \rangle + \frac{L_{J'}}{2}\alpha_t\mathbb{E}\|\delta_t \nabla \log \pi_{\boldsymbol{\theta}_t}(a_t|s_t)\|^2.$$

Note that from Cauchy-Schwartz inequality, we have

$$-\mathbb{E}\langle \nabla J(\boldsymbol{\theta}_t), \Delta h(O_t, \eta_t, \boldsymbol{\omega}_t, \boldsymbol{\theta}_t) \rangle \leq B\sqrt{\mathbb{E}\|\nabla J(\boldsymbol{\theta}_t)\|^2}\sqrt{2\mathbb{E}y_t^2 + 8\mathbb{E}\|\boldsymbol{z}_t\|^2}.$$

From Lemma C.6 and choosing $\tau = \tau_T$, we have

$$\mathbb{E}[\Theta(O_t, \boldsymbol{\theta}_t)] \leq D_1(\tau_T + 1)\sum_{k=t-\tau_T+1}^{t}\mathbb{E}\|\boldsymbol{\theta}_k - \boldsymbol{\theta}_{k-1}\| + D_2 m\rho^{\tau_T-1}$$

$$\leq D_1(\tau_T + 1)G\sum_{k=t-\tau_T}^{t-1}\alpha_k + D_2 m\rho^{\tau_T-1}$$

$$\leq GD_1(\tau_T + 1)^2\alpha_{t-\tau_T} + D_2\frac{1}{\sqrt{T}}.$$

It has been shown that

$$\mathbb{E}_{O'}\|\nabla h'(O, \boldsymbol{\theta})\|^2 = \mathbb{E}_{O'}\|((\boldsymbol{\phi}(s')^\top\boldsymbol{\omega}^* - V_{\boldsymbol{\theta}}(s')) - (\boldsymbol{\phi}(s)^\top\boldsymbol{\omega}^* - V_{\boldsymbol{\theta}}(s)))\nabla \log \pi_{\boldsymbol{\theta}}(a|s)\|^2$$
$$\leq \mathbb{E}_{O'}[B^2((\boldsymbol{\phi}(s')^\top\boldsymbol{\omega}^* - V_{\boldsymbol{\theta}}(s')) - (\boldsymbol{\phi}(s)^\top\boldsymbol{\omega}^* - V_{\boldsymbol{\theta}}(s)))^2]$$
$$\leq \mathbb{E}_{O'}[2B^2(\boldsymbol{\phi}(s')^\top\boldsymbol{\omega}^* - V_{\boldsymbol{\theta}}(s'))^2 + (\boldsymbol{\phi}(s)^\top\boldsymbol{\omega}^* - V_{\boldsymbol{\theta}(s)})^2]$$
$$= 4B^2\mathbb{E}_{O'}[(\boldsymbol{\phi}(s)^\top\boldsymbol{\omega}^* - V_{\boldsymbol{\theta}}(s))^2]$$
$$= 4B^2\epsilon_{\text{app}}^2.$$

Therefore, we have

$$-\langle \nabla J(\boldsymbol{\theta}_t), \mathbb{E}_{O'}[\Delta h'(O', \boldsymbol{\theta}_t)] \rangle \leq L_J\sqrt{\|\mathbb{E}_{O'}[\Delta h'(O_t, \boldsymbol{\theta}_t)]\|^2}$$
$$\leq L_J\sqrt{\mathbb{E}_{O'}\|\Delta h'(O_t, \boldsymbol{\theta}_t)\|^2}$$
$$\leq 2BL_J\epsilon_{\text{app}},$$

where we use $\|\nabla J(\boldsymbol{\theta})\| \leq L_J$ which comes from Lemma B.1. Plugging the three terms yields

$$\mathbb{E}\|\nabla J(\boldsymbol{\theta}_t)\|^2 \leq \frac{1}{\alpha_t}(\mathbb{E}[J(\boldsymbol{\theta}_{t+1})] - \mathbb{E}[J(\boldsymbol{\theta}_t)]) + B\sqrt{\mathbb{E}\|\nabla J(\boldsymbol{\theta}_t)\|^2}\sqrt{2\mathbb{E}y_t^2 + 8\mathbb{E}\|\boldsymbol{z}_t\|^2}$$
$$+ 2BL_J\epsilon_{\text{app}} + GD_1(\tau_T + 1)^2\alpha_{t-\tau_T} + D_2\frac{1}{\sqrt{T}} + \frac{L_{J'}}{2}G^2\alpha_t.$$

Summing over $t$ from $\tau_T$ to $T-1$ gives

$$\sum_{t=\tau_T}^{T-1} \mathbb{E}\|\nabla J(\boldsymbol{\theta}_t)\|^2 \leq \underbrace{\sum_{t=\tau_T}^{T-1} \frac{1}{\alpha_t}(\mathbb{E}[J(\boldsymbol{\theta}_{t+1})] - \mathbb{E}[J(\boldsymbol{\theta}_t)])}_{I_1} + B\sum_{t=\tau_T}^{T-1} \sqrt{\mathbb{E}\|\nabla J(\boldsymbol{\theta}_t)\|^2}\sqrt{2\mathbb{E}y_t^2 + 8\mathbb{E}\|\boldsymbol{z}_t\|^2}$$

$$+ (GD_1(\tau_T+1)^2 + D_2)\frac{T-\tau_T}{\sqrt{T}} + 2BL_J\epsilon_{\text{app}}(T-\tau_T).$$

For term $I_1$, from Abel summation by parts, we have

$$\begin{aligned}
I_1 &= \sum_{t=\tau_T}^{T-1} \frac{1}{\alpha_t}(\mathbb{E}[J(\boldsymbol{\theta}_{t+1})] - \mathbb{E}[J(\boldsymbol{\theta}_t)]) \\
&= \sum_{t=\tau_T+1}^{T-1} (\frac{1}{\alpha_{t-1}} - \frac{1}{\alpha_t})\mathbb{E}[J(\boldsymbol{\theta}_t)] - \mathbb{E}[J(\boldsymbol{\theta}_{\tau_T})]\frac{1}{\alpha_{\tau_T}} + \frac{1}{\alpha_{T-1}}\mathbb{E}[J(\boldsymbol{\theta}_T)] \\
&\leq \sum_{t=\tau_T+1}^{T-1} (\frac{1}{\alpha_t} - \frac{1}{\alpha_{t-1}})U_r + \frac{1}{\alpha_{\tau_T}}U_r + \frac{1}{\alpha_{T-1}}U_r \\
&= \frac{2U_r}{\alpha_{T-1}} \\
&= \frac{2U_r}{c}\sqrt{T}.
\end{aligned}$$

Overall, we have

$$\begin{aligned}
\sum_{t=\tau_T}^{T-1} \mathbb{E}\|\nabla J(\boldsymbol{\theta}_t)\|^2 &\leq \frac{2U_r}{c}\sqrt{T} + (GD_1(\tau_T+1)^2 + D_2)\frac{T-\tau_T}{\sqrt{T}} + 2BL_J\epsilon_{\text{app}}(T-\tau_T) \\
&\quad + B\sum_{t=\tau_T}^{T-1} \sqrt{\mathbb{E}\|\nabla J(\boldsymbol{\theta}_t)\|^2}\sqrt{2\mathbb{E}y_t^2 + 8\mathbb{E}\|\boldsymbol{z}_t\|^2} \\
&\leq \frac{2U_r}{c}\sqrt{T} + (GD_1(\tau_T+1)^2 + D_2)\frac{T-\tau_T}{\sqrt{T}} + 2BL_J\epsilon_{\text{app}}(T-\tau_T) \\
&\quad + B(\sum_{t=\tau_T}^{T-1} \mathbb{E}\|\nabla J(\boldsymbol{\theta}_t)\|^2)^{\frac{1}{2}}(2\sum_{t=\tau_T}^{T-1} \mathbb{E}y_t^2 + 8\sum_{t=\tau_T}^{T-1} \mathbb{E}\|\boldsymbol{z}_t\|^2)^{\frac{1}{2}}.
\end{aligned}$$

Therefore, we get

$$\begin{aligned}
G_T &\leq (\frac{4U_r}{c} + GD_1(\tau_T+1)^2 + D_2)\frac{1}{\sqrt{T}} + 2BL_J\epsilon_{\text{app}} + B\sqrt{G_T(2Y_T + 8Z_T)} \\
&= \mathcal{O}(\frac{\log^2 T}{\sqrt{T}}) + \mathcal{O}(\epsilon_{\text{app}}) + B\sqrt{G_T(2Y_T + 8Z_T)},
\end{aligned}$$

which concludes the proof. □

## C.4 Step 4: Interconnected iteration system analysis

In this subsection, we perform an interconnected iteration system analysis to prove Theorem 3.5.

**Proof of Theorem 3.5.**

*Proof.* Combining (17), (18), and (25), we have

$$Y_T \le \mathcal{O}(\frac{\log^2 T}{\sqrt{T}}) + cG\sqrt{Y_T G_T},$$

$$Z_T \le \mathcal{O}(\frac{\log^2 T}{\sqrt{T}}) + \mathcal{O}(\epsilon_{\text{app}}) + \frac{2}{\lambda}\sqrt{Y_T Z_T} + \frac{2cBL_*}{\lambda}\sqrt{Z_T(2Y_T + 8Z_T)},$$

$$G_T \le \mathcal{O}(\frac{\log^2 T}{\sqrt{T}}) + \mathcal{O}(\epsilon_{\text{app}}) + B\sqrt{G_T(2Y_T + 8Z_T)}.$$

Denote

$$l_1 := cG,$$
$$l_2 := \frac{2}{\lambda},$$
$$l_3 := \frac{2cBL_*}{\lambda},$$
$$l_4 := B.$$

Then we have

$$Y_T \le \mathcal{O}(\frac{\log^2 T}{\sqrt{T}}) + l_1\sqrt{Y_T G_T},$$

$$Z_T \le \mathcal{O}(\frac{\log^2 T}{\sqrt{T}}) + \mathcal{O}(\epsilon_{\text{app}}) + l_2\sqrt{Y_T Z_T} + l_3\sqrt{Z_T(2Y_T + 8Z_T)},$$

$$G_T \le \mathcal{O}(\frac{\log^2 T}{\sqrt{T}}) + \mathcal{O}(\epsilon_{\text{app}}) + l_4\sqrt{G_T(2Y_T + 8Z_T)}.$$

For $G_T$, we get

$$G_T \le \mathcal{O}(\frac{\log^2 T}{\sqrt{T}}) + \mathcal{O}(\epsilon_{\text{app}}) + \frac{1}{2}G_T + l_4^2(Y_T + 4Z_T),$$

$$G_T \le \mathcal{O}(\frac{\log^2 T}{\sqrt{T}}) + \mathcal{O}(\epsilon_{\text{app}}) + 2l_4^2(Y_T + 4Z_T). \tag{26}$$

For $Z_T$, we have

$$Z_T \le \mathcal{O}(\frac{\log^2 T}{\sqrt{T}}) + \mathcal{O}(\epsilon_{\text{app}}) + \frac{1}{4}Z_T + l_2^2 Y_T + 4l_3 Z_T + \frac{l_3}{2}Y_T.$$

If it satisfies $4l_3 \le \frac{1}{4}$, we further have

$$Z_T \le \mathcal{O}(\frac{\log^2 T}{\sqrt{T}}) + \mathcal{O}(\epsilon_{\text{app}}) + (2l_2^2 + l_3)Y_T. \tag{27}$$

For $Y_T$, we get

$$Y_T \le \mathcal{O}(\frac{\log^2 T}{\sqrt{T}}) + \frac{l_1}{2}(Y_T + G_T).$$

Plugging (26) and (27) into the above inequality gives

$$Y_T \le \mathcal{O}(\frac{\log^2 T}{\sqrt{T}}) + \mathcal{O}(\epsilon_{\text{app}}) + \frac{l_1}{2}(Y_T + 2l_4^2 Y_T + 8l_4^2 Z_T)$$

$$\le \mathcal{O}(\frac{\log^2 T}{\sqrt{T}}) + \mathcal{O}(\epsilon_{\text{app}}) + \frac{l_1}{2}(Y_T + 2l_4^2 Y_T + 8l_4^2(2l_2^2 + l_3)Y_T)$$

$$= \mathcal{O}(\frac{\log^2 T}{\sqrt{T}}) + \mathcal{O}(\epsilon_{\text{app}}) + \frac{l_1}{2}(1 + 2l_4^2 + 8l_4^2(2l_2^2 + l_3))Y_T.$$

Therefore, if $l_1(1 + 2l_4^2 + 8l_4^2(2l_2^2 + l_3)) \leq 1$, we have

$$Y_T \leq \mathcal{O}(\frac{\log^2 T}{\sqrt{T}}) + \mathcal{O}(\epsilon_{\text{app}}).$$

Overall, we require

$$4l_3 \leq \frac{1}{4},$$

$$l_1(1 + 2l_4^2 + 8l_4^2(2l_2^2 + l_3)) \leq 1.$$

According to the definition of $l_1, l_2, l_3, l_4$, we have

$$\frac{8cBL_*}{\lambda} \leq \frac{1}{4},$$

$$cG(1 + 2B^2 + 8B^2(\frac{8}{\lambda^2} + \frac{2cBL_*}{\lambda})) \leq 1.$$

Thus we choose

$$c = \min\{\frac{\lambda}{32BL_*}, \frac{\lambda^2}{G(\lambda^2 + 3B^2\lambda^2 + 64B^2)}\}, \tag{28}$$

which satisfies the above two inequalities. Therefore, we have

$$Y_T = \mathcal{O}(\frac{\log^2 T}{\sqrt{T}}) + \mathcal{O}(\epsilon_{\text{app}}),$$

and consequently,

$$Z_T = \mathcal{O}(\frac{\log^2 T}{\sqrt{T}}) + \mathcal{O}(\epsilon_{\text{app}}),$$

$$G_T = \mathcal{O}(\frac{\log^2 T}{\sqrt{T}}) + \mathcal{O}(\epsilon_{\text{app}}).$$

Thus we conclude our proof. $\qquad\square$

# D   Proof of Supporting Lemmas

The following three lemmas only deal with the Markovian noise, which are originally proved in [31] and updated in [30]. We include the proof with slight modifications for proving Theorem 3.5.

**Proof of Lemma C.1.**

*Proof.* We will divide the proof of this lemma into four steps.

**Step 1:** show that for any $\boldsymbol{\theta}_1, \boldsymbol{\theta}_2, \eta, O = (s, a, s')$, we have

$$|\Phi(O, \eta, \boldsymbol{\theta}_1) - \Phi(O, \eta, \boldsymbol{\theta}_2)| \leq 4U_r L_J \|\boldsymbol{\theta}_1 - \boldsymbol{\theta}_2\|. \tag{29}$$

By the definition of $\Phi(O, \eta, \boldsymbol{\theta})$ in (13), we have

$$\begin{aligned}
|\Phi(O, \eta, \boldsymbol{\theta}_1) - \Phi(O, \boldsymbol{\theta}, \boldsymbol{\theta}_2)| &= |(\eta - J(\boldsymbol{\theta}_1))(r - J(\boldsymbol{\theta}_1)) - (\eta - J(\boldsymbol{\theta}_2))(r - J(\boldsymbol{\theta}_2))| \\
&\leq |(\eta - J(\boldsymbol{\theta}_1))(r - J(\boldsymbol{\theta}_1)) - (\eta - J(\boldsymbol{\theta}_1))(r - J(\boldsymbol{\theta}_2))| \\
&\quad + |(\eta - J(\boldsymbol{\theta}_1))(r - J(\boldsymbol{\theta}_2)) - (\eta - J(\boldsymbol{\theta}_2))(r - J(\boldsymbol{\theta}_2))| \\
&\leq 4U_r|J(\boldsymbol{\theta}_1) - J(\boldsymbol{\theta}_2)| \\
&\leq 4U_r L_J \|\boldsymbol{\theta}_1 - \boldsymbol{\theta}_2\|.
\end{aligned}$$

**Step 2:** show that for any $\boldsymbol{\theta}, \eta_1, \eta_2, O$, we have

$$|\Phi(O, \eta_1, \boldsymbol{\theta}) - \Phi(O, \eta_2, \boldsymbol{\theta}) \leq 2U_r|\eta_1 - \eta_2|. \tag{30}$$

By definition, we have

$$\begin{aligned}
|\Phi(O, \eta_1, \boldsymbol{\theta}) - \Phi(O, \eta_2, \boldsymbol{\theta})| &= |(\eta_1 - J(\boldsymbol{\theta}))(r - J(\boldsymbol{\theta})) - (\eta_2 - J(\boldsymbol{\theta}))(r - J(\boldsymbol{\theta}))| \\
&\leq 2U_r|\eta_1 - \eta_2|.
\end{aligned}$$

**Step 3:** show that for original tuple $O_t$ and the auxiliary tuple $\widetilde{O}_t$, conditioned on $s_{t-\tau-1}$ and $\boldsymbol{\theta}_{t-\tau}$, we have

$$|\mathbb{E}[\Phi(O_t, \eta_{t-\tau}, \boldsymbol{\theta}_{t-\tau}) - \mathbb{E}[\Phi(\widetilde{O}_t, \eta_{t-\tau}, \boldsymbol{\theta}_{t-\tau})]| \leq 2U_r^2|\mathcal{A}|L_\pi \sum_{k=t-\tau}^{t} \mathbb{E}\|\boldsymbol{\theta}_k - \boldsymbol{\theta}_{t-\tau}\|. \quad (31)$$

By definition, we have

$$\mathbb{E}[\Phi(O_t, \eta_{t-\tau}, \boldsymbol{\theta}_{t-\tau}) - \mathbb{E}[\Phi(\widetilde{O}_t, \eta_{t-\tau}, \boldsymbol{\theta}_{t-\tau})] = (\eta_{t-\tau} - J(\boldsymbol{\theta}_{t-\tau}))\mathbb{E}[r(s_t, a_t) - r(\widetilde{s}_t, \widetilde{a}_t)].$$

By definition of total variation norm, we have

$$\mathbb{E}[r(s_t, a_t) - r(\widetilde{s}_t, \widetilde{a}_t)] \leq 2U_r d_{TV}(\mathbb{P}(O_t \in \cdot|s_{t-\tau+1}, \boldsymbol{\theta}_{t-\tau}), \mathbb{P}(\widetilde{O}_t \in \cdot|s_{t-\tau+1}, \boldsymbol{\theta}_{t-\tau})). \quad (32)$$

By Lemma B.6, we get

$$\begin{aligned}
&d_{TV}(\mathbb{P}(O_t \in \cdot|s_{t-\tau+1}, \boldsymbol{\theta}_{t-\tau}), \mathbb{P}(\widetilde{O}_t \in \cdot|s_{t-\tau+1}, \boldsymbol{\theta}_{t-\tau})) \\
&= d_{TV}(\mathbb{P}((s_t, a_t) \in \cdot|s_{t-\tau+1}, \boldsymbol{\theta}_{t-\tau}), \mathbb{P}((\widetilde{s}_t, \widetilde{a}_t) \in \cdot|s_{t-\tau+1}, \boldsymbol{\theta}_{t-\tau})) \\
&\leq d_{TV}(\mathbb{P}(s_t \in \cdot|s_{t-\tau+1}, \boldsymbol{\theta}_{t-\tau}), \mathbb{P}(\widetilde{s}_t \in \cdot|s_{t-\tau+1}, \boldsymbol{\theta}_{t-\tau})) + \frac{1}{2}L_\pi\mathbb{E}\|\boldsymbol{\theta}_t - \boldsymbol{\theta}_{t-\tau}\| \\
&\leq d_{TV}(\mathbb{P}(O_{t-1} \in \cdot|s_{t-\tau+1}, \boldsymbol{\theta}_{t-\tau}), \mathbb{P}(\widetilde{O}_{t-1} \in \cdot|s_{t-\tau+1}, \boldsymbol{\theta}_{t-\tau})) + \frac{1}{2}L_\pi\mathbb{E}\|\boldsymbol{\theta}_t - \boldsymbol{\theta}_{t-\tau}\|.
\end{aligned}$$

Repeat the above argument from $t$ to $t - \tau + 1$, we have

$$d_{TV}(\mathbb{P}(O_t \in \cdot|s_{t-\tau+1}, \boldsymbol{\theta}_{t-\tau}), \mathbb{P}(\widetilde{O}_t \in \cdot|s_{t-\tau+1}, \boldsymbol{\theta}_{t-\tau})) \leq \frac{1}{2}|\mathcal{A}|L_\pi \sum_{k=t-\tau}^{t} \mathbb{E}\|\boldsymbol{\theta}_k - \boldsymbol{\theta}_{t-\tau}\|. \quad (33)$$

Plugging (33) into (32), we have

$$|\mathbb{E}[\Phi(O_t, \eta_{t-\tau}, \boldsymbol{\theta}_{t-\tau}) - \mathbb{E}[\Phi(\widetilde{O}_t, \eta_{t-\tau}, \boldsymbol{\theta}_{t-\tau})]| \leq 2U_r^2|\mathcal{A}|L_\pi \sum_{k=t-\tau}^{t} \mathbb{E}\|\boldsymbol{\theta}_k - \boldsymbol{\theta}_{t-\tau}\|.$$

**Step 4:** show that conditioned on $s_{t-\tau+1}$ and $\boldsymbol{\theta}_{t-\tau}$, we have

$$\mathbb{E}[\Phi(\widetilde{O}_t, \eta_{t-\tau}, \boldsymbol{\theta}_{t-\tau})] \leq 4U_r^2 m\rho^{\tau-1}. \quad (34)$$

Note that according to definition, we have

$$\mathbb{E}[\Phi(O'_{t-\tau}, \eta_{t-\tau}, \boldsymbol{\theta}_{t-\tau})|\boldsymbol{\theta}_{t-\tau}] = 0,$$

where $O'_{t-\tau} = (s'_{t-\tau}, a'_{t-\tau}, s'_{t-\tau+1})$ is the tuple generated by $s'_{t-\tau} \sim \mu_{t-\tau}, a'_{t-\tau} \sim \pi_{\boldsymbol{\theta}_{t-\tau}}, s'_{t-\tau+1} \sim \mathcal{P}$. From the uniform ergodicity in Assumption 3.2, it shows that

$$d_{TV}(\mathbb{P}(\widetilde{s}_t = \cdot|s_{t-\tau+1}, \boldsymbol{\theta}_{t-\tau}), \mu_{\boldsymbol{\theta}_{t-\tau}}) \leq m\rho^{\tau-1}.$$

Then we have

$$\begin{aligned}
\mathbb{E}[\Phi(\widetilde{O}_t, \eta_{t-\tau}, \boldsymbol{\theta}_{t-\tau})] &= \mathbb{E}[\Phi(\widetilde{O}_t, \eta_{t-\tau}, \boldsymbol{\theta}_{t-\tau}) - \Phi(O'_{t-\tau}, \eta_{t-\tau}, \boldsymbol{\theta}_{t-\tau})] \\
&= \mathbb{E}[(\eta_{t-\tau} - J(\boldsymbol{\theta}_{t-\tau}))(r(\widetilde{s}_t, \widetilde{a}_t) - r(s'_{t-\tau}, a'_{t-\tau}))] \\
&\leq 4U_r^2 d_{TV}(\mathbb{P}(\widetilde{O}_{t-\tau} = \cdot|s_{t-\tau+1}, \boldsymbol{\theta}_{t-\tau}), \mu_{\boldsymbol{\theta}_{t-\tau}} \otimes \pi_{\boldsymbol{\theta}_{t-\tau}} \otimes \mathcal{P}) \\
&\leq 4U_r^2 m\rho^{\tau-1}.
\end{aligned}$$

Combing (29), (30), (31), and (34), we have

$$\begin{aligned}
\mathbb{E}[\Phi(O_t, \eta_t, \boldsymbol{\theta}_t)] &= \mathbb{E}[\Phi(O_t, \eta_t, \boldsymbol{\theta}_t) - \Phi(O_t, \eta_t, \boldsymbol{\theta}_{t-\tau})] + \mathbb{E}[\Phi(O_t, \eta_t, \boldsymbol{\theta}_{t-\tau}) - \Phi(O_t, \eta_{t-\tau}, \boldsymbol{\theta}_{t-\tau})] \\
&\quad + \mathbb{E}[\Phi(O_t, \eta_{t-\tau}, \boldsymbol{\theta}_{t-\tau}) - \Phi(\widetilde{O}_t, \eta_{t-\tau}, \boldsymbol{\theta}_{t-\tau})] + \mathbb{E}[\Phi(\widetilde{O}_t, \eta_{t-\tau}, \boldsymbol{\theta}_{t-\tau})] \\
&\leq 4U_r L_J\|\boldsymbol{\theta}_t - \boldsymbol{\theta}_{t-\tau}\| + 2U_r|\eta_t - \eta_{t-\tau}| + 2U_r^2|\mathcal{A}|L_\pi \sum_{i=t-\tau}^{t} \mathbb{E}\|\boldsymbol{\theta}_i - \boldsymbol{\theta}_{t-\tau}\| \\
&\quad + 4U_r^2 m\rho^{\tau-1},
\end{aligned}$$

which concludes the proof. □

**Proof of Lemma C.3**.

*Proof.* We will divide the proof of this lemma into four steps.

**Step 1:** show that for any $\boldsymbol{\theta}_1, \boldsymbol{\theta}_2, \boldsymbol{\omega}$ and tuple $O = (s, a, s')$, we have

$$|\Psi(O, \boldsymbol{\omega}, \boldsymbol{\theta}_1) - \Psi(O, \boldsymbol{\omega}, \boldsymbol{\theta}_2) \le C_1 \|\boldsymbol{\theta}_1 - \boldsymbol{\theta}_2\|, \tag{35}$$

where $C_1 = 2U_\delta^2 |\mathcal{A}| L_\pi (1 + \lceil \log_\rho m^{-1} \rceil + \frac{1}{1-\rho}) + 2U_\delta L_J + 2U_\delta L_*$.

By definition of $\Psi(O, \boldsymbol{\omega}, \boldsymbol{\theta})$ in (13), we have

$$
\begin{aligned}
& |\Psi(O, \boldsymbol{\omega}, \boldsymbol{\theta}_1) - \Psi(O, \boldsymbol{\omega}, \boldsymbol{\theta}_2)| \\
={} & |\langle \boldsymbol{\omega} - \boldsymbol{\omega}_1^*, g(O, \boldsymbol{\omega}, \boldsymbol{\theta}_1) - \bar{g}(\boldsymbol{\omega}, \boldsymbol{\theta}_1)\rangle - \langle \boldsymbol{\omega} - \boldsymbol{\omega}_2^*, g(O, \boldsymbol{\omega}, \boldsymbol{\theta}_2) - \bar{g}(\boldsymbol{\omega}, \boldsymbol{\theta}_2)\rangle| \\
\le{} & \underbrace{|\langle \boldsymbol{\omega} - \boldsymbol{\omega}_1^*, g(O, \boldsymbol{\omega}, \boldsymbol{\theta}_1) - \bar{g}(\boldsymbol{\omega}, \boldsymbol{\theta}_1)\rangle - \langle \boldsymbol{\omega} - \boldsymbol{\omega}_1^*, g(O, \boldsymbol{\omega}, \boldsymbol{\theta}_2) - \bar{g}(\boldsymbol{\omega}, \boldsymbol{\theta}_2)\rangle|}_{I_1} \\
& + \underbrace{|\langle \boldsymbol{\omega} - \boldsymbol{\omega}_1^*, g(O, \boldsymbol{\omega}, \boldsymbol{\theta}_2) - \bar{g}(\boldsymbol{\omega}, \boldsymbol{\theta}_2)\rangle - \langle \boldsymbol{\omega} - \boldsymbol{\omega}_2^*, g(O, \boldsymbol{\omega}, \boldsymbol{\theta}_2) - \bar{g}(\boldsymbol{\omega}, \boldsymbol{\theta}_2)\rangle|}_{I_2}.
\end{aligned}
$$

For term $I_1$, we have

$$
\begin{aligned}
I_1 ={} & |\langle \boldsymbol{\omega} - \boldsymbol{\omega}_1^*, g(O, \boldsymbol{\omega}, \boldsymbol{\theta}_1) - \bar{g}(\boldsymbol{\omega}, \boldsymbol{\theta}_1)\rangle - \langle \boldsymbol{\omega} - \boldsymbol{\omega}_1^*, g(O, \boldsymbol{\omega}, \boldsymbol{\theta}_2) - \bar{g}(\boldsymbol{\omega}, \boldsymbol{\theta}_2)\rangle| \\
={} & |\langle \boldsymbol{\omega} - \boldsymbol{\omega}_1^*, g(O, \boldsymbol{\omega}, \boldsymbol{\theta}_1) - g(O, \boldsymbol{\omega}, \boldsymbol{\theta}_2)\rangle| + |\langle \boldsymbol{\omega} - \boldsymbol{\omega}_1^*, \bar{g}(\boldsymbol{\omega}, \boldsymbol{\theta}_1) - \bar{g}(\boldsymbol{\omega}, \boldsymbol{\theta}_2)\rangle| \\
={} & |\langle \boldsymbol{\omega} - \boldsymbol{\omega}_1^*, \phi(s)(J(\boldsymbol{\theta}_1) - J(\boldsymbol{\theta}_2))\rangle| + |\langle \boldsymbol{\omega} - \boldsymbol{\omega}_1^*, \bar{g}(\boldsymbol{\omega}, \boldsymbol{\theta}_1) - \bar{g}(\boldsymbol{\omega}, \boldsymbol{\theta}_2)\rangle| \\
\le{} & 2U_{\boldsymbol{\omega}} L_J \|\boldsymbol{\theta}_1 - \boldsymbol{\theta}_2\| + 2U_{\boldsymbol{\omega}} \|\bar{g}(\boldsymbol{\omega}, \boldsymbol{\theta}_1) - \bar{g}(\boldsymbol{\omega}, \boldsymbol{\theta}_2)\| \\
\le{} & 2U_{\boldsymbol{\omega}} L_J \|\boldsymbol{\theta}_1 - \boldsymbol{\theta}_2\| + 2U_{\boldsymbol{\omega}} \cdot 2U_\delta d_{TV}(\mu_{\boldsymbol{\theta}_1} \otimes \pi_{\boldsymbol{\theta}_1} \otimes \mathcal{P}, \mu_{\boldsymbol{\theta}_2} \otimes \pi_{\boldsymbol{\theta}_2} \otimes \mathcal{P}) \\
\le{} & 2U_{\boldsymbol{\omega}} L_J \|\boldsymbol{\theta}_1 - \boldsymbol{\theta}_2\| + 2U_\delta^2 d_{TV}(\mu_{\boldsymbol{\theta}_1} \otimes \pi_{\boldsymbol{\theta}_1} \otimes \mathcal{P}, \mu_{\boldsymbol{\theta}_2} \otimes \pi_{\boldsymbol{\theta}_2} \otimes \mathcal{P}) \\
\le{} & (2U_\delta L_J + 2U_\delta^2 |\mathcal{A}| L_\pi (1 + \lceil \log_\rho m^{-1} \rceil + \frac{1}{1 - \rho}) \|\boldsymbol{\theta}_1 - \boldsymbol{\theta}_2\|,
\end{aligned}
$$

where we use the fact that $U_\delta = 2U_r + 2U_{\boldsymbol{\omega}}$ and the last inequality comes from Lemma B.5.

For term $I_2$, from Cauchy-Schwartz inequality, we have

$$
\begin{aligned}
I_2 ={} & |\langle \boldsymbol{\omega} - \boldsymbol{\omega}_1^*, g(O, \boldsymbol{\omega}, \boldsymbol{\theta}_2) - \bar{g}(\boldsymbol{\omega}, \boldsymbol{\theta}_2)\rangle - \langle \boldsymbol{\omega} - \boldsymbol{\omega}_2^*, g(O, \boldsymbol{\omega}, \boldsymbol{\theta}_2) - \bar{g}(\boldsymbol{\omega}, \boldsymbol{\theta}_2)\rangle| \\
={} & |\langle \boldsymbol{\omega}_1^* - \boldsymbol{\omega}_2^*, g(O, \boldsymbol{\omega}, \boldsymbol{\theta}_2) - \bar{g}(\boldsymbol{\omega}, \boldsymbol{\theta}_2)\rangle| \\
\le{} & 2U_\delta \|\boldsymbol{\omega}_1^* - \boldsymbol{\omega}_2^*\| \\
\le{} & 2U_\delta L_* \|\boldsymbol{\theta}_1 - \boldsymbol{\theta}_2\|.
\end{aligned}
$$

Combining the results from $I_1$ and $I_2$, we get

$$|\Psi(O, \boldsymbol{\omega}, \boldsymbol{\theta}_1) - \Psi(O, \boldsymbol{\omega}, \boldsymbol{\theta}_2) \le C_1 \|\boldsymbol{\theta}_1 - \boldsymbol{\theta}_2\|,$$

where $C_1 = 2U_\delta^2 |\mathcal{A}| L_\pi (1 + \lceil \log_\rho m^{-1} \rceil + \frac{1}{1-\rho}) + 2U_\delta L_J + 2U_\delta L_*$.

**Step 2:** show that for any $\boldsymbol{\theta}, \boldsymbol{\omega}_1, \boldsymbol{\omega}_2$ and tuple $O(s, a, s')$, we have

$$|\Psi(O, \boldsymbol{\omega}_1, \boldsymbol{\theta}) - \Psi(O, \boldsymbol{\omega}_2, \boldsymbol{\theta})| \le 6U_\delta \|\boldsymbol{\omega}_1 - \boldsymbol{\omega}_2\|. \tag{36}$$

By definition, we have

$$
\begin{aligned}
& |\Psi(O, \boldsymbol{\omega}_1, \boldsymbol{\theta}) - \Psi(O, \boldsymbol{\omega}_2, \boldsymbol{\theta})| \\
={} & |\langle \boldsymbol{\omega}_1 - \boldsymbol{\omega}^*, g(O, \boldsymbol{\omega}_1, \boldsymbol{\theta}) - \bar{g}(\boldsymbol{\omega}_1, \boldsymbol{\theta})\rangle - \langle \boldsymbol{\omega}_2 - \boldsymbol{\omega}^*, g(O, \boldsymbol{\omega}_2, \boldsymbol{\theta}) - \bar{g}(\boldsymbol{\omega}_2, \boldsymbol{\theta})\rangle| \\
\le{} & |\langle \boldsymbol{\omega}_1 - \boldsymbol{\omega}^*, g(O, \boldsymbol{\omega}_1, \boldsymbol{\theta}) - \bar{g}(\boldsymbol{\omega}_1, \boldsymbol{\theta})\rangle - \langle \boldsymbol{\omega}_1 - \boldsymbol{\omega}^*, g(O, \boldsymbol{\omega}_2, \boldsymbol{\theta}) - \bar{g}(\boldsymbol{\omega}_2, \boldsymbol{\theta})\rangle| \\
& + |\langle \boldsymbol{\omega}_1 - \boldsymbol{\omega}^*, g(O, \boldsymbol{\omega}_2, \boldsymbol{\theta}) - \bar{g}(\boldsymbol{\omega}_2, \boldsymbol{\theta})\rangle - \langle \boldsymbol{\omega}_2 - \boldsymbol{\omega}^*, g(O, \boldsymbol{\omega}_2, \boldsymbol{\theta}) - \bar{g}(\boldsymbol{\omega}_2, \boldsymbol{\theta})\rangle| \\
\le{} & 2U_{\boldsymbol{\omega}} \|(g(O, \boldsymbol{\omega}_1) - g(O, \boldsymbol{\omega}_2)) - (\bar{g}(\boldsymbol{\omega}_1, \boldsymbol{\theta}) - \bar{g}(\boldsymbol{\omega}_2, \boldsymbol{\theta}))\| + 2U_\delta \|\boldsymbol{\omega}_1 - \boldsymbol{\omega}_2\| \\
\le{} & 6U_\delta \|\boldsymbol{\omega}_1 - \boldsymbol{\omega}_2\|,
\end{aligned}
$$

where the last inequality is due to $\|g(O, \boldsymbol{\omega}_1, \boldsymbol{\theta}) - g(O, \boldsymbol{\omega}_2, \boldsymbol{\theta})\| = |(\phi(s') - \phi(s))^\top (\boldsymbol{\omega}_1 - \boldsymbol{\omega}_2)| \le 2\|\boldsymbol{\omega}_1 - \boldsymbol{\omega}_2\|$, $\|\bar{g}(\boldsymbol{\omega}_1, \boldsymbol{\theta}) - \bar{g}(\boldsymbol{\omega}_2, \boldsymbol{\theta})\| \le 2\|\boldsymbol{\omega}_1 - \boldsymbol{\omega}_2\|$, and $2U_{\boldsymbol{\omega}} \le U_\delta$.

**Step 3:** show that for tuples $O_t = (s_t, a_t, s_{t+1})$ and $\widetilde{O}_t = (\widetilde{s}_t, \widetilde{a}_t, \widetilde{s}_{t+1})$. Conditioning on $s_{t-\tau+1}$ and $\boldsymbol{\theta}_{t-\tau}$, we have

$$\mathbb{E}[\Psi(O_t, \boldsymbol{\omega}_{t-\tau}, \boldsymbol{\theta}_{t-\tau}) - \Psi(\widetilde{O}_t, \boldsymbol{\omega}_{t-\tau}, \boldsymbol{\theta}_{t-\tau})] \leq U_\delta^2 |\mathcal{A}| L_\pi \sum_{k=t-\tau}^{t} \mathbb{E}\|\boldsymbol{\theta}_k - \boldsymbol{\theta}_{t-\tau}\|. \tag{37}$$

By the definition of total variation norm, we have

$$\mathbb{E}[\Psi(O_t, \boldsymbol{\omega}_{t-\tau}, \boldsymbol{\theta}_{t-\tau}) - \Psi(\widetilde{O}_t, \boldsymbol{\omega}_{t-\tau}, \boldsymbol{\theta}_{t-\tau})]$$
$$\leq \mathbb{E}[\langle \boldsymbol{\omega}_{t-\tau} - \boldsymbol{\omega}_{t-\tau}^*, g(O_t, \boldsymbol{\omega}_{t-\tau}, \boldsymbol{\theta}_{t-\tau}) - g(\widetilde{O}_t, \boldsymbol{\omega}_{t-\tau}, \boldsymbol{\theta}_{t-\tau}))]$$
$$\leq 2U_\delta^2 d_{TV}(\mathbb{P}(O_t \in \cdot | s_{t-\tau+1}, \boldsymbol{\theta}_{-\tau}), \mathbb{P}(\widetilde{O}_t \in \cdot | s_{t-\tau+1}, \boldsymbol{\theta}_{t-\tau}))$$
$$\overset{(1)}{\leq} U_\delta^2 |\mathcal{A}| L_\pi \sum_{k=t-\tau}^{t} \mathbb{E}\|\boldsymbol{\theta}_k - \boldsymbol{\theta}_{t-\tau}\|$$
$$\leq U_\delta^2 |\mathcal{A}| L_\pi G \tau(\tau+1) \alpha_{t-\tau},$$

where (1) follows from (33).

**Step 4:** show that conditioning on $s_{t-\tau+1}$ and $\boldsymbol{\theta}_{t-\tau}$,

$$\mathbb{E}[\Psi(\widetilde{O}_t, \boldsymbol{\omega}_{t-\tau}, \boldsymbol{\theta}_{t-\tau})] \leq 2U_\delta^2 m \rho^{\tau-1} \tag{38}$$

From the definition of $\Psi(O, \boldsymbol{\omega}, \boldsymbol{\theta})$, we have

$$\mathbb{E}[\Psi(O'_{t-\tau}, \boldsymbol{\omega}_{t-\tau}, \boldsymbol{\theta}_{t-\tau}) | s_{t-\tau+1}, \boldsymbol{\theta}_{t-\tau}] = 0,$$

where $O'_{t-\tau}$ is the tuple generated by $s'_{t-\tau} \sim \mu_{\boldsymbol{\theta}_{t-\tau}}, a'_{t-\tau} \sim \pi_{\boldsymbol{\theta}_{t-\tau}}, s'_{t-\tau+1} \sim \mathcal{P}$. From Assumption 3.2, we have

$$d_{TV}(\mathbb{P}(\widetilde{s}_t = \cdot | s_{t-\tau+1}, \boldsymbol{\theta}_{t-\tau}), \mu_{\boldsymbol{\theta}_{t-\tau}}) \leq m \rho^{\tau-1}.$$

Then, it holds that

$$\mathbb{E}[\Psi(\widetilde{O}_t, \boldsymbol{\omega}_{t-\tau}, \boldsymbol{\theta}_{t-\tau})] = \mathbb{E}[\Psi(\widetilde{O}_t, \boldsymbol{\omega}_{t-\tau}, \boldsymbol{\theta}_{t-\tau}) - \Psi(O'_{t-\tau}, \boldsymbol{\omega}_{t-\tau}, \boldsymbol{\theta}_{t-\tau})]$$
$$= \mathbb{E}\langle \boldsymbol{\omega}_{t-\tau} - \boldsymbol{\omega}_{t-\tau}^*, g(\widetilde{O}_t, \boldsymbol{\omega}_{t-\tau}, \boldsymbol{\theta}_{t-\tau} - g(O'_{t-\tau}, \boldsymbol{\omega}_{t-\tau}, \boldsymbol{\theta}_{t-\tau})\rangle$$
$$\leq 4U_{\boldsymbol{\omega}} U_\delta d_{TV}(\mathbb{P}(\widetilde{O}_t = \cdot | s_{t-\tau+1}, \boldsymbol{\theta}_{t-\tau}), \mu_{\boldsymbol{\theta}_{t-\tau}} \otimes \pi_{\boldsymbol{\theta}_{t-\tau}} \otimes \mathcal{P})$$
$$\leq 2U_\delta^2 d_{TV}(\mathbb{P}(\widetilde{O}_t = \cdot | s_{t-\tau+1}, \boldsymbol{\theta}_{t-\tau}), \mu_{\boldsymbol{\theta}_{t-\tau}} \otimes \pi_{\boldsymbol{\theta}_{t-\tau}} \otimes \mathcal{P})$$
$$= 2U_\delta^2 d_{TV}(\mathbb{P}((\widetilde{s}_t, \widetilde{a}_t) \in \cdot | s_{t-\tau+1}, \boldsymbol{\theta}_{t-\tau}), \mu_{\boldsymbol{\theta}_{t-\tau}} \otimes \pi_{\boldsymbol{\theta}_{t-\tau}})$$
$$= 2U_\delta^2 d_{TV}(\mathbb{P}(\widetilde{s}_t = \cdot | s_{t-\tau+1}, \boldsymbol{\theta}_{t-\tau}), \mu_{\boldsymbol{\theta}_{t-\tau}})$$
$$\leq 2U_\delta^2 m \rho^{\tau-1}.$$

Combining (35), (36), (37), and (38), we have

$$\mathbb{E}[\Psi(O_t, \boldsymbol{\omega}_t, \boldsymbol{\theta}_t)] = \mathbb{E}[\Psi(O_t, \boldsymbol{\omega}_t, \boldsymbol{\theta}_t) - \Psi(O_t, \boldsymbol{\omega}_t, \boldsymbol{\theta}_{t-\tau})]$$
$$+ \mathbb{E}[\Psi(O_t, \boldsymbol{\omega}_t, \boldsymbol{\theta}_{t-\tau}) - \Psi(O_t, \boldsymbol{\omega}_{t-\tau}, \boldsymbol{\theta}_{t-\tau})]$$
$$+ \mathbb{E}[\Psi(O_t, \boldsymbol{\omega}_{t-\tau}, \boldsymbol{\theta}_{t-\tau}) - \Psi(\widetilde{O}_t, \boldsymbol{\omega}_{t-\tau}, \boldsymbol{\theta}_{t-\tau})]$$
$$+ \mathbb{E}[\Psi(\widetilde{O}_t, \boldsymbol{\omega}_{t-\tau}, \boldsymbol{\theta}_{t-\tau})]$$
$$\leq C_1 \|\boldsymbol{\theta}_t - \boldsymbol{\theta}_{t-\tau}\| + U_\delta^2 |\mathcal{A}| L_\pi G \tau(\tau+1) \alpha_{t-\tau}$$
$$+ 2U_\delta^2 m \rho^{\tau-1} + 6U_\delta \|\boldsymbol{\omega}_t - \boldsymbol{\omega}_{t-\tau}\|,$$

where $C_1 = 2U_\delta^2 |\mathcal{A}| L_\pi (1 + \lceil \log_\rho m^{-1} \rceil + \frac{1}{1-\rho}) + 2U_\delta (L_J + L_*)$. $\qquad \square$

**Proof of Lemma C.4.**

*Proof.* We will divide the proof of this lemma into four steps.

**Step 1:** show that
$$\|\Xi(O_t, \boldsymbol{\omega}_t, \boldsymbol{\theta}_t) - \Xi(O_t, \boldsymbol{\omega}_t, \boldsymbol{\theta}_{t-\tau})\| \leq (3U_\delta L_h + 2U_\delta BL_*)\|\boldsymbol{\theta}_t - \boldsymbol{\theta}_{t-\tau}\| \tag{39}$$

Since $\Xi(O_t, \boldsymbol{\omega}_t, \boldsymbol{\theta}_t) = \langle \boldsymbol{\omega}_t - \boldsymbol{\omega}_t^*, \mathbb{E}_{O'}[h(O', \boldsymbol{\theta})] - h(O, \boldsymbol{\theta})\rangle$, we define $\mathbb{E}_{\boldsymbol{\theta}}[h(O', \boldsymbol{\theta})] := \mathbb{E}_{O'}[h(O', \boldsymbol{\theta})]$, where $\mathbb{E}_{\boldsymbol{\theta}}$ is the shorthand of $\mathbb{E}_{O' \sim (\mu_{\boldsymbol{\theta}}, \pi_{\boldsymbol{\theta}}, \mathcal{P})}$. In the following, we will show that each term in $\Xi(O_t, \boldsymbol{\omega}_t, \boldsymbol{\theta}_t)$ is Lipschitz.

Term $\boldsymbol{\omega}_t$ is not related to $\boldsymbol{\theta}$ and term $\boldsymbol{\omega}_t^* := \boldsymbol{\omega}^*(\boldsymbol{\theta}_t)$ is $L_*$-Lipschitz.

For term $h(O, \boldsymbol{\theta})$, denote $\delta(O, \boldsymbol{\theta}) := r(s, a) - r(\boldsymbol{\theta}) + (\boldsymbol{\phi}(s') - \boldsymbol{\phi}(s))^\top \boldsymbol{\omega}^*$, we have
$$\begin{aligned}
&\|h(O, \boldsymbol{\theta}_1) - h(O, \boldsymbol{\theta}_2)\| \\
&= \|\delta(O, \boldsymbol{\theta}_1)\nabla \log \pi_{\boldsymbol{\theta}_1}(a|s) - \delta(O_t, \boldsymbol{\theta}_2)\nabla \log \pi_{\boldsymbol{\theta}_2}(a|s)\| \\
&\leq \|\delta(O, \boldsymbol{\theta}_1)\nabla \log \pi_{\boldsymbol{\theta}_1}(a|s) - \delta(O, \boldsymbol{\theta}_1)\nabla \log \pi_{\boldsymbol{\theta}_2}(a|s)\| \\
&\quad + \|\delta(O, \boldsymbol{\theta}_1)\nabla \log \pi_{\boldsymbol{\theta}_2}(a|s) - \delta(O, \boldsymbol{\theta}_2)\nabla \log \pi_{\boldsymbol{\theta}_2}(a|s)\| \\
&\leq U_\delta L_l \|\boldsymbol{\theta}_1 - \boldsymbol{\theta}_2\| + B|\delta(O, \boldsymbol{\theta}_1) - \delta(O, \boldsymbol{\theta}_2)| \\
&\leq U_\delta L_l \|\boldsymbol{\theta}_1 - \boldsymbol{\theta}_2\| + B(|r(\boldsymbol{\theta}_1) - r(\boldsymbol{\theta}_2)| + \|\boldsymbol{\phi}(s') - \boldsymbol{\phi}(s)\| \cdot \|\boldsymbol{\omega}^*(\boldsymbol{\theta}_1) - \boldsymbol{\omega}^*(\boldsymbol{\theta}_2)\|) \\
&\leq (U_\delta L_l + 2BL_*)\|\boldsymbol{\theta}_1 - \boldsymbol{\theta}_2\| + B|\mathbb{E}_{s \sim \mu_{\boldsymbol{\theta}_1}, a \sim \pi_{\boldsymbol{\theta}_1}}[r(s, a)] - \mathbb{E}_{s \sim \mu_{\boldsymbol{\theta}_1}, a \sim \pi_{\boldsymbol{\theta}_2}}[r(s, a)]| \\
&\leq (U_\delta L_l + 2BL_*)\|\boldsymbol{\theta}_1 - \boldsymbol{\theta}_2\| + 2BU_r d_{TV}(\mu_{\boldsymbol{\theta}_1} \otimes \pi_{\boldsymbol{\theta}_1}, \mu_{\boldsymbol{\theta}_2} \otimes \pi_{\boldsymbol{\theta}_2}) \\
&\leq (U_\delta L_l + 2BL_* + 2BU_r|\mathcal{A}|L_\pi(1 + \lceil \log_\rho m^{-1}\rceil + \frac{1}{1-\rho}))\|\boldsymbol{\theta}_1 - \boldsymbol{\theta}_2\|.
\end{aligned}$$

Hence we have $h(O, \boldsymbol{\theta})$ is $L_h$-Lipschitz, where $L_h$ denotes the above coefficient.

For term $\mathbb{E}_{\boldsymbol{\theta}}[h(O', \boldsymbol{\theta})]$, we have
$$\begin{aligned}
&\|\mathbb{E}_{\boldsymbol{\theta}_1}[h(O', \boldsymbol{\theta}_1)] - \mathbb{E}_{\boldsymbol{\theta}_2}[h(O', \boldsymbol{\theta}_2)]\| \\
&\leq \|\mathbb{E}_{\boldsymbol{\theta}_1}[h(O', \boldsymbol{\theta}_1)] - \mathbb{E}_{\boldsymbol{\theta}_1}[h(O', \boldsymbol{\theta}_2)]\| + \|\mathbb{E}_{\boldsymbol{\theta}_1}[h(O', \boldsymbol{\theta}_2)] - \mathbb{E}_{\boldsymbol{\theta}_2}[h(O', \boldsymbol{\theta}_2)]\| \\
&\leq \mathbb{E}_{\boldsymbol{\theta}_1}[\|h(O', \boldsymbol{\theta}_1) - h(O', \boldsymbol{\theta}_2)\|] + \|\mathbb{E}_{\boldsymbol{\theta}_1}[h(O', \boldsymbol{\theta}_2)] - \mathbb{E}_{\boldsymbol{\theta}_2}[h(O', \boldsymbol{\theta}_2)]\| \\
&\leq L_h\|\boldsymbol{\theta}_1 - \boldsymbol{\theta}_2\| + \|\mathbb{E}_{\boldsymbol{\theta}_1}[h(O', \boldsymbol{\theta}_2)] - \mathbb{E}_{\boldsymbol{\theta}_2}[h(O', \boldsymbol{\theta}_2)]\| \\
&\leq L_h\|\boldsymbol{\theta}_1 - \boldsymbol{\theta}_2\| + 2BU_r d_{TV}(\mu_{\boldsymbol{\theta}_1} \otimes \pi_{\boldsymbol{\theta}_1}, \mu_{\boldsymbol{\theta}_2} \otimes \pi_{\boldsymbol{\theta}_2}) \\
&\leq [L_h + 2BU_r|\mathcal{A}|L_\pi(1 + \lceil \log_\rho m^{-1}\rceil + \frac{1}{1-\rho})]\|\boldsymbol{\theta}_1 - \boldsymbol{\theta}_2\| \\
&\leq 2L_h\|\boldsymbol{\theta}_1 - \boldsymbol{\theta}_2\|.
\end{aligned}$$

Then we have $\boldsymbol{\omega}_t - \boldsymbol{\omega}_t^*$ is $U_\delta$-bounded and $L_*$-Lipschitz; $h(O, \boldsymbol{\theta}) - \mathbb{E}_{\boldsymbol{\theta}}[h(O', \boldsymbol{\theta})]$ is $3L_h$-Lipschitz and $2U_\delta B$-bounded. By the triangle inequality, we have
$$\|\Xi(O_t, \boldsymbol{\omega}_t, \boldsymbol{\theta}_t) - \Xi(O_t, \boldsymbol{\omega}_t, \boldsymbol{\theta}_{t-\tau})\| \leq (3U_\delta L_h + 2U_\delta BL_*)\|\boldsymbol{\theta}_t - \boldsymbol{\theta}_{t-\tau}\| \leq C_2\|\boldsymbol{\theta}_t - \boldsymbol{\theta}_{t-\tau}\|,$$
where $C_2 := 3BU_\delta^2|\mathcal{A}|L_\pi(1 + \lceil \log_\rho m^{-1}\rceil + \frac{1}{1-\rho}) + 3U_\delta^2 L_l + 8U_\delta BL_*$.

**Step 2:** show that
$$\|\Xi(O, \boldsymbol{\omega}_t, \boldsymbol{\theta}) - \Xi(O, \boldsymbol{\omega}_{t-\tau}, \boldsymbol{\theta})\| \leq 2U_\delta B\|\boldsymbol{\omega}_t - \boldsymbol{\omega}_{t-\tau}\|. \tag{40}$$

Actually, we have
$$\|\Xi(O, \boldsymbol{\omega}_1, \boldsymbol{\theta}) - \Xi(O, \boldsymbol{\omega}_2, \boldsymbol{\theta})\| = \|\langle \boldsymbol{\omega}_1 - \boldsymbol{\omega}_2, \mathbb{E}_{O'}[h(O', \boldsymbol{\theta})] - h(O, \boldsymbol{\theta})\rangle\| \leq 2U_\delta B\|\boldsymbol{\omega}_1 - \boldsymbol{\omega}_2\|$$

**Step 3:** show that for tuples $O_t = (s_t, a_t, s_{t+1})$ and $\widetilde{O}_t = (\widetilde{s}_t, \widetilde{a}_t, \widetilde{s}_{t+1})$. Conditioning on $s_{t-\tau+1}$ and $\boldsymbol{\theta}_{t-\tau}$, we have

$$\mathbb{E}[\Xi(O_t, \boldsymbol{\omega}_{t-\tau}, \boldsymbol{\theta}_{t-\tau}) - \Xi(\widetilde{O}_t, \boldsymbol{\omega}_{t-\tau}, \boldsymbol{\theta}_{t-\tau})] \leq 2U_\delta^2 B|\mathcal{A}|L_\pi \sum_{k=t-\tau}^t \mathbb{E}\|\boldsymbol{\theta}_k - \boldsymbol{\theta}_{t-\tau}\|. \tag{41}$$

By definition of $\Xi(O, \boldsymbol{\omega}, \boldsymbol{\theta})$, we have
$$\begin{aligned}
&\|\mathbb{E}[\Xi(O_t, \boldsymbol{\omega}_{t-\tau}, \boldsymbol{\theta}_{t-\tau}) - \Xi(\widetilde{O}_t, \boldsymbol{\omega}_{t-\tau}, \boldsymbol{\theta}_{t-\tau})]\| \\
&= \|\mathbb{E}[\langle \boldsymbol{\omega}_{t-\tau} - \boldsymbol{\omega}_{t-\tau}^*, h(\widetilde{O}_t, \boldsymbol{\theta}_{t-\tau}) - h(O_t, \boldsymbol{\theta}_{t-\tau})]\| \\
&= \|\mathbb{E}[\langle \boldsymbol{\omega}_{t-\tau} - \boldsymbol{\omega}_{t-\tau}^*, h(\widetilde{O}_t, \boldsymbol{\theta}_{t-\tau})\rangle - \mathbb{E}[\langle \boldsymbol{\omega}_{t-\tau} - \boldsymbol{\omega}_{t-\tau}^*, h(O_t, \boldsymbol{\theta}_{t-\tau})\rangle]\| \\
&\leq 4U_\delta^2 B d_{TV}(\mathbb{P}(O_t \in \cdot | s_{t-\tau+1}, \boldsymbol{\theta}_{t-\tau}), \mathbb{P}(\widetilde{O}_t \in \cdot | s_{t-\tau+1}, \boldsymbol{\theta}_{t-\tau})), \tag{42}
\end{aligned}$$

where the inequality comes from the definition of total variation distance. The total variation norm between $O_t$ and $\widetilde{O}_t$ has been computed in (33). Plugging (33) into (42), we get

$$\|\mathbb{E}[\Xi(O_t, \boldsymbol{\omega}_{t-\tau}, \boldsymbol{\theta}_{t-\tau}) - \Xi(\widetilde{O}_t, \boldsymbol{\omega}_{t-\tau}, \boldsymbol{\theta}_{t-\tau})]\| \leq 2U_\delta^2 B|\mathcal{A}|L_\pi \sum_{k=t-\tau}^{t} \mathbb{E}\|\boldsymbol{\theta}_k - \boldsymbol{\theta}_{t-\tau}\|$$

$$\leq 2U_\delta^2 B|\mathcal{A}|L_\pi G\tau(\tau+1)\alpha_{t-\tau}.$$

**Step 4:** Show that conditioning on $s_{t-\tau+1}$ and $\boldsymbol{\theta}_{t-\tau}$, we have

$$\|\mathbb{E}[\Xi(\widetilde{O}_t, \boldsymbol{\omega}_{t-\tau}, \boldsymbol{\theta}_{t-\tau})]\| \leq 4U_\delta^2 Bm\rho^{\tau-1}. \tag{43}$$

It can be shown that

$$\|\mathbb{E}[\Xi(\widetilde{O}_t, \boldsymbol{\omega}_{t-\tau}, \boldsymbol{\theta}_{t-\tau})]\| \overset{(1)}{=} \|\mathbb{E}[\Xi(\widetilde{O}_t, \boldsymbol{\omega}_{t-\tau}, \boldsymbol{\theta}_{t-\tau}) - \Xi(O'_{t-\tau}, \boldsymbol{\omega}_{t-\tau}, \boldsymbol{\theta}_{t-\tau})]\|$$

$$= \|\mathbb{E}[\langle \boldsymbol{\omega}_{t-\tau} - \boldsymbol{\omega}^*_{t-\tau}, h(O'_{t-\tau}, \boldsymbol{\theta}_{t-\tau})\rangle - \langle \boldsymbol{\omega}_{t-\tau} - \boldsymbol{\omega}^*_{t-\tau}, h(\widetilde{O}_t, \boldsymbol{\theta}_{t-\tau})\rangle]\|$$

$$\overset{(2)}{\leq} 4U_\delta^2 Bd_{TV}(\mathbb{P}(\widetilde{O}_t \in \cdot|s_{t-\tau+1}, \boldsymbol{\theta}_{t-\tau}), \mu_{\boldsymbol{\theta}_{t-\tau}} \otimes \pi_{\boldsymbol{\theta}_{t-\tau}} \otimes \mathcal{P}),$$

where (1) is due to the fact that $O'_t$ is from the stationary distribution which satisfies $\mathbb{E}[\Xi(O'_t, \boldsymbol{\omega}_{t-\tau}, \boldsymbol{\theta}_{t-\tau})] = 0$ and (2) follows from the definition of total variation distance. From Assumption 3.2, we know that

$$d_{TV}(\mathbb{P}(\widetilde{s}_t \in \cdot), \mu_{\boldsymbol{\theta}_{t-\tau}}) \leq m\rho^{\tau-1}.$$

We also have the fact that

$$\mathbb{P}(\widetilde{O}_t \in \cdot|s_{t-\tau+1}, \boldsymbol{\theta}_{t-\tau}) = \mathbb{P}(\widetilde{s}_t \in \cdot|s_{t-\tau+1}, \boldsymbol{\theta}_{t-\tau}) \otimes \pi_{\boldsymbol{\theta}_{t-\tau}} \otimes \mathcal{P}.$$

Therefore, we have

$$\|\mathbb{E}[\Xi(\widetilde{O}_t, \boldsymbol{\omega}_{t-\tau}, \boldsymbol{\theta}_{t-\tau})\| \leq 4U_\delta^2 Bm\rho^{\tau-1}.$$

Combining (39)-(43), we can decompose the Markovian bias as

$$\mathbb{E}[\Xi(O_t, \boldsymbol{\omega}_t, \boldsymbol{\theta}_t)] = \mathbb{E}[\Xi(O_t, \boldsymbol{\omega}_t, \boldsymbol{\theta}_t) - \Xi(O_t, \boldsymbol{\omega}_t, \boldsymbol{\theta}_{t-\tau})]$$

$$+ \mathbb{E}[\Xi(O_t, \boldsymbol{\omega}_t, \boldsymbol{\theta}_{t-\tau}) - \Xi(O_t, \boldsymbol{\omega}_{t-\tau}, \boldsymbol{\theta}_{t-\tau})]$$

$$+ \mathbb{E}[\Xi(O_t, \boldsymbol{\omega}_{t-\tau}, \boldsymbol{\theta}_{t-\tau}) - \Xi(\widetilde{O}_t, \boldsymbol{\omega}_{t-\tau}, \boldsymbol{\theta}_{t-\tau})]$$

$$+ \mathbb{E}[\Xi(\widetilde{O}_t, \boldsymbol{\omega}_{t-\tau}, \boldsymbol{\theta}_{t-\tau})]$$

$$\leq C_2\|\boldsymbol{\theta}_t - \boldsymbol{\theta}_{t-\tau}\| + 2U_\delta B\|\boldsymbol{\omega}_t - \boldsymbol{\omega}_{t-\tau}\|$$

$$+ 2U_\delta^2 B|\mathcal{A}|L_\pi G\tau(\tau+1)\alpha_{t-\tau} + 4U_\delta^2 Bm\rho^{\tau-1}.$$

Thus we conclude our proof. $\square$

**Proof of Lemma C.6.**

*Proof.* We will divide the proof of this lemma into three steps.

**Step 1:** show that

$$|\Theta(O_t, \boldsymbol{\theta}_{t-\tau}) - \Theta(\widetilde{O}_t, \boldsymbol{\theta}_{t-\tau})| \leq (2U_\delta BL_{J'} + 3L_J L_h)\|\boldsymbol{\theta}_t - \boldsymbol{\theta}_{t-\tau}\|, \tag{44}$$

where $L_h = U_\delta L_l + (2 + 2\lambda^{-2} + 3\lambda^{-1})BU_r|\mathcal{A}|L_\pi(1 + \lceil \log_\rho m^{-1}\rceil + 1/(1-\rho))$.

Since $\Theta(O, \boldsymbol{\theta}) = \langle \nabla J(\boldsymbol{\theta}), \mathbb{E}_{\boldsymbol{\theta}}[h(O', \boldsymbol{\theta})] - h(O, \boldsymbol{\theta})\rangle$, we will show that each term in $\Theta(O, \boldsymbol{\theta})$ is Lipschitz.

For the term $\nabla J(\boldsymbol{\theta})$, by Lemma B.3 we know it's $L_{J'}$-Lipschitz.

For term $h(O, \boldsymbol{\theta})$, denote $\delta(O, \boldsymbol{\theta}) := r(s,a) - r(\boldsymbol{\theta}) + (\boldsymbol{\phi}(s') - \boldsymbol{\phi}(s))^\top \boldsymbol{\omega}^*$, we have

$$
\begin{aligned}
&\|h(O, \boldsymbol{\theta}_1) - h(O, \boldsymbol{\theta}_2)\| \\
&= \|\delta(O, \boldsymbol{\theta}_1)\nabla \log \pi_{\boldsymbol{\theta}_1}(a|s) - \delta(O_t, \boldsymbol{\theta}_2)\nabla \log \pi_{\boldsymbol{\theta}_2}(a|s)\| \\
&\le \|\delta(O, \boldsymbol{\theta}_1)\nabla \log \pi_{\boldsymbol{\theta}_1}(a|s) - \delta(O, \boldsymbol{\theta}_1)\nabla \log \pi_{\boldsymbol{\theta}_2}(a|s)\| \\
&\quad + \|\delta(O, \boldsymbol{\theta}_1)\nabla \log \pi_{\boldsymbol{\theta}_2}(a|s) - \delta(O, \boldsymbol{\theta}_2)\nabla \log \pi_{\boldsymbol{\theta}_2}(a|s)\| \\
&\le U_\delta L_l \|\boldsymbol{\theta}_1 - \boldsymbol{\theta}_2\| + B|\delta(O, \boldsymbol{\theta}_1) - \delta(O, \boldsymbol{\theta}_2)| \\
&\le U_\delta L_l \|\boldsymbol{\theta}_1 - \boldsymbol{\theta}_2\| + B(|r(\boldsymbol{\theta}_1) - r(\boldsymbol{\theta}_2)| + \|\boldsymbol{\phi}(s') - \boldsymbol{\phi}(s)\| \cdot \|\boldsymbol{\omega}^*(\boldsymbol{\theta}_1) - \boldsymbol{\omega}^*(\boldsymbol{\theta}_2)\|) \\
&\le (U_\delta L_l + 2BL_*)\|\boldsymbol{\theta}_1 - \boldsymbol{\theta}_2\| + B|\mathbb{E}_{s \sim \mu_{\boldsymbol{\theta}_1}, a \sim \pi_{\boldsymbol{\theta}_1}}[r(s,a)] - \mathbb{E}_{s \sim \mu_{\boldsymbol{\theta}_1}, a \sim \pi_{\boldsymbol{\theta}_2}}[r(s,a)]| \\
&\le (U_\delta L_l + 2BL_*)\|\boldsymbol{\theta}_1 - \boldsymbol{\theta}_2\| + 2BU_r d_{TV}(\mu_{\boldsymbol{\theta}_1} \otimes \pi_{\boldsymbol{\theta}_1}, \mu_{\boldsymbol{\theta}_2} \otimes \pi_{\boldsymbol{\theta}_2}) \\
&\le (U_\delta L_l + 2BL_* + 2BU_r|\mathcal{A}|L_\pi(1 + \lceil \log_\rho m^{-1} \rceil + \frac{1}{1-\rho}))\|\boldsymbol{\theta}_1 - \boldsymbol{\theta}_2\|.
\end{aligned}
$$

Hence we have $h(O, \boldsymbol{\theta})$ is $L_h$-Lipschitz, where $L_h$ denotes the above coefficient.

For term $\mathbb{E}_{\boldsymbol{\theta}}[h(O', \boldsymbol{\theta})]$, we have

$$
\begin{aligned}
&\|\mathbb{E}_{\boldsymbol{\theta}_1}[h(O', \boldsymbol{\theta}_1)] - \mathbb{E}_{\boldsymbol{\theta}_2}[h(O', \boldsymbol{\theta}_2)]\| \\
&\le \|\mathbb{E}_{\boldsymbol{\theta}_1}[h(O', \boldsymbol{\theta}_1)] - \mathbb{E}_{\boldsymbol{\theta}_1}[h(O', \boldsymbol{\theta}_2)]\| + \|\mathbb{E}_{\boldsymbol{\theta}_1}[h(O', \boldsymbol{\theta}_2)] - \mathbb{E}_{\boldsymbol{\theta}_2}[h(O', \boldsymbol{\theta}_2)]\| \\
&\le \mathbb{E}_{\boldsymbol{\theta}_1}[\|h(O', \boldsymbol{\theta}_1) - h(O', \boldsymbol{\theta}_2)\|] + \|\mathbb{E}_{\boldsymbol{\theta}_1}[h(O', \boldsymbol{\theta}_2)] - \mathbb{E}_{\boldsymbol{\theta}_2}[h(O', \boldsymbol{\theta}_2)]\| \\
&\le L_h\|\boldsymbol{\theta}_1 - \boldsymbol{\theta}_2\| + \|\mathbb{E}_{\boldsymbol{\theta}_1}[h(O', \boldsymbol{\theta}_2)] - \mathbb{E}_{\boldsymbol{\theta}_2}[h(O', \boldsymbol{\theta}_2)]\| \\
&\le L_h\|\boldsymbol{\theta}_1 - \boldsymbol{\theta}_2\| + 2BU_r d_{TV}(\mu_{\boldsymbol{\theta}_1} \otimes \pi_{\boldsymbol{\theta}_1}, \mu_{\boldsymbol{\theta}_2} \otimes \pi_{\boldsymbol{\theta}_2}) \\
&\le [L_h + 2BU_r|\mathcal{A}|L_\pi(1 + \lceil \log_\rho m^{-1} \rceil + \frac{1}{1-\rho})]\|\boldsymbol{\theta}_1 - \boldsymbol{\theta}_2\| \\
&\le 2L_h\|\boldsymbol{\theta}_1 - \boldsymbol{\theta}_2\|.
\end{aligned}
$$

Then we have $\nabla J(\boldsymbol{\theta})$ is $L_J$-bounded and $L_{J'}$-Lipschitz; $h(O, \boldsymbol{\theta}) - \mathbb{E}_{\boldsymbol{\theta}}[h(O', \boldsymbol{\theta})]$ is $3L_h$-Lipschitz and $2U_\delta B$-bounded. By the triangle inequality, we have

$$
\Theta(O_t, \boldsymbol{\theta}_t) - \Theta(O_t, \boldsymbol{\theta}_{t-\tau}) \le (2U_\delta BL_{J'} + 3L_J L_h)\|\boldsymbol{\theta}_t - \boldsymbol{\theta}_{t-\tau}\|
$$

**Step 2:** show that for $t \ge \tau > 0$, we have

$$
|\mathbb{E}[\Theta(O_t, \boldsymbol{\theta}_{t-\tau}) - \Theta(\widetilde{O}_t, \boldsymbol{\theta}_{t-\tau})]| \le 2U_\delta BL_J|\mathcal{A}|L_\pi \sum_{k=t-\tau}^{t} \|\boldsymbol{\theta}_k - \boldsymbol{\theta}_{t-\tau}\| \tag{45}
$$

By definition of $\Theta(O, \boldsymbol{\theta})$, we have

$$
\begin{aligned}
&|\mathbb{E}[\Theta(O_t, \boldsymbol{\theta}_{t-\tau}) - \Theta(\widetilde{O}_t, \boldsymbol{\theta}_{t-\tau})]| \\
&= |\mathbb{E}[\langle \nabla J(\boldsymbol{\theta}_{t-\tau}), h(\widetilde{O}_t, \boldsymbol{\theta}_{t-\tau}) - h(O_t, \boldsymbol{\theta}_{t-\tau})\rangle]| \\
&= |\mathbb{E}[\langle \nabla J(\boldsymbol{\theta}_{t-\tau}), h(\widetilde{O}_t, \boldsymbol{\theta}_{t-\tau})\rangle] - \mathbb{E}[\langle \nabla J(\boldsymbol{\theta}_{t-\tau}), h(O_t, \boldsymbol{\theta}_{t-\tau})\rangle]| \\
&\le 4U_\delta BL_J d_{TV}(\mathbb{P}(O_t \in \cdot|s_{t-\tau+1}, \boldsymbol{\theta}_{t-\tau}), \mathbb{P}(\widetilde{O}_t \in \cdot|s_{t-\tau+1}, \boldsymbol{\theta}_{t-\tau})), \tag{46}
\end{aligned}
$$

where the inequality comes from the definition of total variation distance. The total variation norm between $O_t$ and $\widetilde{O}_t$ has been computed in (33). Plugging (33) into (46), we get

$$
|\mathbb{E}[\Theta(O_t, \boldsymbol{\theta}_{t-\tau}) - \Theta(\widetilde{O}_t, \boldsymbol{\theta}_{t-\tau})]| \le 2U_\delta BL_J|\mathcal{A}|L_\pi \sum_{k=t-\tau}^{t} \|\boldsymbol{\theta}_k - \boldsymbol{\theta}_{t-\tau}\|.
$$

**Step 3:** show that for $t \ge \tau > 0$, we have

$$
|\mathbb{E}[\Theta(\widetilde{O}_t, \boldsymbol{\theta}_{t-\tau}) - \Theta(O'_t, \boldsymbol{\theta}_{t-\tau})]| \le 4U_\delta BL_J m\rho^{\tau-1}. \tag{47}
$$

From the definition of $\Theta(O, \boldsymbol{\theta})$, we have

$$
\begin{aligned}
|\mathbb{E}[\Theta(\widetilde{O}_t, \boldsymbol{\theta}_{t-\tau}) - \Theta(O'_t, \boldsymbol{\theta}_{t-\tau})]| &= |\mathbb{E}[\langle \nabla J(\boldsymbol{\theta}_{t-\tau}), h(O'_t, \boldsymbol{\theta}_{t-\tau})\rangle - \langle \nabla J(\boldsymbol{\theta}_{t-\tau}), h(\widetilde{O}_t, \boldsymbol{\theta}_{t-\tau})\rangle]| \\
&\le 4U_\delta BL_J d_{TV}(\mathbb{P}(\widetilde{O}_t \in \cdot|s_{t-\tau+1}, \boldsymbol{\theta}_{t-\tau}), \mu_{\boldsymbol{\theta}_{t-\tau}} \otimes \pi_{\boldsymbol{\theta}_{t-\tau}} \otimes \mathcal{P}).
\end{aligned}
$$

The inequality is due to the definition of total variation distance. From Assumption 3.2, we know that
$$d_{TV}(\mathbb{P}(\widetilde{s}_t \in \cdot), \mu_{\boldsymbol{\theta}_{t-\tau}}) \le m\rho^{\tau-1}.$$
We also have the fact that
$$\mathbb{P}(\widetilde{O}_t \in \cdot|s_{t-\tau+1}, \boldsymbol{\theta}_{t-\tau}) = \mathbb{P}(\widetilde{s}_t \in \cdot|s_{t-\tau+1}, \boldsymbol{\theta}_{t-\tau}) \otimes \pi_{\boldsymbol{\theta}_{t-\tau}} \otimes \mathcal{P}.$$
Therefore, we have
$$|\mathbb{E}[\Theta(\widetilde{O}_t, \boldsymbol{\theta}_{t-\tau} - \Theta(O'_t, \boldsymbol{\theta}_{t-\tau})]| \le 4U_\delta BL_J m\rho^{\tau-1}.$$
Combining (44), (45), and (47), we can decompose the Markovian bias as
$$\mathbb{E}[\Theta(O_t, \boldsymbol{\theta}_t)] = \mathbb{E}[\Theta(O_t, \boldsymbol{\theta}_t) - \Theta(O_t, \boldsymbol{\theta}_{t-\tau})]$$
$$+ \mathbb{E}[\Theta(O_t, \boldsymbol{\theta}_{t-\tau}) - \Theta(\widetilde{O}_t, \boldsymbol{\theta}_{t-\tau})]$$
$$+ \mathbb{E}[\Theta(\widetilde{O}_t, \boldsymbol{\theta}_{t-\tau}) - \Theta(O'_t, \boldsymbol{\theta}_{t-\tau})]$$
$$+ \mathbb{E}[\Theta(O'_t, \boldsymbol{\theta}_{t-\tau})],$$
where $\widetilde{O}_t$ is from the auxiliary Markovian chain defined in (10) and $O'_t$ is from the stationary distribution which satisfies $\mathbb{E}[\Theta(O'_t, \boldsymbol{\theta}_{t-\tau})] = 0$.

Then we have
$$\mathbb{E}[\Theta(O_t, \boldsymbol{\theta}_t)] \le (2U_\delta BL_{J'} + 3L_J L_h)\mathbb{E}\|\boldsymbol{\theta}_t - \boldsymbol{\theta}_{t-\tau}\|$$
$$+ 2U_\delta BL_J|\mathcal{A}|L_\pi \sum_{k=t-\tau}^{t} \|\boldsymbol{\theta}_k - \boldsymbol{\theta}_{t-\tau}\| + 4U_\delta BL_J m\rho^{\tau-1}$$
$$\le (2U_\delta BL_{J'} + 3L_J L_h) \sum_{k=t-\tau+1}^{t} \mathbb{E}\|\boldsymbol{\theta}_k - \boldsymbol{\theta}_{k-1}\|$$
$$+ 2U_\delta BL_J|\mathcal{A}|L_\pi \sum_{k=t-\tau+1}^{t} \sum_{j=t-\tau+1}^{k} \mathbb{E}\|\boldsymbol{\theta}_j - \boldsymbol{\theta}_{j-1}\| + 4U_\delta BL_J m\rho^{\tau-1}$$
$$\le (2U_\delta BL_{J'} + 3L_J L_h) \sum_{k=t-\tau+1}^{t} \mathbb{E}\|\boldsymbol{\theta}_k - \boldsymbol{\theta}_{k-1}\|$$
$$+ 2U_\delta BL_J|\mathcal{A}|L_\pi \tau \sum_{j=t-\tau+1}^{t} \mathbb{E}\|\boldsymbol{\theta}_j - \boldsymbol{\theta}_{j-1}\| + 4U_\delta BL_J m\rho^{\tau-1}$$
$$\le D_1(\tau+1) \sum_{k=t-\tau+1}^{t} \mathbb{E}\|\boldsymbol{\theta}_k - \boldsymbol{\theta}_{k-1}\| + D_2 m\rho^{\tau-1}$$
$$\le D_1(\tau+1)^2 G\alpha + D_2 m\rho^{\tau-1}$$
where $D_1 = \max\{U_\delta BL_{J'} + 3L_J L_h, 2U_\delta BL_J|\mathcal{A}|L_\pi\}$ and $D_2 = 4U_\delta BL_J$. Thus we conclude the proof. $\qquad\square$

# E  IID Sampling Analysis

---
**Algorithm 2** Single-timescale Actor-Critic (i.i.d. sampling)
---
1: **Input** initial actor parameter $\boldsymbol{\theta}_0$, initial critic parameter $\boldsymbol{\omega}_0$, initial reward estimator $\eta_0$, stepsize $\alpha_t$ for actor, $\beta_t$ for critic and $\gamma_t$ for reward estimator.
2: **for** $t = 0, 1, 2, \cdots, T-1$ **do**
3:     Sample $s_t \sim \mu_{\boldsymbol{\theta}_t}$
4:     Take the action $a_t \sim \pi_{\boldsymbol{\theta}_t}(\cdot|s_t)$
5:     Observe next state $s'_t \sim \mathcal{P}(\cdot|s_t, a_t)$ and the reward $r_t = r(s_t, a_t)$
6:     $\delta_t = r_t - \eta_t + \boldsymbol{\phi}(s'_t)^\top \boldsymbol{\omega}_t - \boldsymbol{\phi}(s_t)^\top \boldsymbol{\omega}_t$
7:     $\eta_{t+1} = \eta_t + \gamma_t(r_t - \eta_t)$
8:     $\boldsymbol{\omega}_{t+1} = \Pi_{U_{\boldsymbol{\omega}}}(\boldsymbol{\omega}_t + \beta_t \delta_t \boldsymbol{\phi}(s_t))$
9:     $\boldsymbol{\theta}_{t+1} = \boldsymbol{\theta}_t + \alpha_t \delta_t \nabla_{\boldsymbol{\theta}} \log \pi_{\boldsymbol{\theta}_t}(a_t|s_t)$
10: **end for**
---

Note that under i.i.d. sampling in Algorithm 2, we denote by $s_t$ the samples from the stationary distribution and $s_t'$ the subsequent state following transition kernel $s_t' \sim \mathcal{P}(\cdot|s_t, a_t)$. Correspondingly, we redefine the observation tuple as $O_t = (s_t, a_t, s_t')$ (in contrast to $O_t = (s_t, a_t, s_{t+1})$ in the Markovian sampling case). This modification implies the decoupling of $O_t$ and $O_{t+1}$ since $s_{t+1}$ in tuple $O_{t+1}$ is a new state sampled from the stationary distribution rather than inherited from $O_t$. This intuitively elucidates the vanishment of Markovian noise under i.i.d. sampling.

**Lemma E.1.** *Under i.i.d sampling, we have*

$$\mathbb{E}[\Phi(O_t, \eta_t, \boldsymbol{\theta}_t)] = 0,$$
$$\mathbb{E}[\Psi(O_t, \boldsymbol{\omega}_t, \boldsymbol{\theta}_t)] = 0,$$
$$\mathbb{E}[\Theta(O_t, O_t', \boldsymbol{\theta}_t)] = 0,$$
$$\mathbb{E}[\Xi(O_t, \boldsymbol{\omega}_t, \boldsymbol{\theta}_t)] = 0.$$

*Proof.* Note that the expectation is taken over all the random variables. We use the notation $O_t$ to denote the tuple $(s_t, a_t, s_t')$ and $v_{0:t}$ to denote the sequence $(s_t, a_t, s_t'), (s_t, a_t, s_t'), \cdots, (s_t, a_t, s_t')$. By definition in (13), it can be shown that

$$\mathbb{E}[\Phi(O_t, \eta_t, \boldsymbol{\theta}_t)] = \mathbb{E}_{v_{0:t}}[\Phi(O_t, \eta_t, \boldsymbol{\theta}_t)]$$
$$= \mathbb{E}_{v_{0:t-1}} \mathbb{E}_{v_{0:t}}[(\eta_t - J(\boldsymbol{\theta}_t))(r_t - J(\boldsymbol{\theta}_t))|v_{0:t-1}],$$

where is second equality is due to law of total expectation. Once we know $v_{0:t-1}$, $\eta_t$ and $J(\boldsymbol{\theta}_t)$ is not a random variable any more. It holds that

$$\mathbb{E}[\Phi(O_t, \eta_t, \boldsymbol{\theta}_t)] = \mathbb{E}_{v_{0:t-1}} \mathbb{E}_{v_{0:t}}[(\eta_t - J(\boldsymbol{\theta}_t))(r_t - J(\boldsymbol{\theta}_t))|v_{0:t-1}]$$
$$= \mathbb{E}_{v_{0:t-1}}(\eta_t - J(\boldsymbol{\theta}_t))\mathbb{E}_{v_{0:t}}[(r_t - J(\boldsymbol{\theta}_t))|v_{0:t-1}]$$
$$= \mathbb{E}_{v_{0:t-1}}(\eta_t - J(\boldsymbol{\theta}_t))\mathbb{E}_{O_t}[(r_t - J(\boldsymbol{\theta}_t))|v_{0:t-1}]$$
$$= 0,$$

where the last equation is due to $\mathbb{E}_{O_t}[(r_t - J(\boldsymbol{\theta}_t))|v_{0:t-1}] = 0$ under i.i.d. sampling.

By a similar argument, we have

$$\mathbb{E}[\Psi(O_t, \eta_t, \boldsymbol{\theta}_t)] = \mathbb{E}_{v_{0:t}}[\langle \boldsymbol{\omega}_t - \boldsymbol{\omega}_t^*, g(O, \boldsymbol{\omega}, \boldsymbol{\theta}) - \bar{g}(\boldsymbol{\omega}_t, \boldsymbol{\theta}_t)\rangle]$$
$$= \mathbb{E}_{v_{0:t-1}} \mathbb{E}_{v_{0:t}}[\langle \boldsymbol{\omega}_t - \boldsymbol{\omega}_t^*, g(O_t, \boldsymbol{\omega}_t, \boldsymbol{\theta}_t) - \bar{g}(\boldsymbol{\omega}_t, \boldsymbol{\theta}_t)\rangle|v_{0:t-1}]$$
$$= \mathbb{E}_{v_{0:t-1}} \langle \boldsymbol{\omega}_t - \boldsymbol{\omega}_t^*, \mathbb{E}_{v_{0:t}}[g(O_t, \boldsymbol{\omega}_t, \boldsymbol{\theta}_t) - \bar{g}(\boldsymbol{\omega}_t, \boldsymbol{\theta}_t)]\rangle|v_{0:t-1}]$$
$$= \mathbb{E}_{v_{0:t-1}} \langle \boldsymbol{\omega}_t - \boldsymbol{\omega}_t^*, \mathbb{E}_{O_t}[g(O_t, \boldsymbol{\omega}_t, \boldsymbol{\theta}_t) - \bar{g}(\boldsymbol{\omega}_t, \boldsymbol{\theta}_t)]\rangle|v_{0:t-1}]$$
$$= 0,$$

where we use the fact that $\mathbb{E}_{O_t}[g(O_t, \boldsymbol{\omega}_t, \boldsymbol{\theta}_t) - \bar{g}(\boldsymbol{\omega}_t, \boldsymbol{\theta}_t)\rangle|v_{0:t-1}] = 0$.

Similarly, we have

$$\mathbb{E}[\Theta(O_t, O_t', \boldsymbol{\theta}_t)] = \mathbb{E}_{v_{0:t}}[\langle \nabla J(\boldsymbol{\theta}_t), \mathbb{E}_{O_t'}[h(O_t', \boldsymbol{\theta}_t)] - h(O_t, \boldsymbol{\theta}_t)\rangle]$$
$$= \mathbb{E}_{v_{0:t-1}} \mathbb{E}_{v_{0:t}}[\langle \nabla J(\boldsymbol{\theta}_t), \mathbb{E}_{O_t'}[h(O_t', \boldsymbol{\theta}_t)] - h(O_t, \boldsymbol{\theta}_t)\rangle|v_{0:t-1}]$$
$$= \mathbb{E}_{v_{0:t-1}} \langle \nabla J(\boldsymbol{\theta}_t), \mathbb{E}_{v_{0:t}}[\mathbb{E}_{O_t'}[h(O_t', \boldsymbol{\theta}_t)] - h(O_t, \boldsymbol{\theta}_t)]\rangle|v_{0:t-1}]$$
$$= \mathbb{E}_{v_{0:t-1}} \langle \nabla J(\boldsymbol{\theta}_t), \mathbb{E}_{O_t}[\mathbb{E}_{O_t'}[h(O_t', \boldsymbol{\theta}_t)] - h(O_t, \boldsymbol{\theta}_t)]\rangle|v_{0:t-1}]$$
$$= 0,$$

where we use fact that $O_t = O_t'$ under i.i.d. sampling.

The proof of $\mathbb{E}[\Xi(O_t, \boldsymbol{\omega}_t, \boldsymbol{\theta}_t)] = 0$ is the same as the above argument, which concludes the proof. □

**Proof of Theorem 3.6.**

*Proof.* The proof follows similarly to the Markovian sampling case. Specifically, all the Markovian noises (see the definitions in (13)) present in the former analysis reduce to zero after taking expectations. The detailed results and proof are presented in Lemma E.1. Then, replacing Lemma C.1, Lemma C.3, and Lemma C.6 with Lemma E.1, we will get the desired $\mathcal{O}(T^{-\frac{1}{2}})$ convergence rate and thus an $\mathcal{O}(\epsilon^{-2})$ sample complexity accordingly. □

