# OpenReview forum: "Finite-Time Analysis of Single-Timescale Actor-Critic"
_NeurIPS.cc/2023/Conference — NeurIPS 2023 poster_

### Official Review · Reviewer_JDcB · 2023-06-11

**Soundness:** 3 good
**Presentation:** 3 good
**Contribution:** 3 good
**Rating:** 6
**Confidence:** 4

**Summary:**

The paper studies finding the optimal policy in an infinite-horizon average-reward MDP with an online, sample-based algorithm. The state-of-the-art algorithm in this setting uses two different timescales and is known to have a sample complexity of $\widetilde{\mathcal{O}}(\epsilon^{-2.5})$. This paper shows that a single-timescale version of the actor-critic algorithm enjoys an improved complexity of $\widetilde{\mathcal{O}}(\epsilon^{-2})$, up to a logarithm factor. If we were always able to sample according to the stationary distribution under the current policy, the logarithm factor can be removed.

**Strengths:**

Being able to improve the convergence rate from $\widetilde{\mathcal{O}}(\epsilon^{-2.5})$ to $\widetilde{\mathcal{O}}(\epsilon^{-2})$ is a pretty significant contribution. As the authors noted, this rate already matches the rate of standard SGD for non-convex functions. The paper is also well-written.

**Weaknesses:**

While the presentation of the paper is mostly clear, more discussion how exactly the authors improved the convergence rate and why previous works failed to do that is needed for the audience to appreciate the technical contribution. In fact, many existing works on two-timescale AC do not start off trying to the make updates two-timescale. Making the step sizes for the actor and critic decay at the same rate is allowed, but doing so hurts their convergence rates. Do the authors take advantage of any other structure of the MDP beyond general non-convexity to make a single timescale more favorable?

**Questions:**

- Related work section is somewhat limited. The connection of the paper to existing works on single-loop AC algorithms including the following ones should be discussed.

Hong, M., Wai, H.T., Wang, Z. and Yang, Z., 2023. A two-timescale stochastic algorithm framework for bilevel optimization: Complexity analysis and application to actor-critic. SIAM Journal on Optimization, 33(1), pp.147-180.

Zeng, S., Doan, T.T. and Romberg, J., 2021. A two-time-scale stochastic optimization framework with applications in control and reinforcement learning. arXiv preprint arXiv:2109.14756.

Khodadadian, S., Doan, T.T., Romberg, J. and Maguluri, S.T., 2022. Finite sample analysis of two-time-scale natural actor-critic algorithm. IEEE Transactions on Automatic Control.

Zhou, M. and Lu, J., 2022. Single Time-scale Actor-critic Method to Solve the Linear Quadratic Regulator with Convergence Guarantees. arXiv preprint arXiv:2202.00048.


- I do not feel that the discussion of i.i.d. sampling case and the theorem adds much to the paper. The claimed contribution on Markovian sampling also seems a bit oversold. Many existing works have shown how Markovian samples can be handled at the cost of introducing a log factor, which does not appear if the samples are i.i.d.

---

> ### Author Rebuttal · Authors · 2023-08-09
>
> # Thanks for reviewing our paper!
> **(W1: On general MDP)** Thanks for your comments. In this paper, we only consider the general MDP with a non-convex objective function. The single-timescale approach is superior to the two-timescale approach because the latter updates the actor slower than the critic. Therefore, there is a delay in learning and inefficient usage of data in the two-timescale approach. Considering other structures of MDP to validate the superiority of single-timescale actor-critic would be interesting future work.
>
> **(Q1: On related reference)** Thanks for sharing the relevant references. We will discuss them adequately.
>
> **(Q2: On the discussion of i.i.d. sampling)** Thanks for your comments. We agree that the discussion of the i.i.d. sampling case does not add much to the paper. The i.i.d. case is included for comparison to show that the log factor is caused by the Markovian samples.

---

> > ### Comment · Reviewer_JDcB · 2023-08-10
> >
> > I would like to thank the authors for the response. My most important question is, what exactly is the improvement in analysis made by the authors that enhanced the convergence rate and why prior works failed to do that. As I commented above, many existing works on two-timescale AC do not start off trying to the make updates two-timescale. Making the step sizes for the actor and critic decay at the same rate is allowed, but doing so hurts their convergence rates.

---

> > > ### Author Response · Authors · 2023-08-11
> > >
> > > Thanks for your comment. We are afraid that there is a misunderstanding of the implication of the stepsizes' timescale. For single-timescale algorithms, the step sizes for the actor and critic decay at the same rate, hence they update at approximately the same speed. For two-timescale algorithms, the actor stepsize decays faster than critic stepsize, hence actor updates much slower than critic asymptotically. Due to this artificial slowing down of the **actor** update in the two-timescale approaches, single-timescale approaches typically converge faster than two-timescale approaches [Oleshevsky45\& Gharesifard, 2023, Paragraph 5 of Introduction]. As we also mentioned in our initial comments, "there is a delay in learning and an inefficient usage of data in the two-timescale approach", because the actor updates with a small stepsize at each sample. Many existing works consider the two-timescale approach because it's easier to analyze compared with the single-timescale approach, but not because it converges fast. Hope this clarifies our response.

---

### Official Review · Reviewer_2T31 · 2023-07-06

**Soundness:** 3 good
**Presentation:** 3 good
**Contribution:** 3 good
**Rating:** 6
**Confidence:** 3

**Summary:**

The work studies the actor-critic algorithm’s convergence under the single-timescale update where the step-size of actor and critic variables are only proportional by a  constant.  Authors suggest the epsilon-optimal solution with a sample complexity $\tilde{O}(\epsilon^{-2})$  under standard assumptions and $O(\epsilon^{-2})$ under i.i.d. sampling.

**Strengths:**

This paper tries to analyze the sample complexity of the AC method under more practical Markovian sampling, where the transition tuples are generated from a single trajectory. The theoretical problem is well-motivated.

The paper proposes a new analysis framework for the algorithm and establishes the sample complexity of  $O(\epsilon^{-2})$ which matches the best existing sample complexity for the single-timescale AC algorithm, and a sample complexity of  $\tilde{O}(\epsilon^{-2})$ without i.i.d. assumption.

**Weaknesses:**

After removing the assumption of independence and identically distributed (i.i.d.) data, the author introduces a new assumption, namely that the Markov chain is geometrically mixing and the rollout state distribution rapidly approximates the stationary distribution (assumption 3.2). I am curious whether this assumption, although slightly weaker than the i.i.d. assumption, can still be considered strong, as it may not be commonly applicable in practical scenarios. Furthermore, this assumption appears to conflict with the statement "the transition tuples are generated from a single trajectory."

In the final results, the author requires a condition of $T>2\tau$ to ensure that the obtained samples are sufficiently close to samples from an i.i.d. stationary distribution.

Numerous studies have already examined actor-critic methods and achieved the same sample complexity of $O(\epsilon^{-2})$. The author claimed that their proposed methods can be utilized without relying on the i.i.d. assumption and achieve a slightly inferior complexity of $\tilde{O}(\epsilon^{-2})$ under alternative assumptions. It is hard to judge the contribution in this situation since it is natural to obtain a worse result under milder assumptions.

**Questions:**

The author asserts that their methods are capable of addressing continuous state situations, even when the state space is infinite. However, it is unclear whether the analyses can also be extended to infinite action spaces. Could the author provide some clarification and analyses on this aspect?

Since the paper obtain the same results of  $O(\epsilon^{-2})$  with previous works, could the author discuss and compare their results more specifically? (e.g., the constant comparison or simple control experiments)

**Limitations:**

The paper still needs many strong assumptions to analyze the problem and it is very understandable due to the theoretical nature of the work.

---

> ### Author Rebuttal · Authors · 2023-08-09
>
> # Thanks for reviewing our paper!
> **(W1: On single trajectory and condition $T>2\tau_T$)** Thanks for your comment. We are afraid that there is a misunderstanding concerning our work. Assumption 3.2 does not conflict with the statement "the transition tuples are generated from a single trajectory." As shown in Algorithm 1, all transition tuples are generated from a single trajectory. In the final results, $T$ is the total iteration number and $T>2\tau_T$ means that the theorem holds for large $T$. Each state-action pair $(s_t,a_t)$ is sampled from a single trajectory online consecutively rather than waiting for at least $2\tau_T$. One of the challenges in our analysis is exactly due to such a parsimonious online sampling, because it does not guarantee the sample distributes sufficiently close to the stationary distribution.
>
> **(Q1: On infinite action space)**  Our work is capable of addressing continuous state space but not infinite action space. We stated in Line 123 that "We consider a finite action space, whereas the state
> space can be either a finite set or an (unbounded) real vector space". To the best of our efforts, it remains challenging to investigate the infinite action space setting. One of the key obstacles we have identified is that if the action space is infinite, one cannot ensure the target critic $\omega^\ast(\theta)$ is still Lipschitz continuous (Lemma B.3). We leave this challenge setting for our future work.
>
> **(Q2: On the comparison with previous works)** We compare our work with two important previous results. We improve the work of [Chen et al., 2021] to the Markovian sample and we additionally can show convergence for the critic. We improve the work of [Oleshevsky $\And$ Gharesifard, 2023] to the infinite state space setting and Markovian sample setting. The reader is referred to the second part of Main Contribution for a detailed comparison.

---

> > ### Comment · Reviewer_2T31 · 2023-08-19
> >
> > Thank author for detailed explanation on weakness 1, I have some misunderstanding before. And I totally understand that dealing with continuous action is quite challenging. Thus I'm willing to increase my score to a 6.

---

> > > ### Author Response · Authors · 2023-08-19
> > >
> > > Thank you for recognizing our contribution!

---

### Official Review · Reviewer_9Zts · 2023-07-06

**Soundness:** 3 good
**Presentation:** 3 good
**Contribution:** 3 good
**Rating:** 6
**Confidence:** 4

**Summary:**

This paper studies the actor-critic algorithm with linear function approximation. The authors provide a single-time-scale analysis for the AC and achieve $\epsilon^{-2}$ sample complexity.

**Strengths:**

- The paper is overall well written and easy to follow
- The single-time-scale analysis with markovian noise for actor-critic is new in the literature, to my best knowledge.
- The policy gradient norm error analysis is interesting

**Weaknesses:**

- I'm confused about Assumption 3.4, I wonder how $L_\mu$ contributes to the final results.
- Assumption 3.1, though the authors mentioned a few paper using the same assumption, sounds pretty strong for me. It basically requires even the optimal policy should visit all state-action pairs, which is quite uncommon in practice.

**Questions:**

My concerns are listed in the weakness part.

---

> ### Author Rebuttal · Authors · 2023-08-09
>
> # Thanks for reviewing our paper!
> **(Q1: On the contribution of $L_\mu$ to the final results)** Thanks for seeking clarification. In the current analysis, our final results are linear to the parameter $L_\mu$. To keep $L_\mu$ in the final results, the convergence rate in Theorem 3.5 can be kept as $\mathcal{O}(L_\mu\frac{\log^2 T}{\sqrt{T}})$.
>
> **(Q2: On Assumption 3.1)** Although Assumption 3.1 is standard in theoretical analysis, we do understand that it may not be satisfied easily in practice. Note that the insight on full state-action exploration is a sufficient condition that guarantees Assumption 3.1. The converse may not be true. In addition, here the condition is presented as stronger than necessary for the sake of convenience: we actually only need the condition holds for those $\theta$ that are actually visited during learning, not for all. But such more accurate conditions are difficult to characterize analytically.

---

### Official Review · Reviewer_fvvv · 2023-07-07

**Soundness:** 3 good
**Presentation:** 3 good
**Contribution:** 3 good
**Rating:** 6
**Confidence:** 4

**Summary:**

This paper provides a finite-time analysis of single-timescale, single-sample, average-reward actor-critic with a linear critic under Markovian sampling. It is established that the standard scheme achieves $\epsilon$-approximate stationarity with $\widetilde{O}(\epsilon^{-2})$ sample complexity. The analysis is achieved by extending the small-gain analysis of [A. Oleshevsky & B. Gharesifard, _A small-gain analysis of single-timescale actor-critic_. SICON 2023], which was for the discounted setting, to the average reward setting. This is accomplished by leveraging a uniform ergodicity assumption (Assumption 3.2, line 202) and associated analysis (like, e.g., that of [Y. F. Wu, W. Zhang, P. Xu, Q. Gu, _A finite-time analysis of two time-scale actor-critic methods_. NeurIPS, 2020]) to handle the Markovian sampling that becomes particularly important in the average-reward setting. It is furthermore claimed that the results apply to problems with continuous state spaces. The sample complexity achieved matches the state of the art for finite-time analyses of actor-critic methods, yet applies to the single-timescale, single-sample, average-reward, Markovian sampling setting, which was previously an open question.

**Strengths:**

As stated in the summary, this work establishes that single-timescale, single-sample, average-reward actor-critic under Markovian sampling achieves $\epsilon$-stationarity with $\widetilde{O}(\epsilon^{-2})$ sample complexity. This shows that this version of actor-critic matches the state of the art, resolving what was a previously an open question. Despite the analysis presented in the appendix being notation-heavy and somewhat difficult to follow, and though there are issues in the exposition and assumptions that need to be addressed (see weaknesses and questions below), the main steps in the theoretical results appear to be sound. The analysis appears to build off of the small-gain analysis of [Oleshevsky & Gharesifard, 2023], combined with elements of previous two-timescale analyses for average-reward actor-critic with the uniform ergodicity assumption such as [Wu et al., 2020], as well as some algebraic manipulations of inequalities. Though of uncertain practical utility, this contribution is definitely of interest to the theoretical reinforcement learning community.

**Weaknesses:**

I have some concerns about: (i) clarity about the innovation required in the analysis; (ii) the satisfiability of the assumptions made; and (iii) the dependence of the main result presented in Step 4 in lines 331-339 on having access to potentially unknown problem-specific constants when designing stepsizes. Addressing these issues or being clear about any limitations will likely strengthen the paper.

(i) As stated above, the analysis appears to build off of [Oleshevsky & Gharesifard, 2023] to handle the interconnectedness of the single-timescale analysis, elements of two-timescale analyses like [Wu et al., 2020] to handle Markovian sampling, as well as algebraic manipulations of inequalities to combine it all together. However, it is unclear from the main body whether the analysis is a straightforward (albeit complicated) combination of these existing methods, or if some critical new insight and innovation are required. If the former, the significance of the contribution is weakened. If the latter, it should be stated more clearly. Also, the abstract claims that the analysis applies to continuous state spaces, and on lines 73-74 it is stated that achieving this requires "significantly non-trivial effort in the analysis". It is not clear where in the analysis this non-trivial effort takes place, however. If serious additional effort was required, it should be clearly stated exactly where it occurs.

(ii) Two of the assumptions call into doubt the applicability of the result to continuous state spaces. On lines 198-199, it is stated that Assumption 3.1 holds under certain conditions in the tabular case, but applicability to the continuous state space case is not mentioned. The reader wonders: does Assumption 3.1 hold when $|S| = \infty$? In addition, lines 220-221 state that Assumption 3.4 holds for the finite state-action space setting. Again: what about when $|S| = \infty$? Since the analysis relies on these assumptions, it must be shown that they hold in the continuous state space setting for the corresponding claims in the abstract and introduction to be true. Finally, though uniform ergodicity (Assumption 3.2) is commonly assumed in average-reward analyses, the recent two-timescale analysis of [W. A. Suttle, A. S. Bedi, B. Patel, B. M. Sadler, A. Koppel, D. Manocha, _Beyond exponentially fast mixing in average-reward reinforcement learning via multi-level monte carlo actor-critic_. ICML 2023] removes the need for this assumption. It would be helpful to see a discussion of why Assumption 3.2 is still required in the single-timescale analysis or how it might be eliminated.

(iii) On lines 334-337, it is pointed out that, for Step 4 (cf. lines 331-339) to work, a certain set of inequalities (line 335) needs to be satisfied. It is stated that this condition can be satisfied by choosing the stepsize ratio $c$ to be smaller than some threshold. On line 556 in the appendix, this threshold is given, but it involves the problem-specific constants $\lambda, L_*, G, B$. It thus appears necessary that we have oracle knowledge or that an additional estimation procedure is required for the specific stepsize scheme to be operable.

**Questions:**

* What is the key innovation required in the analysis?
* What is the effort required in accommodating continuous state spaces?
* Under what conditions do Assumptions 3.1 and 3.4 hold for continuous state spaces?
* Why is Assumption 3.2 necessary? Or can it be eliminated?
* How do we choose the stepsize ratio $c$ in practice?

**Limitations:**

Aside from the weaknesses and questions described above, the authors have adequately addressed the limitations of the work.

---

> ### Author Rebuttal · Authors · 2023-08-09
>
> # Thanks for reviewing our paper!
> **(Q1: Key innovation in the analysis)**
> Thanks for seeking clarification. The key innovation lies in conducting a comprehensive analysis of each error term. In two-timescale actor-critic [Wu et al., 2020], convergence is deduced relying on multiplying the error term by the diminishing stepsize ratio $\frac{\alpha_t}{\beta_t} \rightarrow 0$ as $t\rightarrow \infty $. However, in the single-timescale setting, this ratio is constant $\frac{\alpha_t}{\beta_t}=c$, requiring a more in-depth investigation and tighter analysis of each error term to establish convergence. These lead to non-trivial proofs that are distinct from the two main references, and we highlighted them in the proof sketch.
>
> Taking the error term $y_t(J(\theta_t)-J(\theta_{t+1}))$ for example, the two-timescale work bounds it using the $L_J$-Lipschitz continuity of $J(\theta)$. In the single-timescale approach, solely relying on Lipschitz continuity is still too loose. To overcome this, we delve deeper and discover that the gradient of $J(\theta)$ also possesses Lipschitz continuity (Lemma B.2), which means $J(\theta)$ is $L_{J'}$-smoothness. With this new insight, we successfully bound the term $y_t(J(\theta_t)-J(\theta_{t+1}))$ with $\sqrt{Y_TG_T}$, thereby establishing a solvable interconnected system. Together with many other careful analyses, we achieve much tighter bounds and are able to establish convergence of the more challenging single time-scale algorithm.
>
> **(Q2: On the effort required in accommodating continuous state spaces)**
> We note that moving from a finite state space to an infinite state space takes significant and nontrivial efforts in analysis. This is due to the fact that some established results need to rely on intrinsic problem constants such as many Lipschitz constants that rely on the finite size of the state space ($|\mathcal{S}|$), which however becomes infinite in the infinite state space scenario. Additionally, existing analysis concatenates all state-action pairs to create a finite-dimensional feature matrix and often requires summation over all states [Oleshevsky $\And$ Gharesifard, 2023]. These analyses will not be possible when the state space is uncountable. Moreover, some convenient properties no longer exist in the continuous state space setting. For example, the uniform boundedness of Hessian of the problem matrix $A_\theta$ no longer holds ([Oleshevsky $\And$ Gharesifard, 2023], Lemma 4.8). These fundamental challenges require completely different analysis techniques.
>
> **(Q3: On Assumptions 3.1 and 3.4)** In fact, Assumption 3.1 is a fundamental regularity condition, which is often made to guarantee the problem’s solvability in the linear function approximation case and on continuous state space [Wu et al., 2020].
>
> For Assumption 3.4, $\nabla_\theta \mu_\theta(s)$ is a Jacobian matrix such that its $d$th column is the gradient of the $d$th action dimension of the policy with respect to the policy parameters $\theta$. Assumption 3.4 is equivalent to $L_\mu$-smoothness of the stationary state distribution. It's difficult to characterize a general condition under which this assumption holds. But there are cases for which it holds, for example, linear systems with linear state feedback policy.
>
> **(Q4: On Assumption 3.2)** Assumption 3.2 can not be eliminated for the analyzed algorithm. The provided reference analyzes a multi-level Monte Carlo actor-critic which notably samples $2^{j_t}$ state-action pairs for each policy $\pi_{\theta_t}$ update. Taking the average of these multi-samples effectively reduces the statistical error (deviation from stationary distribution). However, we analyze the most general and fundamental **single-sample** single-timescale online actor-critic, which means for each policy update, we only sample one state-action pair. Consequently, the introduction of Assumption 3.2 becomes imperative in order to characterize the disparity between the distribution of the Markovian sample and its corresponding stationary distribution.
>
> **(Q5: On choosing the stepsize ratio in practice)** Due to the theoretical nature of the work, we characterize an upper bound of the stepsize ratio $c$ depending on unknown problem-specific constants, which is a sufficient condition for convergence. In practice, one can choose a relatively small $c$ to guarantee convergence, just like in solving many machine learning problems using SGD methods, people typically choose a small stepsize to have a better convergence guarantee and performance.

---

> > ### Comment · Reviewer_fvvv · 2023-08-14
> > **Clarification request: continuous spaces**
> >
> > Thanks very much for the responses. In the abstract on line 6 and contributions on lines 73-74, you mention that you target the continuous spaces setting. In your response to (Q2) above, you have nicely outlined the **challenges** inherent in the continuous space setting, but I still don't understand how you **overcame** those challenges.
> >
> > Two questions:
> > 1. Does your analysis handle the continuous spaces setting?
> > 2. Can you point out specific locations (e.g., with sections or line numbers) in the analysis where you made "significantly non-trivial effort" to handle this setting? Also, can you explain how these efforts are completely different from previously used techniques?

---

> > > ### Author Response · Authors · 2023-08-15
> > > **Thanks for seeking clarification!**
> > >
> > > **(Q1: On continuous **state** space setting)** Thanks for your careful reading and efforts to help us elucidate our contribution. In both places pointed out in your comments, we had highlighted that we consider continuous **state** space setting.  Indeed, our analysis can handle the continuous state space setting.
> > >
> > > **(Q2: On the effort to handle continuous setting)** Thanks for seeking clarification. [Oleshevsky $\And$ Gharesifard, 2023] can only handle finite state spaces. Their analysis heavily relies on the construction of a finite-dimensional feature matrix $\Phi$ whose column dimension is equal to the number of all state-action pairs. In particular, many key properties and bounds were established relying on this matrix. For example, they establish the implicit upper bound for critic error (Lemma 5.14) based on the Lipschitzness of actor update (Lemma 5.5) and the uniform boundedness of Hessian of the problem matrix $A_\theta$ (Lemma 5.8). These properties no longer hold once moving onto the continuous state space setting. Basically, all their proofs of deriving the implicit bounds are inapplicable to the continuous state space case.
> > >
> > > In contrast, our approach directly establishes the implicit bounds of $Y_T$, $Z_T$, and $G_T$ (see Theorem C.2, Theorem C.5, and Theorem C.7, respectively) through a series of novel characterization of error terms that are completely different from [Oleshevsky $\And$ Gharesifard, 2023]. For example, our implicit upper bound for critic error (Theorem C.5) was established using the Lipschitzness of critic target $\omega^\ast$ (Lemma B.3) and the $L_s$-smoothness of $\omega^\ast$ (Lemma B.4), which do not depend on the finite state space assumption. Moreover, our characterization controls the critic error (referred to as $Z_T$ in our paper) by the product of both critic error and reward estimation error (referred to as $Y_T$, detailed in Line 508, equation (18)), which is a key step to guarantee convergence, and more importantly, it holds under continuous state space. Similarly, we developed closer coupling of all three errors in all three implicit bounds, which eventually enables the establishment of convergence. Overall, our main efforts and technical contribution lie in the proper decomposition and bounding of various error terms that eventually result in a solvable interconnected system. The interconnected system formulation may seem incremental as compared to [Oleshevsky $\And$ Gharesifard, 2023] if without diving into the detailed proof, but we would like to emphasize that the true challenge and difficulty lie in how to characterize those implicit bounds for critic, actor, and reward estimation errors that constitute the interconnected system under a more challenging continuous state space and Markovian sampling setting.
> > >
> > > We hope the above better clarifies our contribution.

---

> > > > ### Comment · Reviewer_fvvv · 2023-08-15
> > > >
> > > > Thanks for your response.
> > > >
> > > > Based on your response Q2, it appears that Theorem C.5 depends on Lemmas B.3 (from ref [28]) and B.4 (from ref [8]) to handle continuous $S$. If so, my concern is that previous results are being used to handle continuous $S$, and that, as far as the current work is concerned, significant additional effort was **not** required to handle continuous $S$. This would undermine the contribution claimed in lines 6 and 73-74. What do you think?
> > > >
> > > > Please note that I appreciate that significant effort was needed to analyze the system of inequalities to achieve the single-timescale analysis. I believe this is unrelated to handling continuous $S$, however.
> > > >
> > > > As a side note, in order to be able to apply Lemmas B.3 and B.4 in the continuous $S$ case, I suspect some condition on the critic feature map $\Phi$ is required.

---

> > > > > ### Author Response · Authors · 2023-08-16
> > > > > **Thanks for your comment!**
> > > > >
> > > > > **(C1: On continuous state space setting)** Thanks for your comment. We realized our last response caused a misunderstanding. To address the continuous state space setting, our efforts are not only spent on analyzing the system of inequalities (implicit bounds), but also significantly more on establishing the system of inequalities. Although our analysis utilized some existing properties as our "building blocks" (namely, Lemma B.3 and Lemma B.4, and in fact, all the Lemmas in Appendix B Preliminary Lemmas are needed overall), it does not undermine our contribution. Because the "significant additional effort" was spent on the proper decomposition and bounding of various error terms, the carefully designed coupling of the implicit bounds, and the restriction of utilizing only those building blocks that hold in continuous state space.
> > > > >
> > > > > As a comparison, [Oleshevsky $\And$ Gharesifard, 2023] (abbreviated [OG] in the following) also developed their proof based on Lemma B.3 (Lemma 5.10 in their paper) and Lemma B.4 (Lemma 5.11 in their paper), and many other existing results, but also based on many properties that cannot be generalized to continuous state space. Our building blocks (Appendix B: Preliminary Lemmas) are only a subset of those utilized by [OG]. For example, as we mentioned in the last reply, the Lipschitzness of actor update ([OG, Lemma 5.5]), on which their proof heavily depends, no longer holds in continuous state space. Hence, our proof as well as the system of inequalities differ significantly from [OG]. The root of these fundamental differences is the continuous state space setting.
> > > > > Hence, we mention in the paper that significant additional effort is required in handling continuous state space.
> > > > >
> > > > > **(C2: On the side note)** Yes, you are right. In order to apply Lemma B.3 and B.4 in continuous state space, we do need Assumption 3.1 on the system matrix $A_\theta$ (constructed using the feature vectors defined on continuous state space, see Eq. (4)). However, the original $\Phi$ as defined in [OG] does not exist in the continuous setting (We don't have the variable $\Phi$ in our paper). [OG] has both $\Phi$ and $A_\theta$ but we only have $A_\theta$.
> > > > >
> > > > > Hope this further clarifies our contribution.

---

> > > > > > ### Comment · Reviewer_fvvv · 2023-08-21
> > > > > >
> > > > > > The author responses have sufficiently addressed my concerns and I am increasing my score.

---

> > > > > > > ### Author Response · Authors · 2023-08-22
> > > > > > >
> > > > > > > Thank you for recognizing our contribution!

---

### Official Review · Reviewer_bt4a · 2023-07-18

**Soundness:** 3 good
**Presentation:** 3 good
**Contribution:** 3 good
**Rating:** 6
**Confidence:** 4

**Summary:**

This works provides finite sample analysis for single timescale actor critic.

**Strengths:**

A finite sample analysis for single timescale actor critic under Markovian noise and infinite state space is definitely a notable contribution to the community.

**Weaknesses:**

My biggest concern is the correctness of this work. This work is a resubmission from ICML 2023. In the ICML round, all reviewers agree that this is a good paper until one expert reviewer that seems to really be in the field pointed out one critical technical error in the proof, leading to the rejection. But I unfortunately do not have access to the ICML reviews now so I will recommend "rejection" for now. If the authors provide the following information in the rebuttal, I am more than happy to evaluate the work again.

1. What errors did the reviewer pointed out in the ICML round (it would help a lot if a copy of the original review can be provided, assuming the authors still have access to it)?
2. Was the reviewer wrong or the authors were wrong?
3. If the authors were wrong, how is the bug fixed in this version? If the reviewer was wrong, what mistake did the reviewer make?


It would help a lot if the authors could answer the above questions in details and in a self-contained way.

**Questions:**

See above

**Limitations:**

See above

---

> ### Author Rebuttal · Authors · 2023-08-09
>
> # Thanks for reviewing our paper!
> **(Q1: What error was pointed out in the ICML submission)** Thanks for reviewing our work again. We are glad to show our refined results. The original comment in the ICML round is: "However, the authors utilize a very quick but wrong inequality to extend the analysis technique in [1] to inifite state space case: When analyzing the interconnected system (line 1122), the term "$bZ_T$" should be corrected to "$b\sqrt{Z_T}$" according to equation (18)
> . This is a quite nontrivial mistake. Since the term "b" is of constant order (line 1112) under the infinite state space setting, the convergence rate of
>  will NOT be $\widetilde{\mathcal{O}}(1/\sqrt{T})+\mathcal{O}(T)$
> . Instead, it will be an absolute constant independent of $T$
>  if we apply the authors' trick of Young's inequality. As a consequence, the whole remaining proof should be wrong." So the error occurred at Line 1122 (now is fixed at Line 557), where according to equation (18) (now still equation (18)), the correct term should be $\sqrt{Z_T}$ but we wrongly took it as $Z_T$ and performed the subsequential analysis. Now we have fixed this careless mistake and proved the results correctly.
>
> **(Q2: Who was wrong)** We were wrong. During the rebuttal, we realized that simply choosing appropriate stepsizes cannot fix the bug. We then have done a major revision after the rebuttal.
>
> **(Q3: How we fixed the bug in NeurIPS submission)** We have developed a new proof under an additional Assumption 3.4 of the $L_s$-smoothness property of the critic target $\omega^\ast(\theta)$, where a different and tighter system of inequalities was developed. The smoothness property is used to bound the term $\langle z_t, \omega^\ast(\theta_t)-\omega^\ast(\theta_{t+1})\rangle$, which tracks both the critic estimation performance $z_t$ and the difference between the drifting critic targets $\omega^\ast(\theta_t)$. In particular, we bound it by $\sqrt{Z_T(2Y_T+8Z_T)}$ (see Theorem C.5). Then we were able to derive a solvable interconnected system, the solution of which leads to our main results. In the ICML version, we erroneously applied a crude bound of $\sqrt{Z_T}$ to bound the same term. However, we mistakenly treated $\sqrt{Z_T}$ as $Z_T$ in our attempts to solve the interconnected system. Indeed, if it were $Z_T$, the system is unsolvable and cannot show convergence as pointed out by the ICML reviewer.

---

> > ### Comment · Reviewer_bt4a · 2023-08-11
> > **Assumption too strong**
> >
> > I am afraid Assumption 3.4 is way too strong. [8] does have such an assumption. But I quote from [8]
> >
> > > Assumption 11 is the counterpart of Assumption 10 that is made for the stationary distribution μθ(a|s). Note that the existence of ∇μθ(s) has been shown in [2]. In this case, under Assumption 10, i) and iii) of Assumption 11 can be obtained from the sensitivity analysis of Markov chain; see e.g., [32, Theorem 3.1]. While we cannot provide a justification of (ii), we found it necessary to ensure the smoothness of the lower-level critic solution y∗(θ).
> >
> > [8] does **NOT** provide a justification for the smoothness of the stationary distribution.
> >
> > The argument the authors make is "This assumption holds for the finite state-action space setting". I would like to see a proof, since [8] does not prove this.
> >
> > I think it's ok for [8] to use this assumption because RL is merely 1 of  their 4 applications of a more general result, but RL is all of this work.

---

> > > ### Author Response · Authors · 2023-08-11
> > >
> > > Thanks for your comment. The proof of Assumption 3.4 under finite state-action space setting can be found in Lemma 14 of [1]. We forgot to cite this reference and will include it in our revised version.
> > >
> > > [1] Qijun Luo and Xiao Li. Finite-time analysis of fully decentralized single-timescale actor-critic. arXiv preprint arXiv:2206.05733, 2022

---

> > > > ### Comment · Reviewer_bt4a · 2023-08-15
> > > > **Thanks for the response**
> > > >
> > > > I have increased my score to 6 in light of the new reference.

---

> > > > > ### Author Response · Authors · 2023-08-15
> > > > >
> > > > > Thank you for recognizing our contribution!

---

### Decision · Program_Chairs · 2023-09-21

**Decision:**

Accept (poster)

**Comment:**

This paper offers a finite-time analysis of the actor-critic algorithm in the context of single-timescale, single-sample, average-reward settings with Markovian sampling. The work extends previous small-gain analyses from discounted to average-reward settings by leveraging a uniform ergodicity assumption. The paper achieves state-of-the-art sample complexity of actor-critic methods. The authors have addressed the concerns raised by the reviewers and all of the reviewers have recommended acceptance for the work.